# Tracking transcription–translation coupling in real time

Nusrat Shahin Qureshi[1,2] & Olivier Duss[1,2]✉

A central question in biology is how macromolecular machines function cooperatively. In bacteria, transcription and translation occur in the same cellular compartment, and can be physically and functionally coupled[1–4]. Although high-resolution structures of the ribosome–RNA polymerase (RNAP) complex have provided initial mechanistic insights into the coupling process[5–10], we lack knowledge of how these structural snapshots are placed along a dynamic reaction trajectory. Here we reconstitute a complete and active transcription–translation system and develop multi-colour single-molecule fluorescence microscopy experiments to directly and simultaneously track transcription elongation, translation elongation and the physical and functional coupling between the ribosome and the RNAP in real time. Our data show that physical coupling between ribosome and RNAP can occur over hundreds of nucleotides of intervening mRNA by mRNA looping, a process facilitated by NusG. We detect active transcription elongation during mRNA looping and show that NusA-paused RNAPs can be activated by the ribosome by long-range physical coupling. Conversely, the ribosome slows down while colliding with the RNAP. We hereby provide an alternative explanation for how the ribosome can efficiently rescue RNAP from frequent pausing without requiring collisions by a closely trailing ribosome. Overall, our dynamic data mechanistically highlight an example of how two central macromolecular machineries, the ribosome and RNAP, can physically and functionally cooperate to optimize gene expression.

A central but understudied question in biology is how different macromolecular processes functionally cooperate. In bacteria, transcription performed by RNA polymerase (RNAP) and translation performed by the ribosome occur in the same cellular compartment, which enables functional coupling of these two processes[1–4]. Such coupling provides opportunities for regulation that have been shown to make transcription more efficient in the presence of active translation. For example, a ribosome closely trailing behind RNAP inhibits Rho-dependent transcription termination by blocking access to the Rho helicase[3]. In vitro, a closely trailing ribosome can also prevent RNAP pausing by suppressing hairpin-stabilized pause formation, or rescue paused RNAPs from backtracking by ribosome–RNAP collisions[10–12].

Transcription and translation rates are also known to be correlated[12], raising the question of how elongation speeds in both processes are communicated between the two macromolecular machineries. One possibility is that communication happens by the alarmone (p)ppGpp, which affects both transcription and translation elongation rates[13]. Alternatively, direct physical interactions between the RNAP and a closely trailing ribosome could allow coordinating elongation rates[10–12]. Indeed, several studies, including recent high-resolution structures, demonstrate a physical link that establishes coupling between both machineries (termed expressome in this study) that is either mediated by direct interactions between the RNAP and the ribosome or by factor-mediated interactions mainly via the transcription factors NusG and/or NusA[5–10].

Those structures broadly fall into one of three categories: collided, coupled and uncoupled. The collided state, which contains the shortest possible intervening mRNA spacer between ribosome and RNAP, allows neither NusA binding nor NusG-based bridging and potentially lacks the ω subunit of the RNAP[6,7]. Furthermore, it could not be captured in vivo under normal growth conditions and could be isolated only when the cells were treated with the transcription inhibitor pseudouridimycin, artificially stalling the RNAP[8]. This raises the question of whether ribosome–RNAP collisions, whereby the ribosome pushes the RNAP forwards as recently suggested[9–11], provide the prevalent mechanism for reactivating paused RNAPs in vivo. By contrast, the coupled state is compatible with known functional aspects of transcription and translation and NusA–NusG bridging, and therefore could provide a functional state during which communication between both machineries could occur. However, cryogenic electron microscopy (cryo-EM) and cryogenic electron tomography (cryo-ET) structures only provide structural snapshots of a dynamic process and do not allow direct monitoring of activity. They also do not provide information on whether physical coupling can occur only when the two machineries are close along the mRNA sequence or whether coupling could also occur via mRNA looping.

Here we reconstitute the complete active transcription–translation coupling system in vitro. We investigate the molecular mechanism of transcription–translation coupling by combining bulk biochemical experiments with multi-colour single-molecule fluorescence

[1]Structural and Computational Biology Unit, European Molecular Biology Laboratory, Heidelberg, Germany. [2]Molecular Systems Biology Unit, European Molecular Biology Laboratory, Heidelberg, Germany. ✉e-mail: olivier.duss@embl.de

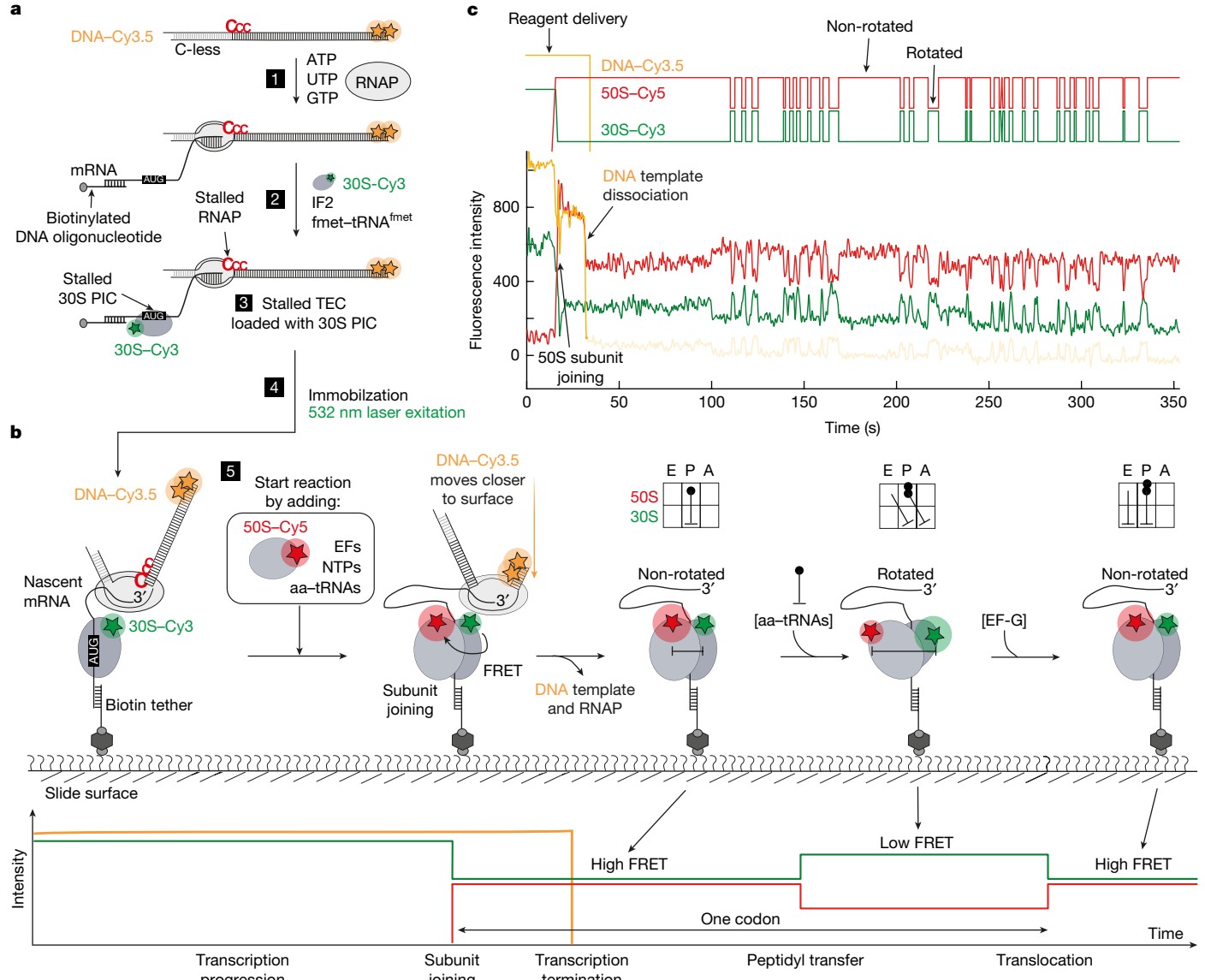

**Fig. 1 | Single-molecule assay for real-time tracking of co-transcriptional translation elongation. a**, Experimental design for preparation of transcription–translation complexes. C-less, sequence lacking cytidines; CCC, three consecutive cytidines. **b**, Single-molecule assay for observing transcription and translation of the same mRNA molecules in real time. EFs, elongation factors. Grids at top right depict tRNA conformation in the A, P and E sites of the ribosome during the translation elongation cycle. Scheme at bottom illustrates the expected fluorescence signals. **c**, Representative single-molecule trace showing fluorescence intensities of transcription (yellow) and translation (red and green). Trace was smoothed for visualization. An idealized trace is displayed on top.

microscopy assays to place various published structural snapshots along a reaction path. By simultaneously tracking transcription and translation elongation and the physical and functional coupling between both machineries, we find that the ribosome slows down while colliding with the RNAP, providing a biophysical explanation for why the collided state has so far not been detected in vivo and may only occur as an infrequent state—for example, when RNAP is stalled on a DNA lesion[14]. We see that physical coupling can occur during active transcription elongation, by looping-out of the mRNA. This physical coupling allows the ribosome to activate a stalled RNAP by long-range communication, without the need for collisions, and provides an explanation for how the central machineries in gene expression coordinate their activities.

## Imaging of an active transcription–translation system

To directly visualize the process of transcription–translation coupling in real time, we reconstituted a complete and active in vitro transcription–translation system. We established single-molecule fluorescence microscopy assays that enable us to simultaneously track transcription and translation elongation occurring on the same nascent mRNA. To this end, we prepared a stalled transcription elongation complex (TEC) (Fig. 1a and Methods) consisting of a 3′-2×Cy3.5-labelled DNA template, *Escherichia coli* RNAP and a nascently transcribed mRNA containing a ribosome-binding site[15] (RBS). We then loaded a Cy3-labelled 30S ribosomal subunit (labelled at an extension on helix 44 (h44) of the 16S rRNA) in the presence of the initiation factor IF2a and fmet–tRNA[fmet] on to the nascent mRNA[16,17]. The resulting translation pre-initiation complex (PIC) was immobilized via the 5′ end of the nascent mRNA to a polyethylene glycol (PEG)–biotin-functionalized glass surface for single-molecule multi-colour imaging.

Elongation of both machineries was triggered at the beginning of imaging by addition of NTPs, Cy5-labelled 50S subunit[17] (labelled at an extension on helix 101 (h101) of the 23S rRNA), elongation factor-G (EF-G), ternary complex (elongation factor thermo unstable

(EF-Tu)-aminoacyl-tRNA-GTP) and elongation factor thermo stable (EF-Ts) (Fig. 1b). Consistent with previous reports[18–21], subunit joining was observed by the appearance of a Cy3–Cy5 Förster resonance energy transfer (FRET) signal (Fig. 1c and Extended Data Fig. 1a,b) immediately following the addition of all components (within 3 s for 50% of the molecules at 50 nM Cy5–50S). Translation elongation at single-amino-acid resolution can be monitored by the 30S-h44–50S-h101 FRET pair under our experimental conditions (Methods), which reports on the two main steps of translation elongation—peptidyl transfer and translocation (Fig. 1b)—in which the ribosome transitions from a non-rotated, high-FRET state to a rotated, low-FRET state[16,17,22]. We monitored translation of individual codons over several minutes and reliably quantified the lifetime in the rotated and non-rotated states at up to 500 nM aminoacyl-tRNA (aa–tRNA) and 160 nM EF-G concentrations (Extended Data Fig. 1g) with values one order of magnitude lower than in vivo[23,24], but in agreement with previous single-molecule experiments[19,25].

Simultaneously to translation elongation, we monitored transcription progression by following the DNA template signal. After delivery of all four NTPs, transcription immediately resumes and could be monitored over several minutes. Transcription termination is characterized by a single-step disappearance of the Cy3.5 signal[15]. Plotting the time from NTP addition until DNA template dissociation for individual molecules allowed us to calculate average transcription rates (Extended Data Fig. 1g), which are NTP concentration dependent and are in agreement with previous bulk in vitro transcription and single-molecule studies, as well as in vivo rates[12,15,26,27]. Overall, our combined in vitro reconstituted transcription–translation system recapitulates the activities of each system previously studied in isolation.

## Ribosome slowdown upon RNAP collision

The full control of all components in our single-molecule assays enables us to compare the translation elongation kinetics of single ribosomes translating either unhindered on an mRNA (if the RNAP transcribes faster than the ribosome translates the mRNA) or in the presence of a paused RNAP leading to a collision between the ribosome and the RNAP. We used a DNA template that allowed us to stall the ribosome and the RNAP with 46 nucleotides (nt) between the ribosome P-site to the RNAP active site (Fig. 2a, Supplementary Tables 1 and 2 and Methods). This state was previously structurally characterized and classified as a 'coupled state TTC-B'[7] (Fig. 2g). On the basis of the cryo-EM structures, this coupled state allows the translation of two to three amino acids until both machineries substantially assume a 'collided state' and maximally six amino acids until no intervening mRNA separates both machineries[6,7,11] (TTC-A, 28 nt separating P-site and RNAP active site; Fig. 2g). Delivery of all translation elongation factors together with NTPs triggered simultaneous translocation of both machineries (Fig. 2b). The actively transcribed mRNA was also being translated and we were able to track translation of up to 31 codons until the stop codon was reached (Fig. 2d).

To investigate the effect of a stalled RNAP on ribosome translocation, we next repeated the experiment without delivery of NTPs (Fig. 2c). As expected, withholding NTPs during the single-molecule experiment led to collision of the ribosome into the artificially paused RNAP. Under these conditions, the largest fraction of ribosomes (79%) completed 4 to 6 translation cycles before the end of the experiment or before photobleaching occurred (Fig. 2d). Comparing the median dwell times for translation of the individual codons of unhindered ribosomes with ribosomes that are in the process of colliding with the RNAP showed that the ribosome slows down while colliding with the RNAP (Fig. 2e, compare blue with orange data). The non-rotated and rotated state dwell times gradually increase once the ribosome gets to five to six amino acids away from the stalled RNAP (Fig. 2e and Extended Data Fig. 2a–c). In the absence of a stalled RNAP, the dwell times for the non-rotated state (rate-limited by aa–tRNA arrival) or rotated state (rate-limited by EF-G arrival) can be fitted to a single-exponential function in agreement with

a single rate-limiting step[28] (Extended Data Fig. 2a). By contrast, under colliding conditions, the dwell times no longer follow single-exponential kinetic behaviour, suggesting additional rate-limiting steps while transitioning (Fig. 2g) from coupled to collided state (Fig. 2f and Extended Data Fig. 2a–c). By increasing the aa–tRNA and EF-G concentrations tenfold during colliding conditions, ribosome slowdown is not substantially affected (Extended Data Fig. 2d), suggesting that the physical basis for slowdown is not due to steric interference affecting the delivery of aa–tRNA and/or EF-G but due to the increased mechanical force required to perform a translation cycle[10] while structurally being in or transiently visiting the collided state[6,7]. Finally, we observe that under colliding conditions (Extended Data Figs. 1h,i and 2e), the ribosome preferentially ends up in the rotated state (50 ± 2% of molecules compared with 17 ± 5% for elongating conditions), similarly to what has been found in vivo for a pseudouridimycin stalled expressome[8].

## Intervening mRNA length affects coupling

Available cryo-EM structures represent either collided states or coupled states up to an intervening mRNA length of 47 nt between the ribosome P-site and RNAP active site (6 amino acids inter-translation distance; Fig. 2g, left structure). Structures with longer intervening mRNA have so far not been obtained in vitro[6,7,9]. Furthermore, cryo-ET structures obtained from active expressomes in vivo did not show any density for the intervening mRNA[8]. This raises the question of whether the ribosome and the RNAP can also physically couple if their intervening mRNA length is longer than can be visualized by current methods, and what the kinetics and function of such coupling–uncoupling interactions would be.

To this end, we developed single-molecule assays to directly track in real time the coupling and uncoupling dynamics between a stalled RNAP and a stalled ribosome at various lengths of intervening mRNA between both machineries (46–457 nt between P-site and RNAP active site; hereafter called mRNA-46 to mRNA-457; Fig. 3a). We labelled the ribosome and the RNAP with a donor and acceptor dye, such that FRET is only observed if both machineries are in a coupled state (Fig. 3b and Extended Data Fig. 3). The ribosome was labelled at its mRNA-entry channel (rRNA extension at helix 33a (h33a)[16,29]) with a donor Cy3 dye and the RNAP at the N terminus or C terminus of the β' subunit with an acceptor Cy5 dye using ybbR-tag-mediated dye coupling[30,31] (Methods). Labelling both the ribosome[16,17,25] and the RNAP (Extended Data Fig. 3b) at these positions did not detectably affect activity, and results in an inter-dye distance of 34–57 Å if the RNAP and the ribosome are in a coupled state (Extended Data Fig. 3a).

After immobilizing the pre-assembled expressomes on a microscope slide (Methods), we used alternating-laser excitation[32] (ALEX) at wavelengths of 532 and 638 nm (Fig. 3d,e) to distinguish uncoupling events from photobleaching (Methods). By plotting the FRET efficiency distributions, we could describe the inter-ribosome–RNAP distance distribution for the ensemble of the expressome molecules. We observe a gradual shift of the FRET efficiency distribution towards smaller values in agreement with increasing RNAP–ribosome separation when increasing intervening mRNA length (Fig. 3f and Extended Data Fig. 4a). For mRNA-46, we find two well-defined peaks, which we assign to uncoupled and coupled expressomes, in agreement with the reported cryo-EM structures[6,7] (Extended Data Fig. 3). By contrast, for mRNA-85, we observe a broadening of the distribution, which shifts towards a single peak with FRET efficiency ($E_{FRET}$) ≈ 0 for the longest mRNA-457. While hidden Markov modelling (HMM) enables the assignment of various states (three states predicted; Methods), dozens of different cryo-EM structures were determined describing a broad structural distribution of coupled expressomes[6,7] (Extended Data Fig. 3c,d and Source Data), including additional loosely coupled states (TTC-LC) detected with intervening mRNA sequences longer than 50 nucleotides[33]. Therefore, unambiguous detection and assignment of the various structural states to specific $E_{FRET}$ values is not possible.

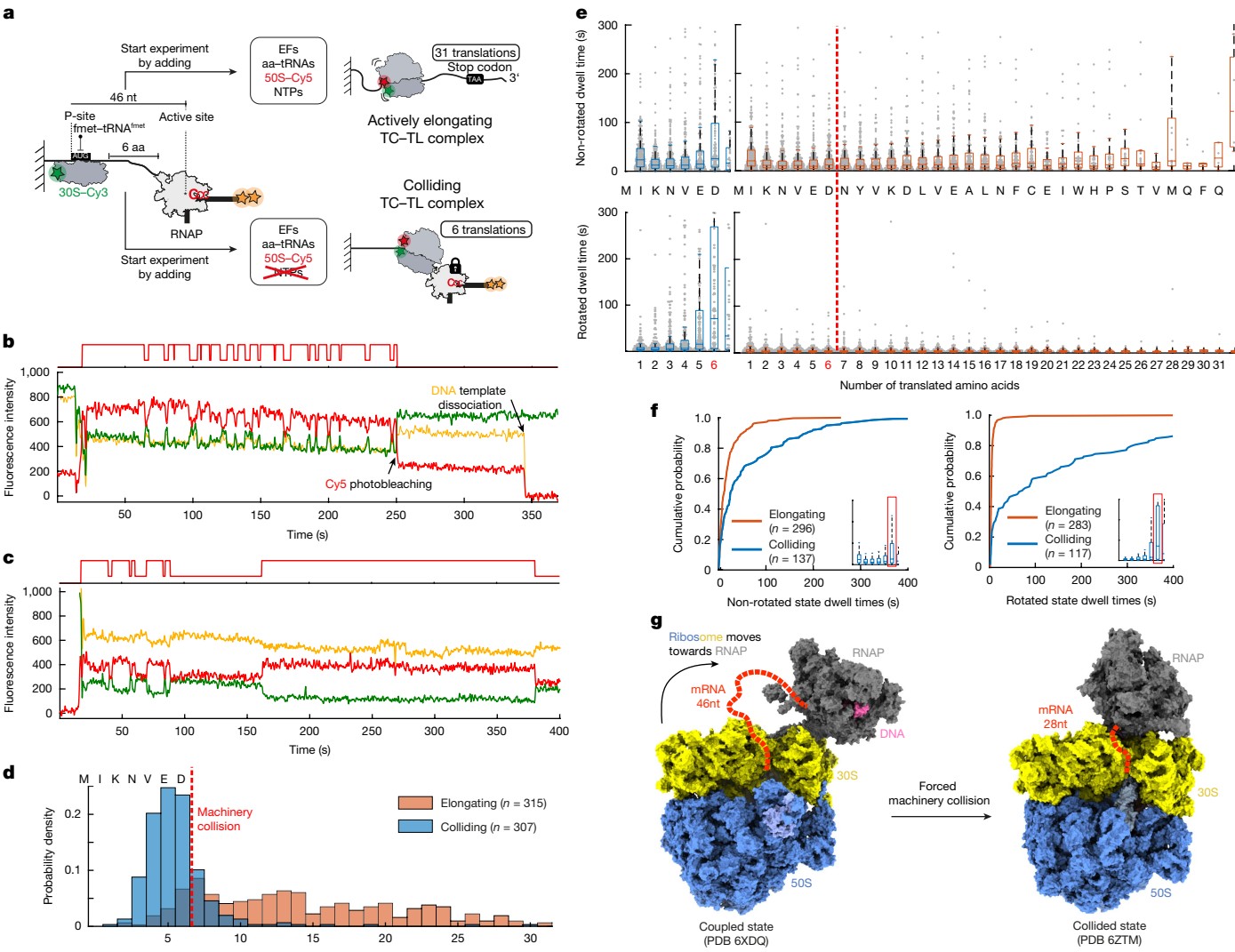

**Fig. 2 | Ribosome slows down while colliding with RNAP. a**, Experimental design. TC–TL complex, transcription–translation complex. **b,c**, Representative, smoothed traces for elongating (**b**) and colliding conditions (**c**). **d**, Probability density histograms for number of translated amino acids in colliding and elongating conditions. **e**, Box plots of non-rotated (top) and rotated (bottom) dwell times during colliding (left, blue) and elongating (right, orange) conditions with 150 nM total aa–tRNAs and 50 nM EF-G. Boxes show median (centre line), 25th percentile (bottom edge), 75th percentile (top edge) and whiskers extend to 1.5 times the interquartile range. Note that not all outliers are visible on this *y*-axis scale. The red dashed line denotes the position at which ribosome and RNAP would collide under colliding conditions. **f**, Cumulative probability distribution of the dwell times for the sixth amino acid incorporation for non-rotated state (left) and rotated state (right). Number of analysed molecules (*n*) is indicated (**d**,**f**) or reported in Methods (**e**). Data were combined from three biological replicates (**d**–**f**). **g**, Expressome structures visualizing the start and end states of the ribosome slowdown after colliding with the RNAP, represented by the coupled state (Protein Data Bank (PDB): 6XDQ) transitioning into the collided state (PDB: 6ZTM).

However, taking the available structural knowledge into account, we can classify our data into three broad classes: uncoupled complexes ($E_{FRET} = 0$); loosely coupled expressome complexes TTC-LC[33] ($E_{FRET} \approx 0.1$); and coupled expressomes in state TTC-B[6,7] ($E_{FRET} \approx 0.3$). Targeted hybridization of DNA oligonucleotides to the intervening mRNA results in an increase in the inter-ribosome–RNA distance distribution for mRNA-46 and a complete loss of coupling for mRNA-85 and the longer mRNAs (Extended Data Fig. 4b–d) further validating our assignment. Using this classification, we plotted the fraction of coupled states (both TTC-B and TTC-LC) as a function of the intervening mRNA length and find that the fraction of coupled expressomes decreases from 65 ± 4% for mRNA-46 to 8 ± 1% for mRNA-457 (Fig. 3c and Extended Data Fig. 5b).

Next, we determined how long the expressomes remain coupled. For mRNA-46, all the molecules remain coupled until photobleaching, with a lower bound of 150 ± 5 s for the median coupled lifetime (Extended Data Fig. 5c). This is in agreement with the formation of stable complexes observed in the cryo-EM structures[6,7] and would enable a ribosome–RNAP complex to remain coupled during transcription of a complete *E. coli* gene. For the longer intervening mRNA lengths, the majority of the expressomes also remain coupled till photobleaching (Extended Data Fig. 5c,e–h). A small fraction of expressomes show uncoupling and recoupling (Extended Data Fig. 5a). with recoupling becoming slower with increasing intervening mRNA lengths, taking in average 10.1 ± 1.4 s and 24.3 ± 5 s to recouple for the mRNA-85 and mRNA-193 constructs, respectively (Extended Data Fig. 5d–h).

Overall, these experiments demonstrate that ribosome and RNAP can physically interact even if the ribosome does not closely trail behind the RNAP and that the expressomes become less compact and decrease in physical coupling efficiency with increasing intervening mRNA length.

## Nus factors affect expressome compaction

The transcription factor NusG is responsible for bridging the ribosome with the RNAP[5,34–37] and restrains RNAP motions[6]. By contrast, only a

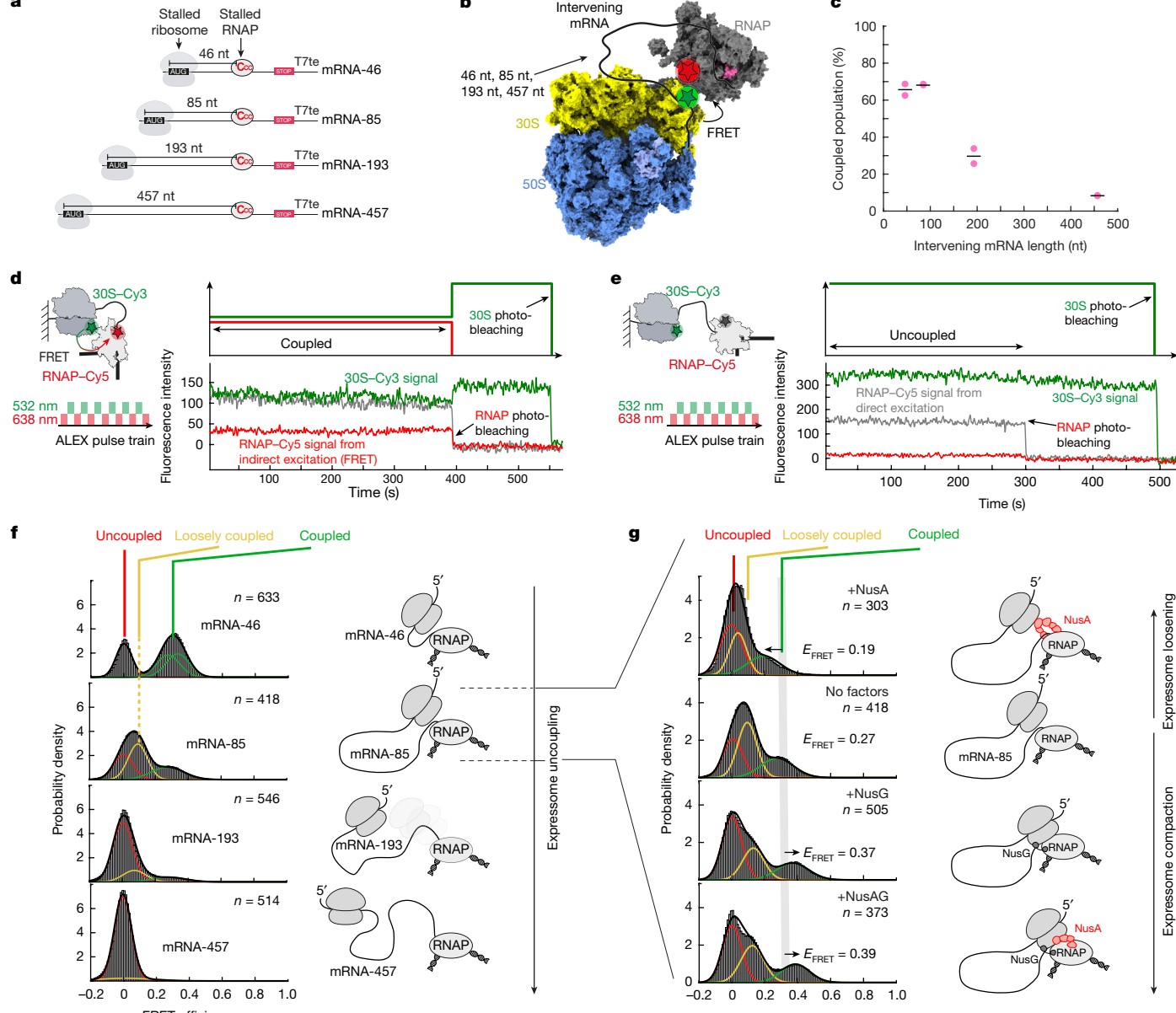

**Fig. 3 | Real-time tracking of ribosome–RNAP coupling in dependence of intervening mRNA length and Nus factor composition. a**, DNA template design. T7te, T7 terminator. **b**, Location of labelling sites within coupled expressome (PDB: 6XDQ). **c**, Total population of coupled states (loosely coupled and coupled) in dependence of intervening mRNA length. Mean of two replicates is shown as black line. Number of analysed molecules (*n*) and data used are the same as depicted in **f**. **d**,**e**, Schematic (left) and representative smoothed traces (right) for coupled (**d**) and uncoupled (**e**) expressomes.

**f**, FRET distribution histograms with varying mRNA length. Uncoupled ($E_{FRET} \approx 0$), loosely coupled ($E_{FRET} \approx 0.1$) and coupled states ($E_{FRET} \approx 0.3$) are indicated. Number of analysed molecules (*n*) is indicated. Data were combined from two replicates. **g**, FRET distribution histograms with different Nus factor compositions for mRNA-85. Number of analysed molecules (*n*) is indicated. Data were combined from two replicates. FRET distribution histogram shown for mRNA-85 without factors is the same as shown in **f**. NusAG, NusA plus NusG.

single NusA-only bound expressome structure was obtained at low resolution at a short intervening mRNA sequence of 39 nt, suggesting loosening of the expressome structure[7]. We repeated our coupling experiments for mRNA-85 in the presence of either one or both Nus factors. Our data show that in the presence of NusG, irrespective of NusA, the $E_{FRET}$ value of the coupled state increases in agreement with compaction of the expressome (Fig. 3g and Extended Data Fig. 4e). By contrast, addition of NusA alone resulted in a loosening of the expressome structure. Overall, these findings illustrate how the two transcription factors NusA and NusG modulate expressome compaction and are in agreement with NusG being the main driver of expressome stability.

## Nus factors drive coupling during transcription

Next, we investigated the effect of transcription elongation on the coupling efficiency. To this end, we immobilized expressomes with 46 nt intervening mRNA sequence and containing Cy5–RNAP, Cy3–30S and the 3′ end of the DNA template labelled with 2×Cy3.5. Under these conditions, 78% of the molecules showed a Cy3–Cy5 FRET signal at the start of the experiment, characteristic for the presence of coupled complexes (Fig. 4a and Extended Data Figs. 3a, 6c and 7a–c). After starting single-molecule imaging, we added NTPs to initiate transcription elongation, but kept the ribosome stalled at the RBS by not adding translation elongation factors. Under these conditions transcription

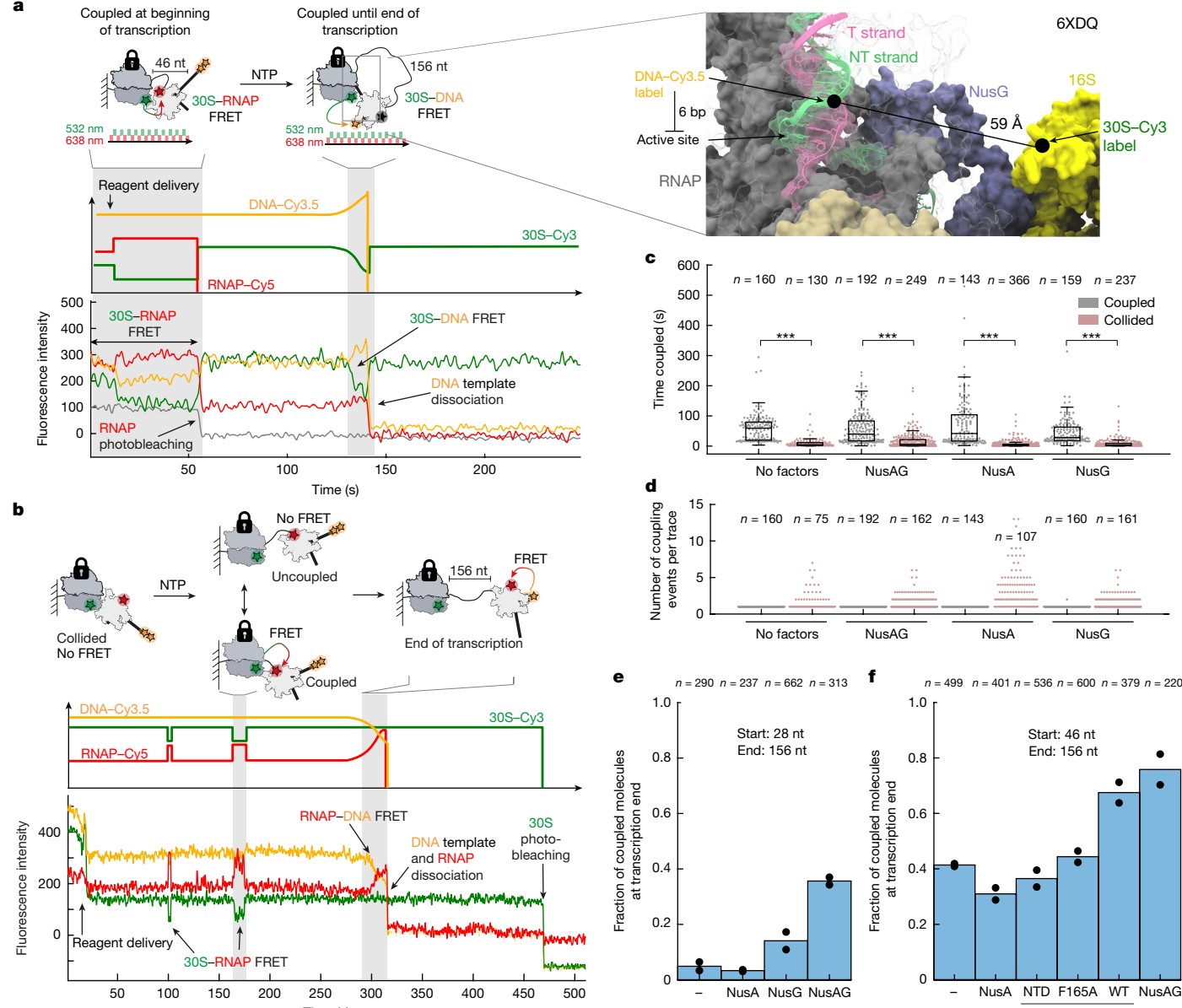

**Fig. 4 | Nus factors increase coupling during active transcription elongation and RNAP–ribosome collisions reduce subsequent coupling efficiency. a,b**, Experimental setup (top) and example traces for tracking coupling during active transcription elongation starting from coupled (**a**, bottom left) or collided state (**b**, bottom). Traces were smoothed for visualization. **a**, Right, expressome structure displaying the distance between the 30S–Cy3 label in h33a and the DNA–Cy3.5 label. **c**, Beeswarm plots representing all coupled dwell times for all molecules during transcription from coupled or collided states. Box plots are overlaid in black. Boxes show median (centre line), 25th percentile (bottom edge) and 75th percentile (top edge), and whiskers extend to 1.5 times the interquartile range. Number of dwell times (*n*) is indicated. Two-sided Wilcoxon–Mann–Whitney test: *$P < 0.05$, **$P < 0.01$, ***$P < 0.001$. Exact *P* values are shown in Source Data. **d**, Number of coupling and recoupling events per trace during transcription from coupled or collided states. Number of evaluated molecules (*n*) is shown. **e,f**, Mean fraction of molecules coupled at the end of transcription starting from collided (**e**) or coupled (**f**) states. Black dots represent values from two biological replicates. Total number of evaluated molecules (*n*) is shown. WT, wild type.

immediately resumes (Extended Data Figs. 8 and 9). In total, 37% of the traces showed a 30S–Cy3 to RNAP–Cy5 FRET signal until the end of transcription without any uncoupling or recoupling events (Fig. 4d and Extended Data Figs. 6c and 7a), demonstrating constant coupling between ribosome and RNAP during active transcription elongation. However, for the remaining 63% of the molecules, the RNAP–Cy5 signal photobleached before the end of transcription (Fig. 4a and Extended Data Fig. 7b,c), allowing us to only provide a lower estimate of approximately 60 s for the median coupled lifetime during active transcription elongation (Fig. 4c). Of note, for a fraction of the traces (42 ± 1%; Fig. 4f), a FRET signal between 30S–Cy3 and DNA–Cy3.5 appeared at the end of the 156-nucleotide mRNA transcription (Fig. 4a). This FRET, for which

both dyes are not affected by photobleaching under our experimental conditions, shows that the ribosome is positioned close to the end of the DNA template, a configuration that can only be achieved when the two machineries are coupled at the end of transcription (Fig. 4a and Extended Data Fig. 3d). Using this Cy3–Cy3.5 FRET signal as a proxy for coupling once the full-length mRNA has been transcribed (156 nucleotides intervening mRNA sequence), we next compared the effect of the different Nus factors on coupling efficiency. We find that in the presence of NusG, and largely independent of NusA, the fraction of molecules that are coupled at the end of transcription is approximately twofold higher, reaching around 80% in the presence of both factors (Fig. 4f). By contrast, NusG(NTD)[38] or NusG(F165A)[34], two mutants that can interact

only with the RNAP, show no increase in functional coupling (Fig. 4f) but retain their ability to increase the transcription rate (Extended Data Fig. 10b,c). That the coupling is mainly maintained by NusG is in agreement with a highly conserved NusG–ribosome interaction surface in contrast to the poor NusA–ribosome interface conservation[6–8,39]. Overall, our data show that RNAP and ribosome can remain physically coupled during active transcription elongation, with hundreds of nucleotides of intervening mRNA looping-out in between, and that the coupling efficiency is increased by transcription factor NusG, in agreement with previous findings that NusG is important for transcription–translation coupling[5,34–37].

## Collisions decrease subsequent coupling

Next, we explored whether ribosome–RNAP collisions may influence the subsequent coupling efficiency. To this end, we immobilized a collided expressome (RNAP–Cy5, 30S–Cy3) to the imaging surface. In contrast to the coupled expressome, the collided expressome shows no FRET between RNAP–Cy5 and 30S–h33a–Cy3 (>100 Å between the two dyes; Fig. 4b and Extended Data Fig. 3a). We triggered transcription elongation of the RNAP by delivering NTPs (Fig. 4b and Extended Data Fig. 6b). Transcription from the collided state immediately resumes after addition of all four NTPs (stalled TEC band completely chased in <10 s; see Extended Data Fig. 8), in agreement with previous studies[11]. As transcription proceeds and the RNAP transitions from being collided to being coupled with the ribosome, a Cy3–Cy5 FRET signal appears (Fig. 4b and Extended Data Fig. 7d). Remarkably, coupling following a collision is more than tenfold shorter-lived compared with coupling in the absence of a prior collision (Fig. 4c). The coupled lifetime reduces to only 7.8 ± 0.8 s (Extended Data Fig. 6a) with the RNAP and the ribosome dynamically coupling and uncoupling multiple times during subsequent transcription elongation (Fig. 4c,d and Extended Data Fig. 6b). The fraction of traces that show coupled transcription–translation complexes at the end of transcription reduces to 4.9 ± 1.5% (versus 41.4 ± 0.5% without prior ribosome–RNAP collision; compare Fig. 4e with Fig. 4f) and increases to 14 ± 3 or 36 ± 1% in the presence of NusG or both Nus factors, respectively (compared to 68 ± 4/76 ± 6% without prior ribosome–RNAP collision). Overall, this demonstrates that RNAP–ribosome collisions reduce subsequent coupling efficiency. We hypothesize that the more transient nature of coupling following a ribosome–RNAP collision may be attributed to a structural or compositional change of the ribosome–RNAP interface after collision–for example, the loss of the omega subunit from RNAP–as suggested by the missing electron density in the cryo-EM maps of the collided state[6,7].

## Ribosome rescues NusA-paused RNAPs

Having demonstrated that the ribosome and the RNAP can also physically couple with long intervening mRNA raised the question of whether this long-range physical coupling can be also functional. In other words, whether the ribosome can also activate a stalled RNAP by long-range coupling interactions.

To address this, we immobilized stalled TECs either in the absence of a ribosome or in the presence of a ribosome in a coupled state with 46 nucleotides of intervening mRNA. We started the single-molecule imaging reactions with the addition of NTPs and measured the time it takes to transcribe full-length mRNA molecules (Fig. 5a,b). In line with previous bulk experiments[40–42], our single-molecule data (Fig. 5c and Extended Data Fig. 10a) and bulk in vitro transcription reactions (Extended Data Fig. 9) show a significant effect of NusA and NusG on transcription pausing or transcription activation, respectively, demonstrating the activity of the Nus factors on transcription elongation. In the absence of Nus factors (median transcription time: 69.2 ± 1.9 s (−70S) versus 62.4 ± 4 s (+70S); 1.1-fold decrease), with NusG alone (45.8 ± 4.8 s (−70S) versus 50.5 ± 11.0 s (+70S); 0.9-fold), or with both

Nus factors (120.2 ± 5.9 s (−70S) versus 86.6 ± 1.6 s (+70S); 1.4-fold decrease), we see only marginal ribosome-induced acceleration of transcription speed. However, with NusA alone (205.5 ± 0.5 s (−70S) versus 100.6 ± 10.3 s (+70S); twofold decrease), transcription is significantly faster in the presence of the ribosome, indicating that a coupled ribosome can rescue a NusA-paused RNAP.

To verify the results of our single-molecule experiments, we performed bulk in vitro transcription assays. For this purpose, we purified the expressome (formed with mRNA-46) using streptavidin beads by immobilizing via biotin-labelled 50S subunits. This enabled us to enrich fully assembled expressomes in our transcription assays (Methods). Although there is only a slight increase in transcription speed for most conditions (without factors, NusG or NusA and NusG) in the presence of the ribosome (Extended Data Fig. 9a–e,g), once again there is a significantly larger ribosome-induced increase in transcription efficiency in the presence of NusA alone (Fig. 5e and Extended Data Fig. 9a–e,g). We find two prominent pause sites for mRNA-46 that are most probably induced by formation of pause hairpins[43] (Extended Data Fig. 9h). Notably, these transcription assays enable us to determine pause-escape lifetimes in the presence and absence of the ribosome and the intervening mRNA distance between ribosome and RNAP at which the ribosome-accelerated pause release occurs (Methods). We find that at the first pause site (at 134 nt, 71 nt intervening mRNA), the ribosome decreases the pause-escape lifetime 1.1-fold from 66 ± 3 s to 60 ± 14 s in the presence of NusA (Extended Data Fig. 9e). At the second pause site (at 169 nt, 106 nt intervening mRNA) we observe a stronger effect (ribosome decreases the pause-escape lifetime 2.2-fold from 182 ± 9 s to 84 ± 17 s). Repeating our experiments by adding an mRNA-assembled ribosome to our actively transcribing RNAPs in *trans*, we find no ribosome-induced activation (Extended Data Fig. 9f), supporting the requirement for a shared mRNA between the two machineries for functional activation.

Structurally, both NusA-mediated RNAP pausing[44] and NusA-associated coupling in the expressome have been described[7]. The superposition of the two structures shows that the NusA conformation differs substantially between the expressome and NusA-bound RNAP structures and that the NusA pause-promoting conformation cannot occur in the coupled expressome because NusA would clash with the 30S ribosomal subunit (Fig. 5d). Overall, our data are consistent with a model in which long-range coupling between the ribosome and a NusA-mediated paused RNAP prevents NusA from assuming a pause-promoting conformation, thereby activating transcription.

## Discussion

Transcription–translation coupling has served as a paradigm on how macromolecular machines can cooperate inside cells. Here, using in vitro reconstitution of the complete active transcription–translation machinery and using multi-colour single-molecule imaging, we have simultaneously tracked transcription and translation elongation, together with the physical and functional coupling between both machineries. Decades of work and recent cryo-EM and cryo-ET structures of the expressome suggest that transcription–translation coupling is general[1,5–9,12,36,45], but recent data also suggest that coupling is more stochastic in *E. coli*[46,47] or that the two processes are mostly decoupled[48]. Our work consolidates both models, as we show that physical and functional coupling can also occur efficiently when both machineries are far apart along the mRNA sequence, by looping of the mRNA (Fig. 5f). Our new model has several major implications for the field and beyond.

The functional relevance of the collided state is debated[7,36,37]. It was structurally characterized by several independent groups[6–10] but failure to detect this state in vivo in the absence of antibiotics[8] was interpreted as evidence that the collisions are transient[37]. However, it was also argued that these collisions are non-physiological[7]. Ribosome–RNAP collisions are a possible in vitro mechanism for the ribosome to rescue

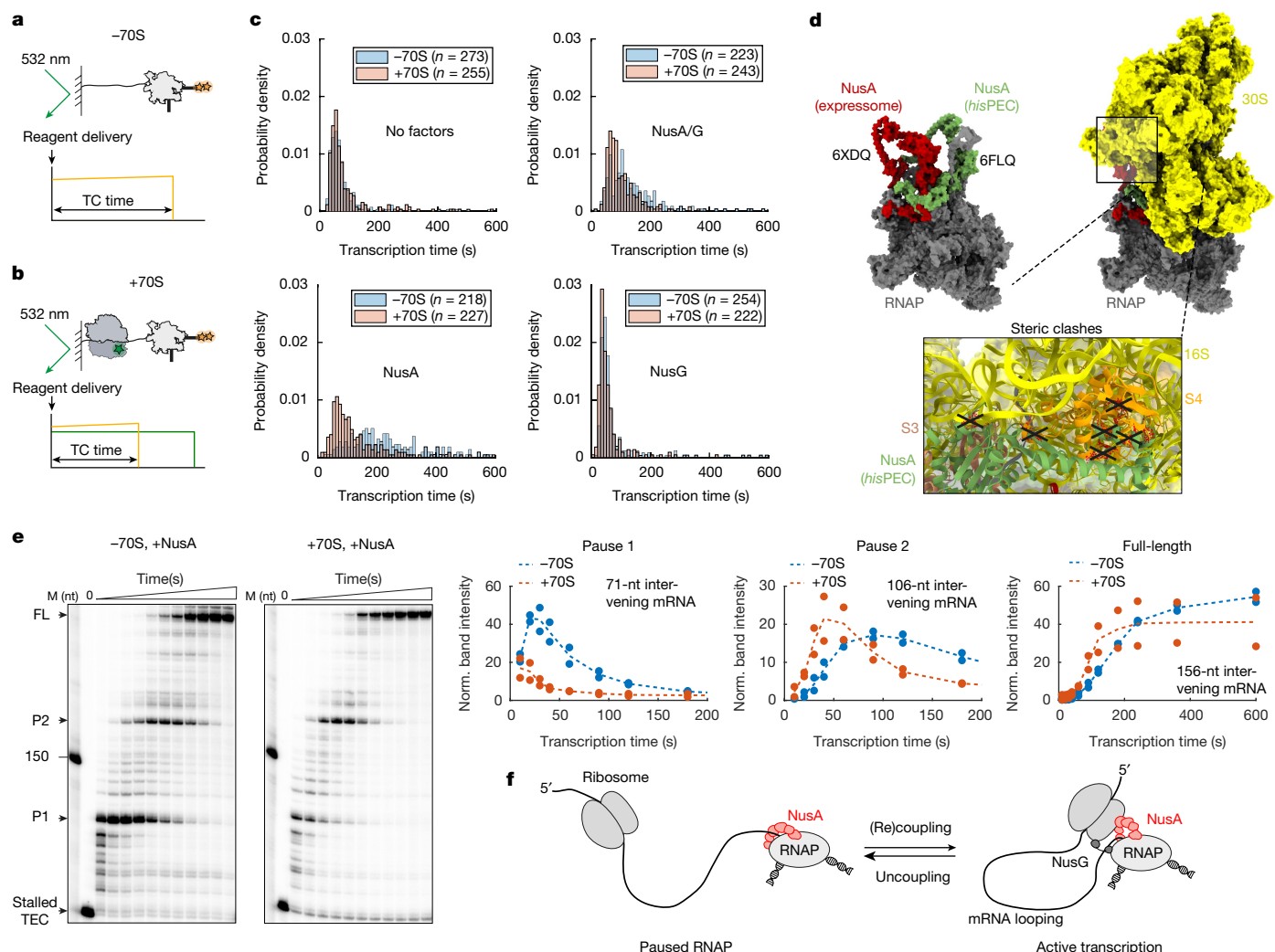

**Fig. 5 | Ribosome reactivates NusA-paused RNAP via mRNA looping. a,b**, Overview of single-molecule experimental setup. The start of the experiment was triggered by delivery of 50 µM NTPs and 1 µM Nus factors at various combinations to either immobilized stalled mRNA-46 TEC (−70S; **a**) or stalled expressome (+70S; **b**). TC, transcription. **c**, Overlay of probability density distribution for single-molecule transcription times without 70S and with 70S in the presence and absence of Nus factors. Number of evaluated molecules (*n*) is indicated. **d**, NusA conformations in the histidine (*his*) operon leader paused elongation complex (*his*PEC)-bound cryo-EM structure (PDB: 6FLQ) and in the expressome-bound cryo-EM structure (PDB: 6XDQ).

The structures were aligned on RNAP Cα backbone atoms using PyMol. NusA conformation in *his*-paused RNAP is incompatible with the expressome owing to steric clashes between NusA and 30S (S3, S4 and 16S rRNA). **e**, Left, single-round transcription assays with mRNA-46 in the presence of NusA and with or without 70S, analysed by denaturing gel electrophoresis. Stalled TEC, pause 1 (P1), pause 2 (P2) and full-length (FL) mRNA bands are shown (full gel is presented in Extended Data Fig. 9). Top right, band intensities (from two biological replicates) are plotted as a function of time. **f**, Model for ribosome-assisted RNAP activation via long-range interactions.

a backtracked RNAP[10–12,14], but functionally only at a separation of 28 to 29 nucleotides of intervening mRNA between RNAP active site and ribosome P-site[10,14]. Structural studies demonstrate that a substantial fraction of the expressomes adopt a collided conformation at a distance of 37 nucleotides or less[6,7] and our data show that the ribosome begins to slow down at an intervening mRNA distance at which the expressome preferentially adopts a collided conformation. We therefore postulate that functional collisions, which occur with 28 to 29 nucleotides of intervening mRNA, are probably infrequent events, but are relevant, for example, for rescue of arrested backtracked RNAPs or for selective destruction of non-functional transcription complexes stalled on DNA lesions[14]. Instead, more efficient ribosome-induced RNAP activation from NusA-dependent hairpin-stabilized pausing via long-range coupling can occur during the entire transcription process, with several hundred nucleotides of separation, and is therefore relevant for an average *E. coli* mRNA length of 1,000 nucleotides[49]. Long-range physical coupling may also occur between the RNAP and a non-leading

ribosome. Such a mechanism would provide a means for the RNAP to sense the ribosome loading occupancy on the mRNA rather than just the physical proximity of the leading ribosome, because a larger number of ribosomes on the nascent mRNA would provide more chances for long-range RNAP–ribosome encounters and thereby increase the probability for RNAP activation.

We show that the mRNA can form loops containing several hundred nucleotides and that coupling remains stable during active transcription, further supported by NusG and NusA. This requires rethinking our understanding of how the ribosome modulates Rho-dependent transcription termination[50,51]. In a model in which the mRNA can loop out by more than 100 nucleotides, Rho could also engage with the mRNA (Rho–mRNA footprint of 57–85 nucleotides[52]) at the same time that the RNAP and the ribosome are physically coupled. Whether Rho could push the ribosome away in order to reach the RNAP for terminating transcription and how this is coordinated with NusG, which can bind either Rho or the ribosome with its C-terminal domain[50], remains to be investigated.

Co-transcriptional RNA looping is likely to be a more general feature for increasing the efficiency of transcription-coupled processes[53]. For example, in bacterial rRNA transcription (mediated by interactions between the ribosomal RNA transcription antitermination complex and the RNAP[54]) and eukaryotic co-transcriptional splicing (mediated by U1 small nuclear ribonucleoprotein (snRNP)–RNA polymerase II interactions[55]) transcription-mediated RNA looping could provide means for increased RNA processing efficiency (RNase III processing and splicing, respectively).

Overall, our work underlines the importance of studying biological processes not in isolation but in the context of their functionally coupled system and highlights the power of multi-colour single-molecule experiments for tracking the dynamics of complex interconnected multi-step processes at high spatiotemporal resolution.

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

## Methods

### Strains and plasmids

Ribosome mutant SQ380 strains (B68 strain, 30S helical extension in h33a; C68 strain, 30S helical extension in h44; and ZS22 strain, 50S helical extension in h101) and all plasmids for initiation and elongation factors were a gift from the Puglisi laboratory[16,17]. The plasmid for overexpressing initiator tRNA^fmet (pBStRNAfmetY2) was a gift from E. Schmitt[56]. The plasmids for *E. coli* RNAP (pVS10) and σ70 (pIA586) were purchased from Addgene (#104398 and #104399, respectively)[57]. The ybbR peptide tag (DSLEFIASKLA)[58,59] was cloned into the pVS10 plasmid by introducing the ybbR DNA sequence with primers, followed by 5′-end phosphorylation with T4 PNK (New England Biolabs, NEB) and simultaneous blunt ligation of the plasmid using T4 DNA ligase (NEB). For the N-terminal ybbR mutant, the tag was inserted between E15 and E16 in the β′ subunit; for the C-terminal β′ mutant, it was inserted between E1377 and G1402, thereby deleting the region A1378–L1401. Gene sequences of ribosomal protein S1, NusA, NusG, NusG(NTD) (M1–P124), NusG(F165A), methionyl-tRNA^fmet formyltransferase (MTF) and methionine tRNA synthetase (MetRS, truncated at K548 to obtain the monomeric form of the enzyme[60]) were cloned into a pESUMO vector backbone using Gibson assembly. Gene sequences were obtained from ASKA collection plasmids[61].

### Sample preparation

*E. coli* ribosomal subunits, initiation factor IF2 and elongation factors (EF-Tu, EF-G and EF-Ts) were prepared and purified as previously described[16,17]. Wild-type RNAP and mutant versions were purified as described[57]. Initiator tRNA^fmet was prepared and purified following the protocol as described[62]. S1, NusA, NusG and NusG mutants were overexpressed as N-terminal fusions with His$_6$–SUMO. They were purified by lysing the cells in IMAC buffer (50 mM Tris-HCl pH 8.0, 300 mM NaCl, 10–20 mM imidazole, 10 mM 2-mercaptoethanol) using a microfluidizer, loading the clarified lysate on 5 ml HisTrap HP columns (Cytiva) and eluting with increasing imidazole concentrations over 20 column volumes. Protein fractions were pooled and the fusion tag was cleaved overnight with His-tagged Ulp1 protease, while dialysing against IMAC buffer without imidazole. The cleaved tag was removed by a second HisTrap purification. Protein fractions were pooled, concentrated using 15 ml Amicon Ultracel 10 or 30 K concentrators and further purified via size exclusion chromatography using either HiLoad S75 (NusG/NusG(NTD)/NusG(F165A)) or HiLoad S200 (NusA, S1) columns (Cytiva) equilibrated with storage buffer (S1 in 50 mM HEPES-KOH pH 7.6, 100 mM KCl, 1 mM DTT and NusA, NusG or NusG mutants in 20 mM Tris-HCl pH 7.6, 100 mM NaCl, 0.5 mM EDTA/DTT). Protein fractions were pooled and concentrated using 15 ml Amicon Ultracel 10 or 30 K concentrators to ~100 μM. Aliquots were flash-frozen with liquid nitrogen and stored at −80 °C. In the case of MTF and MetRS, the preparation was done as described above with following changes (and adapted from refs. 62,63): (1) MTF IMAC buffer contained 10 mM K$_2$HPO$_4$/KH$_2$PO$_4$ pH 7.3, 100 mM KCl, 20 mM imidazole and 10 mM 2-mercaptoethanol; MetRS buffer contained 10 mM K$_2$HPO$_4$/KH$_2$PO$_4$ pH 6.7, 50 mM KCl, 20 mM imidazole, 100 μM ZnCl$_2$ and 10 mM 2-mercaptoethanol. (2) After a second HisTrap purification, MetRS containing fractions were pooled and dialysed against storage buffer (10 mM K$_2$HPO$_4$/KH$_2$PO$_4$ pH 6.7, 10 mM 2-mercaptoethanol and 50% (v/v) glycerol). Aliquots were flash-frozen with liquid nitrogen and stored at −80 °C. (3) After a second HisTrap purification, MTF was further purified via a 5 ml HiTrap Q FF column (Cytiva) and eluted using a linear gradient of 20–100% into Q-sepharose buffer (10 mM K$_2$HPO$_4$/KH$_2$PO$_4$ pH 7.3, 500 mM KCl and 10 mM 2-mercaptoethanol). Protein fractions were pooled and dialysed against storage buffer (10 mM K$_2$HPO$_4$/KH$_2$PO$_4$ pH 6.7, 100 mM KCl, 10 mM 2-mercaptoethanol and 50% (v/v) glycerol). Aliquots were flash-frozen with liquid nitrogen and stored at −80 °C. An SDS–PAGE gel of all protein factors used in

this manuscript is included in Extended Data Fig. 1j. All uncropped gels are presented in Supplementary Fig. 1.

### Charging tRNA^fmet and elongator tRNAs

Typically, initiator tRNA^fmet (20 μM) was simultaneously charged and formylated in 800 μl reactions using 100 μM methionine, 300 μM 10-formyltetrahydrofolate, 200 nM MetRS and 500 nM MTF in charging buffer (50 mM Tris-HCl pH 7.5, 150 mM KCl, 7 mM MgCl$_2$, 0.1 mM EDTA, 2.5 mM ATP and 1 mM DTT), incubating for 5 min at 37 °C (refs. 62,64). The fmet–tRNA^fmet was immediately purified by addition of 0.1 volumes of sodium acetate (pH 5.2), extraction with aqueous phenol (pH ~ 4) and precipitation with 3 volumes of ethanol. The tRNA pellet was solubilized in ice-cold tRNA storage buffer (10 mM sodium acetate pH 5.2, 0.5 mM MgCl$_2$) and further purified with a Nap-5 column (Cytiva) equilibrated with the same buffer. The eluate was aliquoted, flash-frozen with liquid nitrogen and stored at −80 °C. Elongator tRNAs were purchased (tRNA MRE600, Roche) and charged typically at 500 μM concentration in the presence of 0.2 mM amino acid mix (each), 10 mM phosphoenolpyruvate (PEP), 20% (v/v) S150 extract (prepared following ref. 65), 0.05 mg ml$^{-1}$ pyruvate kinase (Roche) and 0.2 U μl$^{-1}$ thermostable inorganic pyrophosphatase (NEB) in total tRNA charging buffer (50 mM Tris-HCl pH 7.5, 50 mM KCl, 10 mM MgCl$_2$, 2 mM ATP and 3 mM 2-mercaptoethanol)[29]. Typically, 200 μl reactions were incubated at 37 °C for 15 min and then immediately purified as described above. To remove NTP contaminations introduced by the S150 extract, the aa–tRNAs were further purified over a S200 increase column (Cytiva), pre-equilibrated in tRNA storage buffer. The eluted fractions were combined, concentrated with 2 ml Amicon Ultracel 3K concentrators, aliquoted and flash-frozen with liquid nitrogen and stored at −80 °C. Charging efficiency (typically >90%) was verified with acidic urea polyacrylamide gel electrophoresis as described[64].

### Dye labelling of expressome components

Hairpin loop extensions of mutant ribosomal subunits were labelled with prNQ087–Cy3 or prNQ159–Cy3B (30S) and prNQ088–Cy5 (50S) DNA oligonucleotides, complementary to mutant helical extensions[16,17,19] (see Supplementary Table 3 for all DNA and RNA oligonucleotide sequences). Just prior to the experiments, each subunit was labelled separately at 2 μM concentration using 1.2 equivalents of the respective DNA oligonucleotide by incubation at 37 °C for 10 min and then at 30 °C for 20 min in a Tris-based polymix buffer (50 mM Tris-acetate pH 7.5, 100 mM potassium chloride, 5 mM ammonium acetate, 0.5 mM calcium acetate, 5 mM magnesium acetate, 0.5 mM EDTA, 5 mM putrescine-HCl and 1 mM spermidine). RNAP–ybbR mutants (core enzyme) were labelled by mixing 7 μM RNAP, 14 μM SFP synthetase and 28 μM CoA-Cy5 dye in a buffer containing 50 mM HEPES-KOH pH 7.5, 50 mM NaCl, 10 mM MgCl$_2$, 2 mM DTT and 10% (v/v) glycerol[66]. Typically, 100 μl reactions were incubated at 25 °C or 37 °C for 2 h and analysed on denaturing protein gels. The holoenzyme was formed by incubating 1.11 μM Cy5-labelled RNAP with 3 equivalents σ70 for 30 min on ice in RNAP storage buffer (20 mM Tris-HCl pH 7.5, 100 mM NaCl, 0.1 mM EDTA, 1 mM DTT, 50% (v/v) glycerol). Aliquots were stored at −20 °C. SFP and free dye were removed on imaging surface prior to experiments.

DNA templates were purchased (TwistBioscience) and amplified via PCR using p0030 forward and p0075 reverse abasic primers, generating single-stranded 5′ overhangs for both DNA strands[15]. The fragments were purified on 2% agarose gels, extracted using a QIAGEN gel extraction kit and buffer exchanged in e55 buffer (10 mM Tris-HCl pH 7.5 and 20 mM KCl) using 0.5 ml Amicon Ultracel 30 K concentrators. The 5′ overhang of the template DNA was hybridized by mixing with 1.2 equivalents of p0088–2×Cy3.5 DNA oligonucleotide at 68 °C for 5 min, followed by slow cool down (~1 h) to room temperature.

## Single-round in vitro transcription assays

Stalled TECs were assembled in transcription buffer (50 mM Tris-HCl pH 8, 20 mM NaCl, 14 mM MgCl$_2$, 0.04 mM EDTA, 40 µg ml$^{-1}$ non-acylated BSA, 0.01% (v/v) Triton X-100 and 2 mM DTT) as described previously[15,67]. In brief, 50 nM DNA template was incubated (20 min, 37 °C) with four equivalents of RNAP in the presence of 100 µM ACU trinucleotide, 5 µM GTP, 5 µM ATP (+150–300 nM $^{32}$P α-ATP, Hartmann Analytic), halting the polymerase at U24, to prevent loading of multiple RNAPs on the same DNA template. Next, re-initiation of transcription was blocked by addition of 10 µg ml$^{-1}$ rifampicin. The RNAP was walked to the desired stalling site by addition of 10 µM UTP and incubating at 37 °C for 20 min. The 30S ribosomal subunit was loaded for 10 min at 37 °C by incubating 25 nM stalled TEC with 250 nM 30S (B68 mutant, pre-incubated with stoichiometric amounts of S1 protein for 5 min at 37 °C) in the presence of 2 µM IF2, 1 µM fmet–tRNA$^{fmet}$ and 4 mM GTP in polymix buffer with 15 mM magnesium acetate. To enrich for fully assembled ribosome–RNAP complexes, the expressome was purified via immobilizing the 50S subunit (ZS22) onto streptavidin magnetic beads (NEB). For this, the 50S subunit was pre-annealed on h101 with prNQ302-prNQ303-prNQ304-p0109–biotin–DNA oligonucleotide. The prNQ303 and prNQ304 DNA oligonucleotides contain a BamHI cleavage site to elute the purified ribosome–RNAP complex from streptavidin beads. 50S loading onto the stalled TEC/30S PIC occurred simultaneously while immobilizing the ribosome–RNAP complex on streptavidin magnetic beads (pre-equilibrated with polymix buffer with 15 mM magnesium acetate) for 10 min at room temperature. Typically, 50 µl beads were loaded with a total volume of 150 µl ribosome–RNAP complex. The immobilized complex was washed once with polymix buffer with 15 mM magnesium acetate and then eluted with 100 µl polymix buffer with 15 mM magnesium acetate, while cleaving with BamHI for 20 min at 37 °C. The eluate was chased with 50 µM NTPs in the presence or absence of Nus factors (at 1 µM each, when present), 4 mM GTP, polymix buffer with 15 mM magnesium acetate and 100 mM potassium glutamate. Time points were taken before NTP addition ($t = 0$ s) and at 10, 20, 30, 40, 60, 90, 120, 180, 240, 360 and 600 s for each condition, mixing 2.5 µl sample with 5 µl stop buffer (7 M urea, 2× TBE, 50 mM EDTA, 0.025% (w/v) bromphenol blue and xylene blue) and incubated at 95 °C for 2 min. Single-round transcription assays with 70S in *trans* were performed as described above with minor changes. Ribosomal subunits were loaded at 1 µM concentration on 8 µM 6(FK) mRNA[17] to ensure that 70S PIC formation was complete. Stalled TEC was added, and the reaction was chased with 50 µM NTPs (each) in the presence of 1 µM NusA, having final concentrations of 350 nM 70S PIC and 6.25 nM stalled TEC. As size reference, the ssRNA ladder (NEB, sizes: 50, 80, 150, 300, 500 and 1,000 nt) was 5′ end labelled with $^{32}$P γ-ATP. The reactions were analysed on 6% denaturing PAGE (7 M urea in 1× TBE), running in 1× TBE and 50 W for 2–3 h. Gels were dried, exposed overnight on a phosphor screen (Cytiva, BAS IP MS 2040 E) and imaged using a Typhoon FLA 9500. Band intensities ($P$) were integrated using ImageLab 6.1 software (Bio-Rad) and divided by the total RNA per lane ($T$) to compensate for pipetting errors, as described[68]. Normalized band intensities ($P/T$) were plotted as function of time. Pause-escape lifetimes were fitted as described[68] by plotting ln(P/T) against time and fitting the pause-escape data range (indicated in plots) with a linear equation ($y = mx + b$, with $m$ being the pause-escape rate). All uncropped gels are presented in Supplementary Fig. 1.

## Single-molecule transcription–translation coupling assays

Stalled TECs were formed as described for single-round transcription assays with following changes: (1) Typically, labelled DNA was used (pre-annealed with p0088–2×Cy3.5). (2) Stalled transcription was performed in the presence of 10 µM ATP and 10 µM GTP. (3) Re-initiation of transcription was blocked with 1 mg ml$^{-1}$ heparin (final concentration). (4) For immobilization, the 5′ end of the nascent mRNA was labelled with a biotin–DNA oligonucleotide (prNQ127-p0109–biotin). (5) When necessary, Cy5-labelled RNAP was used (N-terminal ybbR tag or C-terminal ybbR tag). Typically, ribosomes (250 nM 30S, 500 nM 50S) were loaded for 5 min at 37 °C by incubating 4 nM stalled TEC in the presence of 2 µM IF2, 1 µM fmet–tRNA$^{fmet}$ and 4 mM GTP in polymix buffer with 15 mM magnesium acetate. The loading reaction was diluted to 100–800 pM stalled TEC concentration (DNA template-based) with an IF2-containing (2 µM) polymix buffer at 15 mM magnesium acetate and immobilized on biotin–PEG functionalized slides coated with NeutrAvidin for 10 min at room temperature[69]. Unbound components were washed away with an IF2-containing (2 µM) imaging buffer (polymix buffer with 15 mM magnesium acetate, 100 mM potassium glutamate) and imaging was immediately started. The imaging buffer was supplemented with an oxygen scavenger system (OSC), containing 2.5 mM protocatechuic acid and 190 nM protocatechuate dioxygenase and a cocktail of triplet state quenchers (1 mM 4-citrobenzyl alcohol, 1 mM cyclooctatetraene and 1 mM trolox) to minimize fluorescence instability. For equilibrium experiments, the imaging buffer in addition contained 4 mM GTP and 1 µM Nus factors (each, exact composition indicated in figures). For real-time experiments, the reaction was initiated while imaging (after 10–30 s), with delivery of 50 nM 50S–Cy5 (where applicable), 2 µM IF2, 10–500 nM EF-G (where applicable, in specified concentrations, typically 50 nM), 100–1500 nM ternary complex (where applicable, in specified concentrations, typically 150 nM), 10–1,000 µM NTPs (each, where applicable, in specified concentrations) and 1 µM Nus factors (each, where applicable) in the same imaging buffer containing additional 4 mM GTP. The ternary complex was prepared as described previously[70,71]. For 2-colour translation experiments, a second delivery mix was injected containing 10% of the initial OSC (in imaging buffer) to actively induce photobleaching after 20 min of movie acquisition.

**Ribosomal subunit fluctuations.** Spontaneous intersubunit rotations can complicate analysis and assignment of real translation transitions[22]. Therefore, we chose experimental conditions in which we can minimize the interference from spontaneous intersubunit rotations. We are using polymix buffer, which was shown by the most recent study by Ermolenko and colleagues[22] to reduce the fraction of ribosomes showing spontaneous fluctuations by 20-fold with only 1% of the h44–Cy3/H101–Cy5 ribosomes showing spontaneous fluctuations[22]. Furthermore, we chose slower translation conditions in which the timescale of the spontaneous intersubunit rotations is faster (0.3–2 s$^{-1}$; see ref. 22) and not even all spontaneous fluctuations may be detected at our frametime of 200 ms. The difference in timescales between real translations and intersubunit fluctuations is especially pronounced during ribosome slowdown for which the timescale of translation and spontaneous intersubunit rotations can differ by two orders of magnitude.

## Single-molecule instrumentation and analysis

We performed all single-molecule experiments at 21 °C using a custom-built (by Cairn Research: https://www.cairn-research.co.uk/), objective-based (CFI SR HP Apochromat TIRF 100×C Oil) total internal reflection fluorescence (TIRF) microscope, equipped with an iLAS system (Cairn Research) and Prime95B sCMOS cameras (Teledyne Photometrics). For standard TIRF experiments (2 or 3 colours), we used a diode-based (OBIS) 532 nm laser at 0.6 kW cm$^{-2}$ intensity (on the basis of output power). The fluorescence intensities of Cy3, Cy3.5 and Cy5 dyes were recorded at exposure times of 200 or 300 ms. For alternative laser excitation[32] (ALEX) experiments, we operated the 532 nm laser at 0.73 kW cm$^{-2}$ intensity and in every alternate frame, illuminated the samples with a diode-based 638 nm laser (Omicron LuxX) at 0.12 kW cm$^{-2}$ intensity (200 ms exposure time for each laser, resulting in a frame rate of approximately 2 frames per second). Typically, 10 min movies were recorded. For determination of FRET efficiencies, longer movies (20 min) were acquired to ensure photobleaching of both dyes.

Images were acquired using the MetaMorph software package (Molecular Devices) and single-molecule traces were extracted using the SPARTAN software package (v.3.7.0)[72]. Subsequent analysis was done using the tMAVEN[73] software (when applicable) and with scripts[19,74] written in MATLAB R2021a and previous versions (MathWorks). Data evaluation in brief: three-colour data was assigned by thresholding, while two-colour data was assigned by HMM, with both approaches detailed in the following. Data for Fig. 2 were both recorded as three-colour data (as presented in Fig. 2b,c) and also repeated using two-colour data (data presented in Fig. 2d–f are combined data from two-colour and three-colour replicates; see Source Data).

(1) For two-colour translation experiments (30S–Cy3 or 30S–Cy3B and 50S–Cy5; Fig. 2), molecules were selected in SPARTAN that had a single photobleaching step for donor and acceptor dye. Those traces were baseline corrected, corrected for donor emission bleedthrough and the apparent sensitivity of each fluorophore. Subsequently, traces were exported to tMAVEN. The evaluation windows were selected from initial 50S subunit joining until: (a) photobleaching of one of the dyes occurred, (b) one dye entered a dark state; or (c) distorted fluorescence intensities that made further assignment difficult. Transitions were detected and assigned to two states using HMM (composite → vbHMM + Kmeans). For downstream evaluation (Extended Data Fig. 1c–f), those assignments were exported to MATLAB. First, HMM assignments were visually inspected and manually corrected for anti-correlated intensity changes of both dyes (real FRET transitions; see top trace in Extended Data Fig. 1d). Occasionally, intermediate FRET states were encountered that were corrected to high-FRET states using thresholding. For example, for Cy3B, this manual correction reduced the final average transitions per trace from 14 to 12 for elongating conditions and from 5 to 4 for colliding conditions. Second, traces were corrected for spontaneous subunit fluctuations of the rotated state, encountered especially under colliding conditions (see Extended Data Fig. 1e). To distinguish rotated state fluctuations (1–2 s timescale[22]) from real translations (non-rotated state median (3 replicates) = 12.8 ± 4.5 s, when using 150 nM aa–tRNA) we used dwell times for non-rotated states in elongating conditions as a threshold to cutoff non-translations (see Extended Data Fig. 1c–e). For this, we determined the 5% tile of all non-rotated state dwells in elongating conditions. Whenever 2 consecutive non-rotated dwells were encountered that were both shorter than the 5%-tile threshold (2.18/0.936 s; probability for 2 consecutive non-rotated dwells with <5%-tile duration to occur under normal translation conditions is 0.25%), translation count was stopped by putting those 2 dwells and all subsequent non-rotated state HMM intensities to rotated state HMM intensities (see Extended Data Fig. 1e, top trace). After that, also very short non-rotated dwells (<1%-tile; 0.936/0.5834 s) were removed. The last translation events in each trace were photobleaching-limited. We kept those in for data evaluation, as they inform on ribosome stalling after collision with the RNAP or after encountering the stop codon.

(2) For three-colour translation experiments (30S–Cy3, DNA–Cy3.5 and 50S–Cy5; Fig. 2), HMM assignment was not possible, due to spectral bleedthrough between channels (Fig. 2b,c; see also Extended Data Fig. 7c,d). Therefore, FRET transitions were assigned by use of trace-specific thresholds, selecting only for productive FRET states (high-FRET to low-FRET state or vice versa; Extended Data Figs. 1d,e and 7c,d and supplementary figure 11 in ref. 15). For the box plots shown in Fig. 2e, the following numbers of molecules/dwells were used (see also Extended Data Fig. 2c). Numbers for the first 6 amino acids (see Source Data for full list) are: elongating, non-rotated state: $n$ = 315, 315, 313, 312, 306 and 296; elongating, rotated state: $n$ = 315, 315, 312, 307, 298 and 283; colliding, non-rotated state: $n$ = 307, 306, 302, 275, 213 and 137; colliding, rotated state: $n$ = 307, 305, 293, 254, 187 and 117. Non-rotated and rotated state dwell times were used to calculate cumulative probability density functions of the observed data (ecdf, MATLAB) which were fitted to single-exponential functions in MATLAB (fit, using non-linear least squares methods). If initial double-exponential fitting of the data yielded a population that was represented less than 10%, the data was classified as following single-exponential kinetic behaviour[74].

(3) Two-colour equilibrium coupling experiments (Fig. 3) were acquired using ALEX. Expressome molecules were first selected for assembled expressomes (30S–Cy3 signal, and direct excitation signal of RNAP–Cy5) using SPARTAN. Next, traces were baseline corrected, corrected for the following: (a) donor emission bleedthrough, (b) direct excitation of the acceptor dye and (c) the apparent sensitivity of each fluorophore. Selected traces were exported to tMAVEN. The evaluation windows were selected to include only times, where the RNAP–Cy5 (direct) signal was alive for at least 100 s. To model the number of FRET states we used HMM (Global → vbConsensus + Model selection). The resulting number of states was used to assign FRET states using vbHMM + Kmeans. Selected evaluation windows and assignments were exported for downstream evaluation. FRET efficiencies were extracted with MATLAB using the evaluation windows and fitted to two or three gaussian distributions (depending on modelled states number) using maximum likelihood parameter estimation in MATLAB. For dwell time analysis, HMM assignments based on tMAVEN were visually inspected in MATLAB and corrected for true anti-correlated behaviour for donor and acceptor dyes. States with $E_{FRET}$ = 0 were assigned to uncoupled and all states with $E_{FRET}$ > 0 (loosely coupled and coupled) binned to a single coupled expressomes state. Recoupling rates were obtained by single-exponential fitting of cumulative dwell time distributions for dwells with $E_{FRET}$ = 0.

(4) For three-colour real-time coupling experiments (Fig. 4; ribosome stalled on RBS, RNAP chased with NTP), assembled expressome molecules were selected (30S–Cy3 signal, DNA–2×Cy3.5 signal and in the case of ALEX, direct excitation signal of RNAP–Cy5) and assigned by thresholding, due to spectral bleedthrough between channels (see Extended Data Fig. 7c,d). When applicable, dwell times for transcription or coupling were extracted. For determination of the fraction of coupled molecules until the end of transcription, we assigned the characteristic signal (30S–Cy3/DNA–2×Cy3.5 FRET) and calculated: fraction coupled = (number of coupled molecules with 30S–DNA FRET)/(total number of expressome molecules).

(5) For two- to three-colour transcription data evaluation, only traces with single expressome molecules (containing 30S–Cy3 (if present) and DNA–2×Cy3.5) were used. Transcription times were evaluated by assigning the time after reagent delivery until DNA template dissociation using trace-specific thresholds (Extended Data Fig. 7c,d). Average transcription times (Extended Data Fig. 1g) were obtained by fitting dwell time distributions to a convolution of a Gaussian (describes transcription) and exponential function (describes RNAP-stalling at the 3′ end before termination)[15].

For representation, the single-molecule traces were smoothed by zero-phase digital filtering by 3 points using the filtfilt function in matlab, but unsmoothed data was used for data evaluation.

Structures were visualized with Pymol (v 2.4) or ChimeraX 1.3. All figures were prepared with MATLAB R2021a, Excel 2016 and Adobe Illustrator.

## Statistical analysis

Reported error bars represent the s.d. of replicates as indicated in the figure captions. Errors in dwell time fits represent 95% confidence intervals obtained from fits to single-exponential functions as indicated. Statistical details of individual experiments, including number of replicates, analysed molecules or number of dwells used in the dwell time analyses are described in the manuscript text, figure legends, supplementary information or the Source Data. Every single-molecule

experiment was performed on a different sample with two to three biological replicates (see details on number of replicates in figure captions and more information in Source Data file). $P$ values were determined via two-sided Wilcoxon–Mann–Whitney test in MATLAB and are reported as $*P < 0.05$, $**P < 0.01$ and $***P < 0.001$. Values and test statistic can be found in Source Data. In all box plots, the centre line is the median, box edges indicate 25th and 75th percentiles and whiskers extend to 1.5× interquartile range.

## Reporting summary

Further information on research design is available in the Nature Portfolio Reporting Summary linked to this article.

## Data availability

Structures were downloaded from the Protein Data Bank (https://www.rcsb.org/) using the accession codes shown in the main figures (6XDQ, 6ZTM and 6FLQ) or listed in the Source Data for Extended Data Fig. 3c,d. The Article includes all relevant data generated or analysed during this study. All single-molecule traces, including transition assignments, are available at https://zenodo.org/records/13271669 (ref. 75). The raw data files are available upon request from O.D. Source data are provided with this paper.

## Code availability

The software used for single-molecule data processing and analysis is published[72,73] and freely available online. Additional scripts for downstream processing were made available publicly previously under an open-source license[74].

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

**Acknowledgements** The authors thank M. Reynolds, F. J. Evers, M. Lampe, A. Revyakin, members of the Puglisi laboratory, T. Hoffmann, A. Tsai, K. Gor, P. Shah and E. Geissen for assistance in single-molecule data acquisition and processing; A. Chaban for preparation of some NusA protein batches; D. Will from the EMBL chemical core facility for initial assistance in slide preparation; the EMBL protein expression and purification core facility for purification of recombinant *E. coli* RNAP; J. Mahamid and N. Typas for critical reading of the manuscript; and H. Grötsch, J. Weidenhausen, the Lemke laboratory and the entire Duss laboratory for assistance in the wet laboratory and helpful discussions. This project received funding from the FEBS Excellence Award, of the Deutsche Forschungsgemeinschaft (DFG project number 512397425) and the European Molecular Biology Laboratory to O.D.

**Author contributions** N.S.Q. and O.D. designed experiments. N.S.Q. cloned, expressed and purified all expressome components, performed all biochemical and single-molecule assays and analysed all experiments with input from O.D. N.S.Q. and O.D. wrote the manuscript.

**Funding** Open access funding provided by European Molecular Biology Laboratory (EMBL).

**Competing interests** The authors declare no competing interests.

**Additional information**
**Correspondence and requests for materials** should be addressed to Olivier Duss.

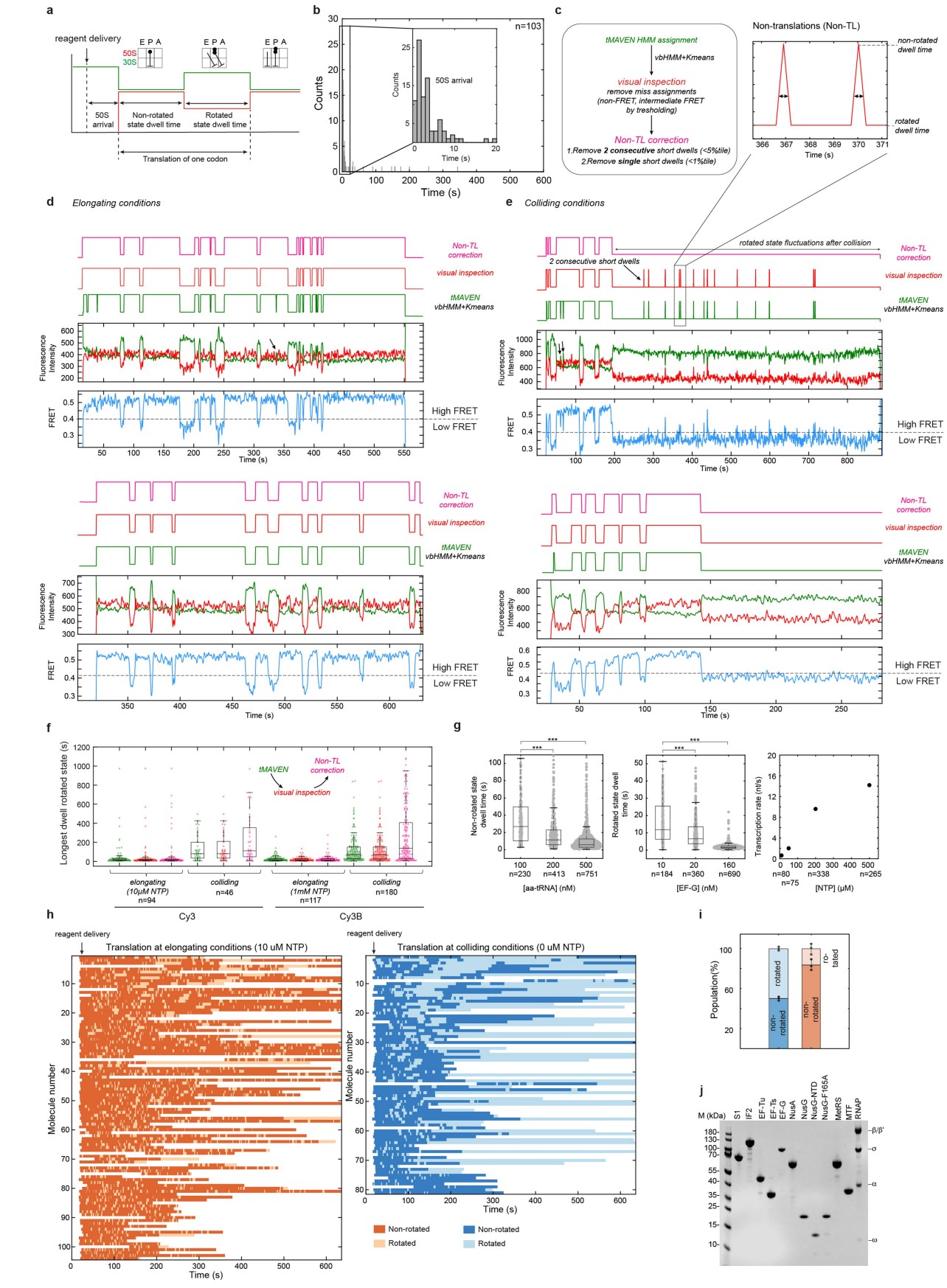

**Extended Data Fig. 1** | See next page for caption.

**Extended Data Fig. 1 | Real-time tracking of translation elongation.**
**a**, Scheme for data evaluation. **b**, 50S arrival time to the mRNA-46 stalled TEC loaded with 30S PIC. Data was acquired with 50 nM 50S-Cy5, 2 μM IF2a, 150 nM aa-tRNA, 450 nM EF-Tu, 300 nM EF-Ts, 50 nM EF-G at 21 °C. The time of 50S arrival to the expressome is plotted as histogram. Inset shows zoom of the first 20 s after reagent delivery. Reagent delivery time is at t = 0 s. Number of analyzed molecules (n) is indicated. **c**, Overview for HMM-assisted assignment of single-molecule traces. **d, e**, Representative smoothed traces for elongating (**d**) and colliding (**e**) conditions are shown. HMM assignment is shown on top. Original HMM assignment based on tMAVEN is shown in green, after visual inspection in red and after non-TL correction in pink (see methods). FRET efficiencies are plotted on bottom. Threshold used during visual inspection is indicated as an horizontal dashed line. **f**, Boxplots for longest dwell per trace is shown for elongating and colliding condition, exemplifying the assignment procedure on two replicates. Median (central mark), 25th percentile (bottom edge) and 75th percentile (top edge) are shown. Whiskers correspond to 1.5x interquartile range. **g**, Beeswarm and boxplots of dwell times for aa-tRNA-dependent non-rotated state (left panel) and EF-G-dependent rotated state (middle panel). Median (central mark), 25th percentile (bottom edge) and 75th percentile (top edge) are shown. Whiskers correspond to 1.5x interquartile range. Asterisks represent p-values determined via two-sided Wilcoxon-Mann-Whitney-test and are reported as $p < 0.05$*, $p < 0.01$**, $p < 0.001$***. Exact values are listed in Source Data. Note that at the given y-axis range not all outliers are visible (see full list in Source Data). Average transcription rate is plotted as function of NTP concentration (right panel). Numbers of evaluated dwell times (n) are indicated in the plots. **h**, Stack of single-molecule traces for elongating (left) and colliding (right) conditions. Each row represents a single transcription-translation complex. Non-rotated states are displayed in dark orange (elongating) or dark blue (colliding). The respective rotated states are shown in light orange or light blue. **i**, Mean fraction of molecules ending in rotated state or non-rotated state for colliding or elongating conditions. Last state before photobleaching was evaluated. Indicated errors are standard deviations from three biological replicates with points representing the value from individual replicates. **j**, SDS-PAGE gel (one replicate) containing all recombinantly expressed protein factors used in this study.

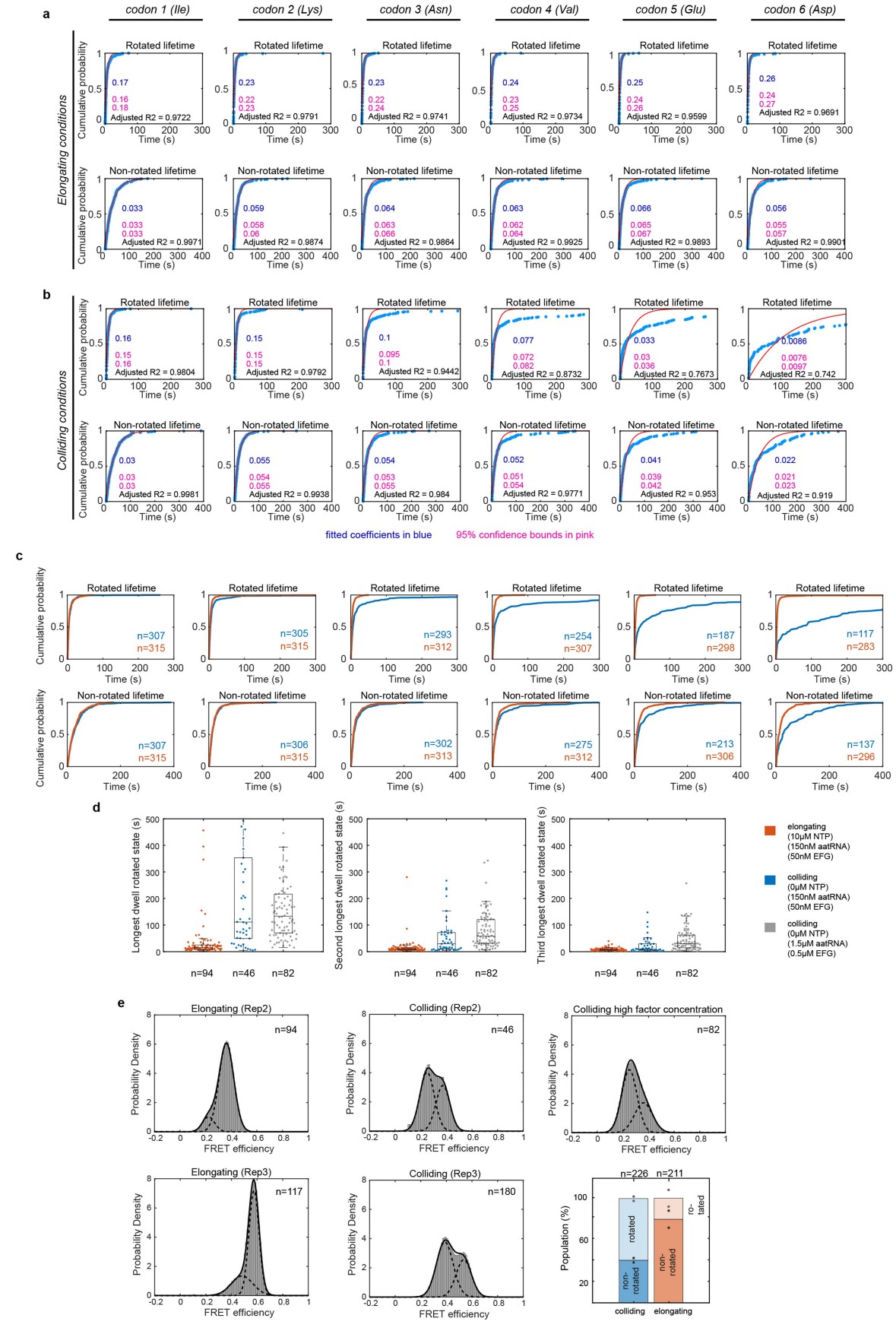

**Extended Data Fig. 2** | See next page for caption.

**Extended Data Fig. 2 | Real-time tracking of translation elongation.**
**a, b**, Under colliding conditions, rotated and non-rotated dwell times deviate from single-exponential behavior: Single exponential fitting of non-rotated and rotated state dwell-times for the first 6 amino acids during elongating (**a**) and colliding conditions (**b**), using y = 1-exp(-b*t), with given coefficient [b] representing the mean transition rate (in $s^{-1}$) for the respective state. 95% confidence bounds of the fits are indicated. The number of fitted dwells (n) are indicated as inset in panel **c**. **c**, Cumulative probability of colliding (blue) and elongating (orange) conditions from panel **a, b** are overlayed. The two right plots are also shown in main Fig. 2f. **d**, Boxplots of longest (left), second longest (middle) and third longest (right) dwell time per trace for all molecules are shown for elongating (orange) and colliding (blue) conditions. Number of evaluated dwells (n) is indicated. Median (central mark), 25th percentile (bottom edge) and 75th percentile (top edge) are shown. Whiskers correspond to 1.5x interquartile range. Colliding state dwells using 10x higher aatRNA/EF-G concentrations are shown in gray. Note: not all dwells are observable at given y-axis (see full list in Source Data). **e**, FRET distribution histograms are shown. Number of analyzed molecules is indicated. Bar graph in bottom right panel shows that the ribosome resides preferentially in the rotated state during the collision process. While approaching RNAP from 46 nt to 28 nt intervening mRNA length, the ribosome is 59 ± 3% of the time present in the rotated state versus 20 ± 12% during elongating conditions. Data points from 2 biological replicates are shown. Note: the FRET efficiency change between replicates is due to the use of different donor fluorophores that have different $R_0$ values (51 Å for Cy3 and 71 Å for Cy3B).

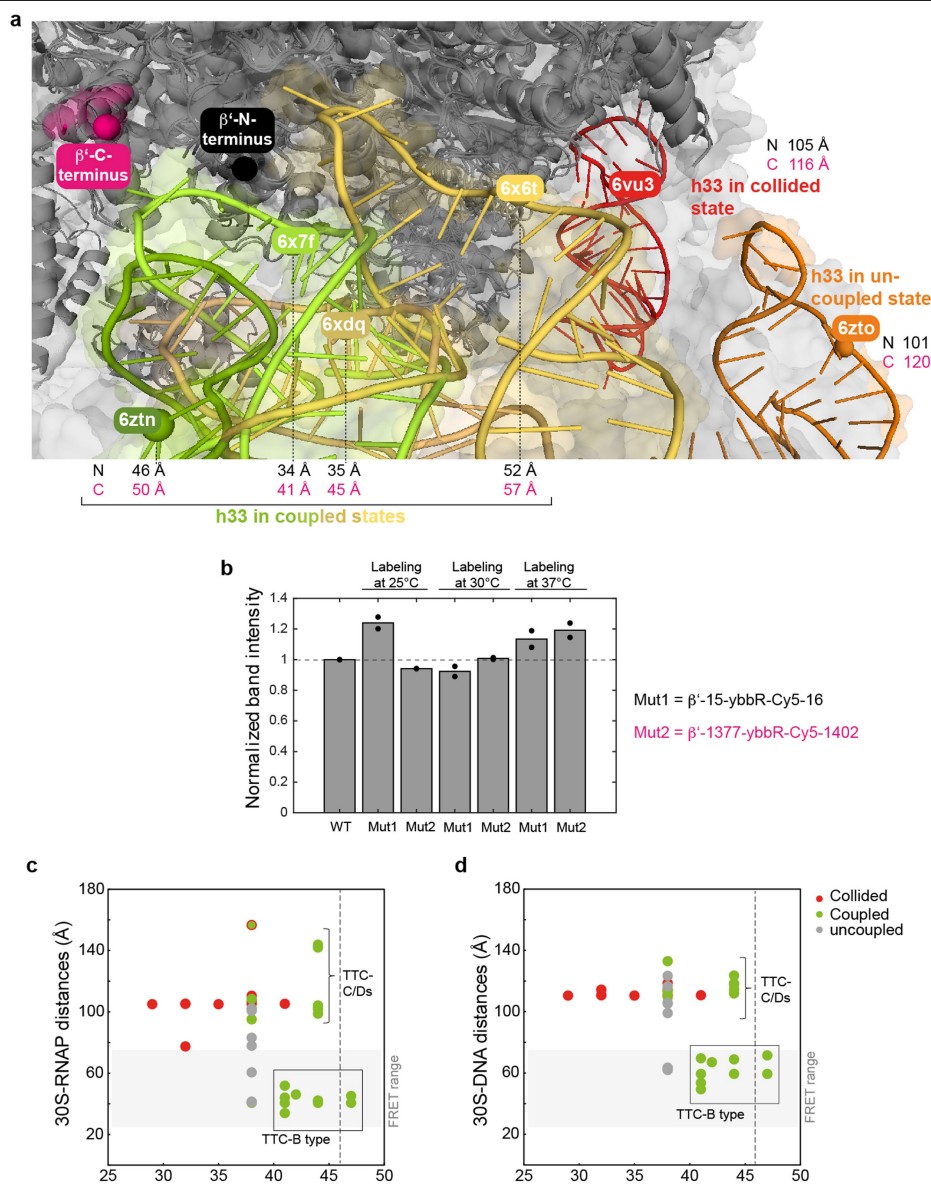

**Extended Data Fig. 3 | Ribosome – RNAP distances and labeling. a**, Overlay of expressome structures (6ztn, 6x7f, 6xdq, 6x6t, 6vu3, 6zto)[6,7] showing the variation of FRET distances between the ribosomal h33a (16S rRNA) and RNAP β' (Nter=black and Cter=pink) labeling sites. Structures were aligned on the RNAP. Helix 33 is color coded according to pdb ID and displayed as cartoon representation. Labeling sites on RNAP and 16S rRNA are indicated as spheres. In all displayed coupled states, the labeling sites are in FRET distance, whereas in the collided state and uncoupled state, they are too far to be detected by FRET (>100 Å). **b**, Introduction of the ybbR-peptide tag as well as the Cy5 label do not significantly affect RNAP activity: RNAP-Cy5 activity test using single-round transcription assays. Area of total RNA (mean of duplicates) was integrated and normalized to WT RNAP. Individual data points are shown. **c**, **d**, Overview of all pdb-deposited expressome structures[6,7] (n = 37), which serve as the structural basis for the ribosome-RNAP (**c**) or ribosome-DNA (**d**) FRET signal. Distances are plotted as a function of the intervening mRNA

illustrating that our FRET signal is specific for the coupled states. Collided state structures are shown in red, coupled state structures are shown in green and uncoupled state structures are shown in gray. For (**c**), distances were measured between E16 C-alpha (RNAP β') and U1025 C3' (16S rRNA) and for (**d**), the distances were measured between the same residue of RNAP and DNA 6 nt downstream from the active site to the non-template strand, where the Cy3.5 DNA label is located during transcription termination. Distances to alternative 30S/RNAP complexes relevant to translation initiation rather than elongation[76], were also evaluated (>130 Å) but cannot be plotted here, as they lack nucleic acids. Some uncoupled structures (6ztp, 6zto cluster 4 and 5) also fall in FRET range, however those structures were only obtained at a shorter 38 nt intervening mRNA length and were not observed at longer mRNA lengths used in our study to track ribosome/RNAP coupling. Vertical dashed line in plots at 46 nt represents the shortest construct used for the coupled state in this study.

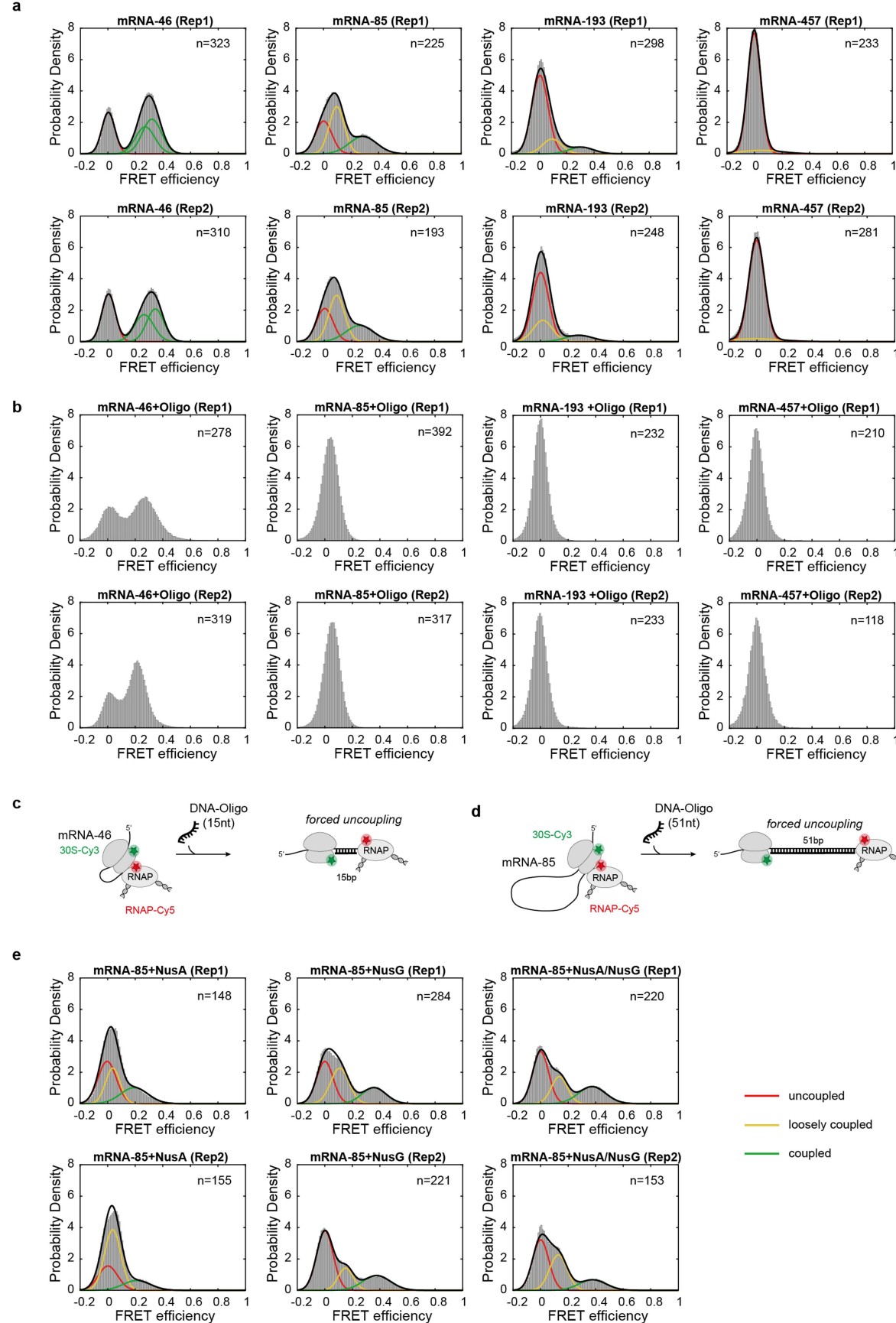

**Extended Data Fig. 4** | See next page for caption.

**Extended Data Fig. 4 | Ribosome-RNAP distance distributions. a**, FRET distribution histograms are displayed in dependence of mRNA length. Number of analyzed molecules are indicated within the plots. Uncoupled state ($E_{FRET}$ ~ 0) is shown in red, loosely coupled state ($E_{FRET}$ ~ 0.1) is shown in orange and coupled state ($E_{FRET}$ ~ 0.3) is shown in green. The combined datasets of both replicates are shown in Fig. 3f. **b-d**, DNA oligonucleotides hybridizing to the intervening mRNA can affect (mRNA-46) or completely disrupt coupling (mRNA-85, mRNA-193, mRNA-457). FRET distribution histograms (**b**) and corresponding schematics (**c, d**) for oligonucleotide induced expressome uncoupling. **e**, FRET distribution histograms are displayed in dependence of Nus factors for mRNA-85. Number of analyzed molecules are indicated within the plots. Uncoupled state ($E_{FRET}$ ~ 0) is shown in red, loosely coupled state ($E_{FRET}$ ~ 0.1) is shown in orange and coupled state ($E_{FRET}$ ~ 0.3) is shown in green. The combined datasets of both replicates are shown in Fig. 3g.

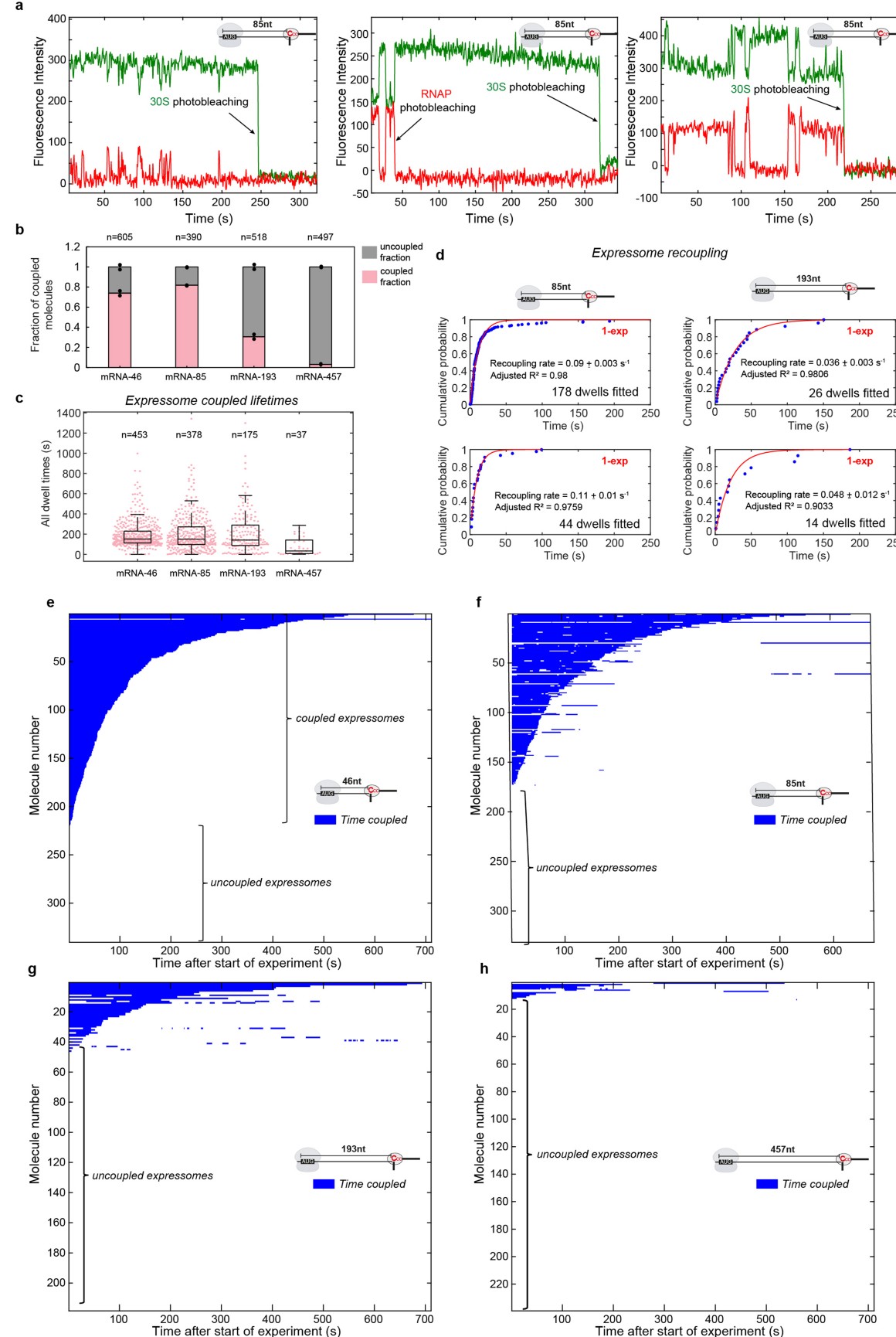

**Extended Data Fig. 5 |** See next page for caption.

**Extended Data Fig. 5 | Ribosome-RNAP coupling kinetics. a**, Representative, smoothed single-molecule traces (mRNA-85) displaying dynamically coupled expressome molecules acquired as equilibrium experiments (no reagent delivery). **b**, Mean fraction of coupled molecules is plotted for different mRNA lengths with values from individual biological replicates indicated (from two replicates). Total number of molecules (n) analyzed is indicated. **c**, Coupled dwell times for all coupling events for all molecules are displayed as beeswarm and boxplots. Median (central mark), 25th percentile (bottom edge) and 75th percentile (top edge) are shown. Whiskers correspond to 1.5x interquartile range. The signals for the expressome being in the coupled state before photobleaching (periods of 30S-Cy3/RNAP-Cy5 FRET) was evaluated for molecules for which the RNAP-Cy5 signal did not photobleach before 100 s. Number of analyzed dwells (n, from two replicates) is indicated. **d**, Expressome recoupling dynamics are shown for mRNA-85 and mRNA-193. The signals for the expressome being uncoupled (periods of no Cy3-Cy5 FRET) were evaluated and the dwells were fitted with a single exponential equation: $y = 1\text{-}\exp(\text{-}b*t)$. The number of fitted dwells (n) is indicated for both biological replicates (top and bottom plots). The errors represent the 95% confidence intervals of the fit. **e-h**, Stack of raw single-molecule traces for mRNA-46 (**e**), mRNA-85 (**f**), mRNA-193 (**g**), mRNA-457 (**h**) in absence of transcription and translation elongation. Each row represents a single transcription-translation complex. Coupled states (characterized by 30S-Cy3/RNAP-Cy5 FRET signal) are shown in blue. Traces are sorted by the total time for which coupling can be detected. White spaces in between coupling events represent uncoupled expressomes. The fraction of uncoupled expressomes increases with mRNA length. In case of the mRNA-46 expressome, we do not detect any uncoupling events and therefore, the apparent coupled state lifetime is limited by photobleaching. Traces without any coupled states during the entire experimental time are represented as empty traces (white) at the bottom of each plot.

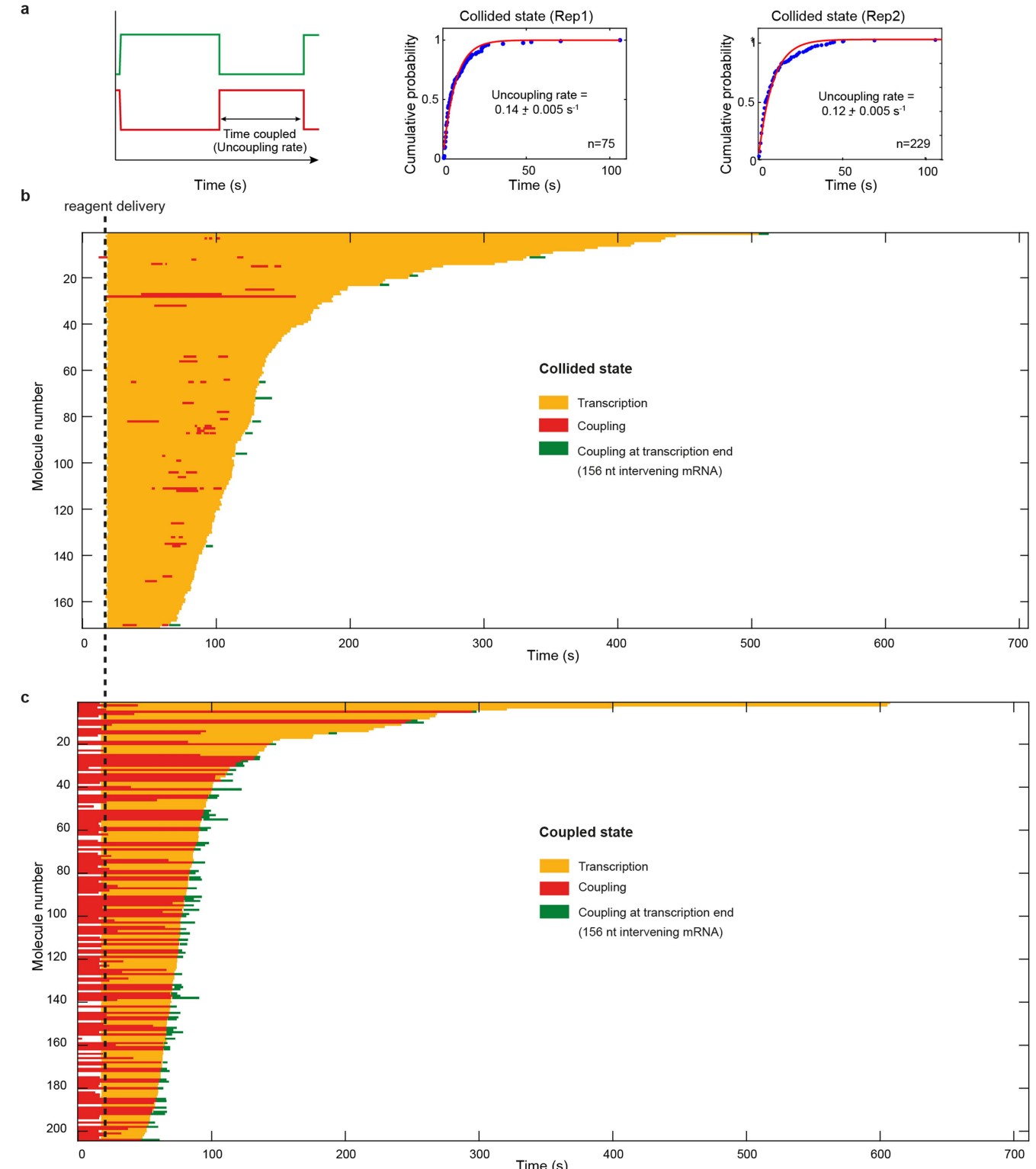

**Extended Data Fig. 6 | Coupling is more transient following a ribosome-RNAP collision. a**, Coupled dwell times (2 biological replicates) during transcription following a collision were fitted to a 1-exponential function, y = 1-exp(-b*t). Evaluated FRET signal is shown on the left. The error represents the 95% confidence intervals of the fit. The number of fitted dwells (n) is indicated for both replicates. **b, c**, Stack of single-molecule traces for transcriptions out of collided (**b**) or coupled (**c**) state. Each row represents a single transcription-translation complex. Experiment start was triggered by delivering 50 μM NTPs to the immobilized and stalled expressome molecules in absence of Nus factors. Transcriptions are depicted in yellow-orange, coupling events are shown in red and traces with coupling at transcription end are marked in green. A large portion of collided expressomes fail to establish coupling completely during subsequent transcription elongation.

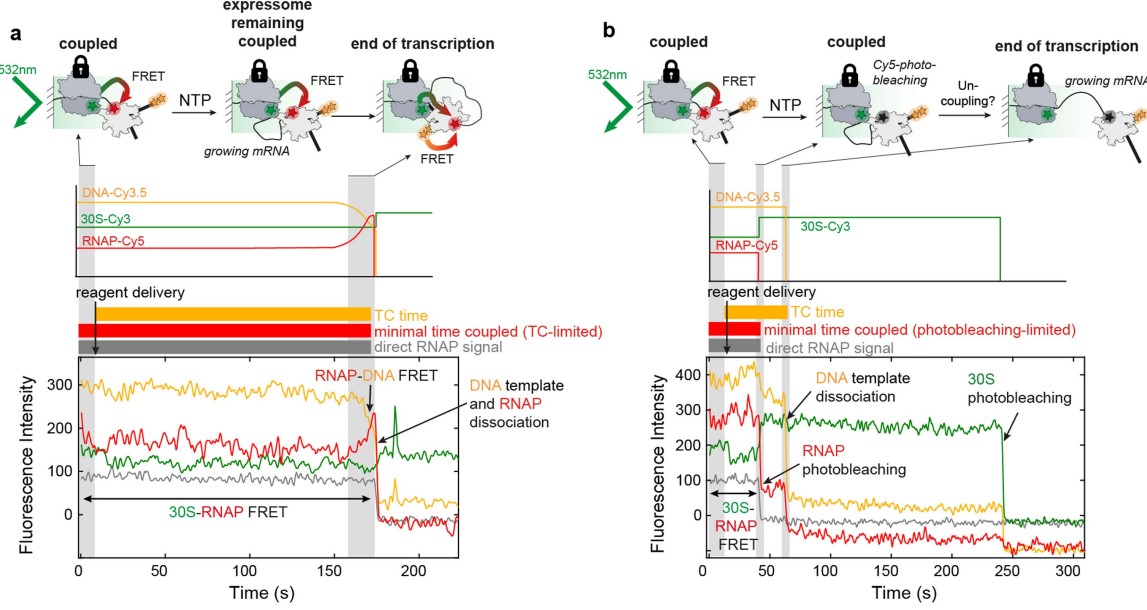

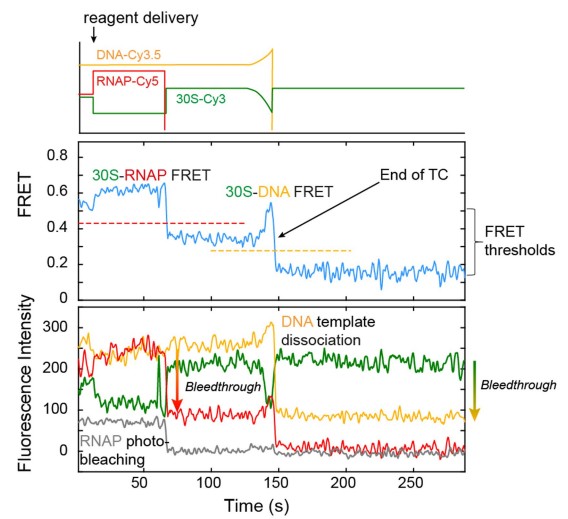

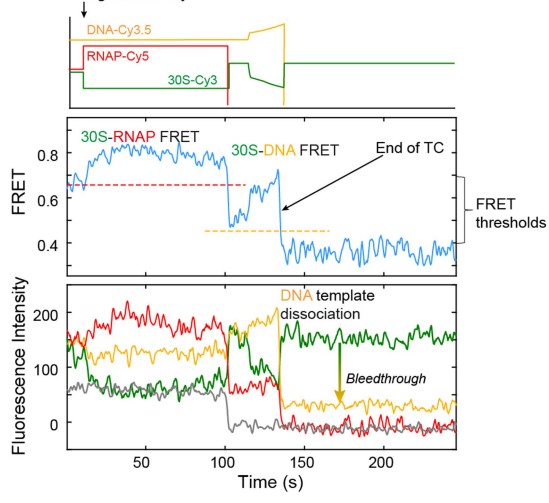

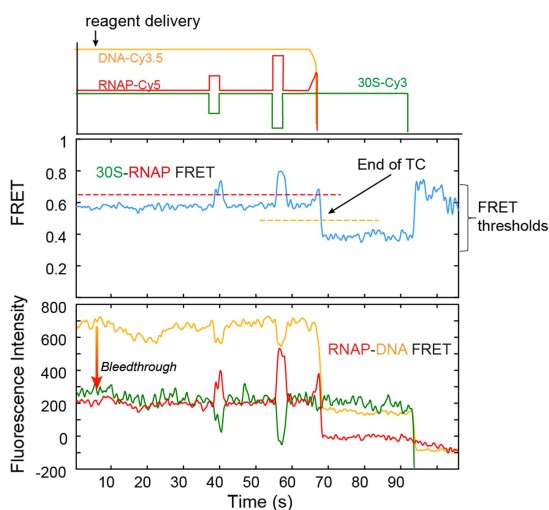

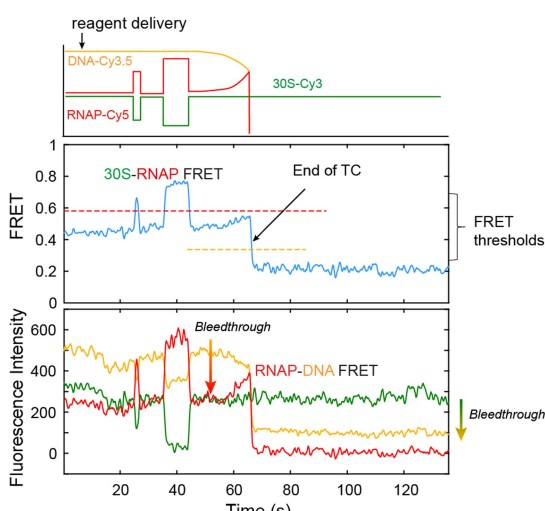

**Extended Data Fig. 7** | See next page for caption.

**Extended Data Fig. 7 | Tracking coupling during transcription elongation.**
Representative smoothed traces for transcription out of coupled (**a-c**) or
collided (**d**) state. Data was acquired with alternative laser excitation at
wavelengths of 532 nm and 638 nm. The reaction was started with delivery
of 50 μM of each NTP. The 70S ribosome was kept stalled on the RBS. **a**, Illustration
of single-molecule trace, where both machines remain coupled throughout
the complete transcription reaction. The steady increase in Cy3.5-Cy5 FRET
efficiency towards the end of transcription (at ~160–170 s) directly shows active
transcription elongation while both machines are coupled containing an
intervening mRNA length of 156 nt. **b**, Single-molecule trace (smoothed) with
photobleaching of the RNAP-Cy5 before transcription is completed. Time-
evolution of coupling cannot be tracked by 30S-Cy3 and RNAP-Cy5 FRET
anymore. Moreover, the expressome uncouples after RNAP-photobleaching
and before transcription end, as also no 30S-Cy3 to DNA-Cy3.5 FRET is detected
(in contrast to Fig. 4a). **c, d**, Assignment of 3-color data by thresholding
exemplified on representative, smoothed traces. Thresholds for 30S-RNAP
FRET transitions are shown as red horizontal dashed lines, which allows for
accurate determination of the coupled dwell times, and thresholds for
30S-DNA (**c**) or DNA-RNAP (**d**) FRET transitions are displayed as yellow orange
horizontal dashed lines, which allows for the accurate determination of
transcription end (see black arrows). Green-yellow arrows depict bleedthrough
from green Cy3-channel to yellow-Cy3.5 channel and yellow-red arrows depict
bleedthrough from yellow-Cy3.5 channel to red-Cy5 channel.

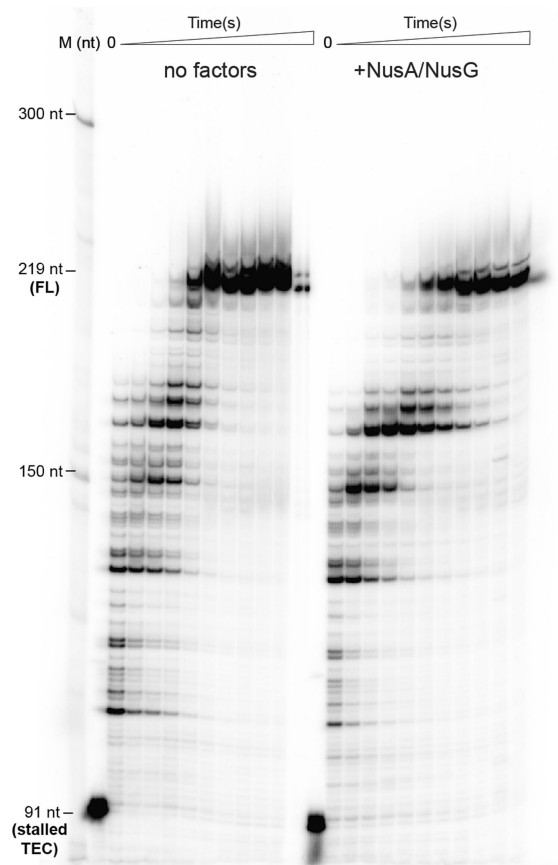 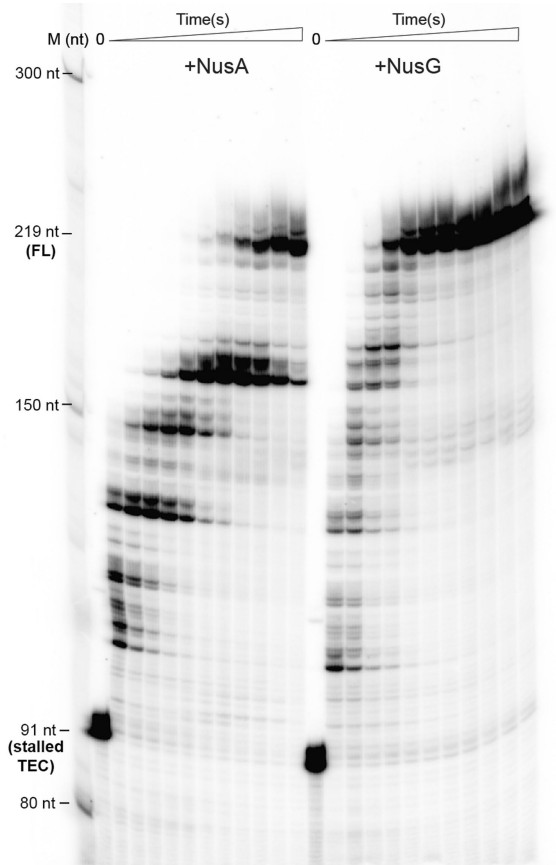

**Extended Data Fig. 8 | Transcription from collided state immediately resumes after addition of NTPs.** Single-round transcription assays with assembled 70S collided expressome using prNQ215 DNA template. Stalled transcription elongation complex (stalled TEC) was formed with 50 nM DNA template, 200 nM *E. coli* RNAP, 100 µM ACU trinucleotide, 5 µM GTP and 5 µM ATP (+150 nM $^{32}$P α-ATP) halting RNAP initially at U24 to prevent loading of multiple RNAPs. Then RNAP was walked to desired stalling site by addition of 10 µM UTP and simultaneous addition of 10 µg/mL rifampicin (to prevent transcription re-initiation). 70S PIC was formed on the stalled TEC in presence of 2 µM IF2a, 1 µM fmet-tRNA$^{fmet}$ and 4 mM GTP. This stalled expressome was chased in presence or absence of NusA and/or NusG with 50 µM NTPs (each) at room temperature and per condition time points were taken at 0, 10, 20, 30, 40, 60, 90, 120, 180, 240, 360 and 600 s. Stalled expressome band (91 nt) was immediately chased after NTP addition (<10 s). The gels stem from one replicate.

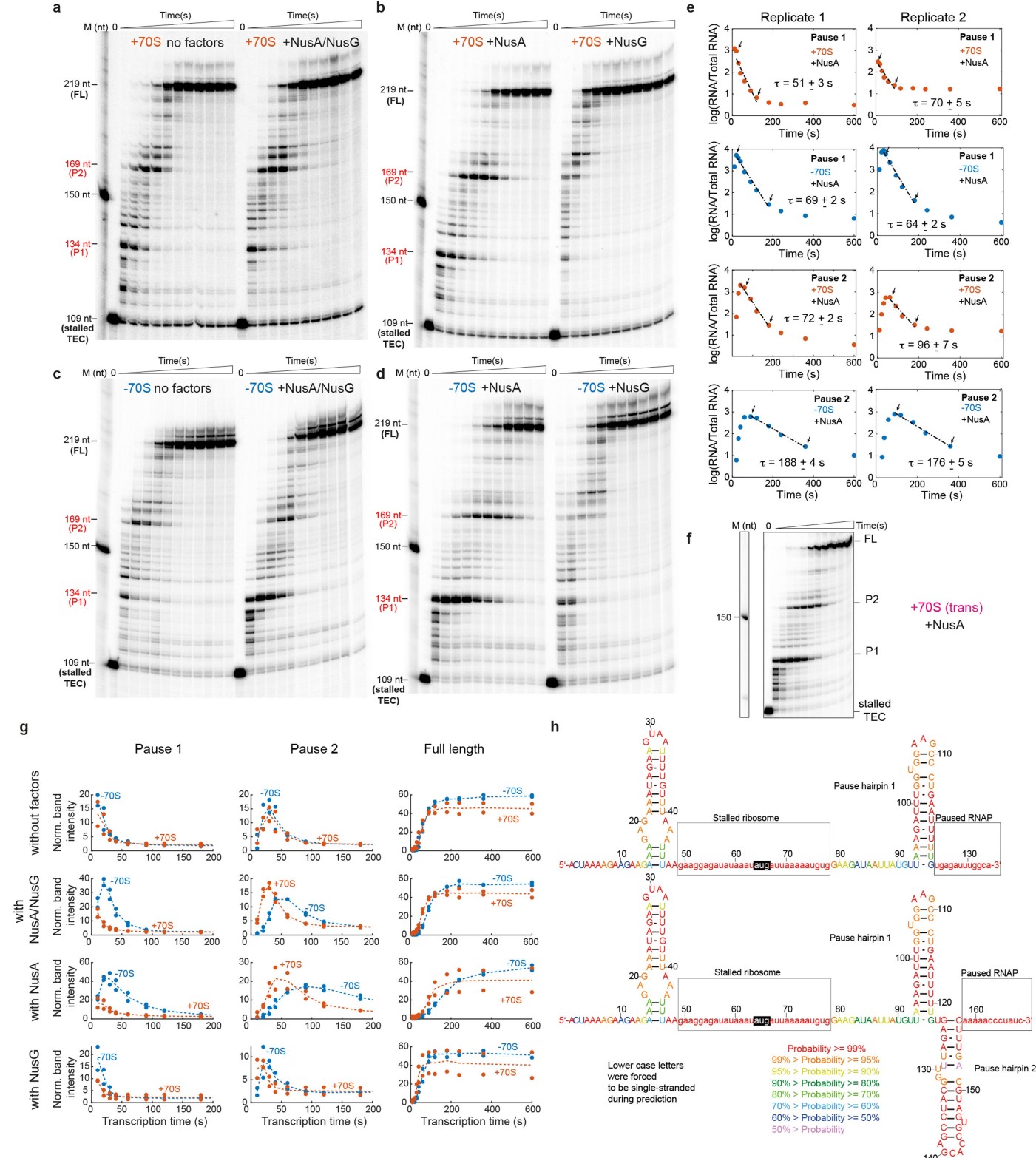

**Extended Data Fig. 9** | See next page for caption.

**Extended Data Fig. 9 | The ribosome activates NusA-paused RNAPs.**
**a-d**, Single-round transcription assays with coupled expressome using the prNQ216 DNA template. h101 of the 50S mutant was pre-annealed with a biotin-oligonucleotide and the formed expressomes were purified using streptavidin beads to enrich for fully assembled transcription-translation complexes. The purified stalled expressome (+70S) or the stalled transcription complex alone (−70S) were chased in presence or absence of Nus factors (w/o factors, w/ NusA/NusG, w/ NusA only, w/ NusG only; 1 μM final concentration for each Nus factor) and with 50 μM NTP (each) at room temperature. For each condition, time points were taken at 0, 10, 20, 30, 40, 60, 90, 120, 180, 240, 360 and 600 s.
**e**, Pause-escape lifetimes for pause 1 and pause 2 in presence of NusA and in presence (orange) or absence (blue) of ribosome. Natural logarithm of normalized band intensities (P/T) was plotted as function of time and pause-escape lifetimes were fitted with a linear fit function (y = m*x + b, with m being the rate constant)[68]. Data range that was used for fitting is indicated with arrows. Pause-escape lifetime (τ) errors were obtained by error propagation of linear least square fit error for the rate constant. **f**, Ribosome can activate RNAP only when sharing the same mRNA: Single-round transcription assays

with mRNA-46 in presence of NusA and with 1 μM 70S (*in trans*, loaded on 6(FK) mRNA) analyzed by denaturing gel electrophoresis. Stalled TEC, pause 1, pause 2 and full-length mRNA bands are shown. The gel stems from one replicate.
**g**, Normalized band intensities (P/T) from gels shown in panels **a-d** are displayed as a function of time. The displayed datapoints are from 2 biological replicates. Bands for pause 1, pause 2 and full-length (FL) RNA were integrated and divided by the total RNA per lane[68]. Parts of panels (**b**), (**d**) and (**g**) are also shown in main Fig. 5e. **h**, Secondary structure prediction of the nascent mRNA (mRNA-46) using the RNA structure web server (https://rna.urmc.rochester.edu/RNAstructureWeb/). Top prediction shows secondary structure at pause site 1 (134 nt) and bottom prediction shows secondary structure at pause site 2 (169 nt). The secondary structure was predicted using default settings on the website for 21 °C, forcing the ribosome binding site (as it is masked by the ribosome) and 12 nt (ref. 43) upstream from 3′-end of the nascent mRNA (as they are masked by the paused RNAP) to be single-stranded. Color code corresponds to probability of base-pair formation. Position of the ribosome and the RNAP are indicated. Start codon (AUG) is highlighted with a black box.

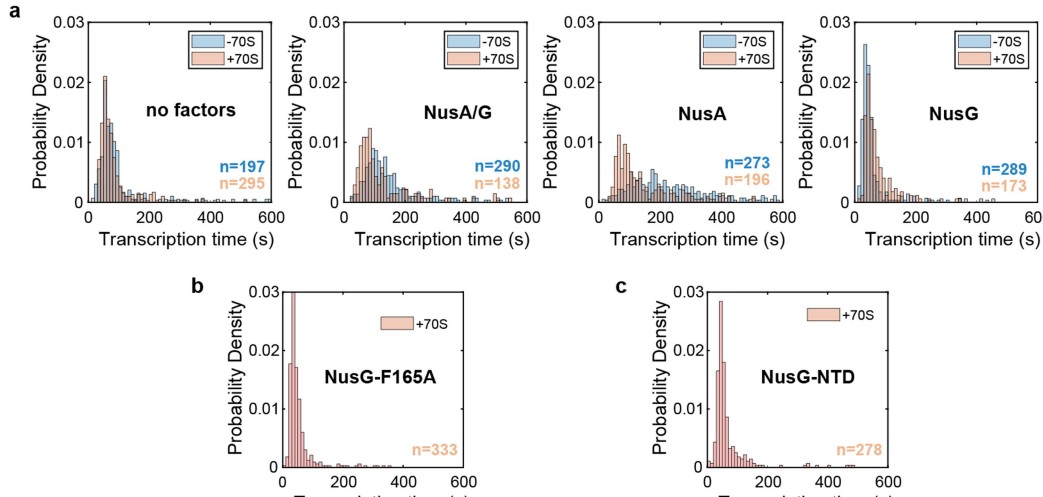

**Extended Data Fig. 10 | The ribosome activates NusA-paused RNAPs.**
**a**, Overlay of probability density distribution for single-molecule transcription times without 70S (blue) and with 70S (orange) in presence and absence of Nus factors. This is a replicate of the data presented in Fig. 5c. Number of evaluated molecules (n) is indicated. **b**,**c**, Probability density distribution for single-molecule transcription times with 70S in presence of 1 μM NusG-NTD (**b**) or 1 μM NusG-F165A (**c**). Number of evaluated molecules (n) is indicated.

# Reporting Summary

## Statistics

For all statistical analyses, confirm that the following items are present in the figure legend, table legend, main text, or Methods section.

| n/a | Confirmed | |
|---|---|---|
| ☐ | ☒ | The exact sample size (*n*) for each experimental group/condition, given as a discrete number and unit of measurement |
| ☐ | ☒ | A statement on whether measurements were taken from distinct samples or whether the same sample was measured repeatedly |
| ☐ | ☒ | The statistical test(s) used AND whether they are one- or two-sided<br>*Only common tests should be described solely by name; describe more complex techniques in the Methods section.* |
| ☒ | ☐ | A description of all covariates tested |
| ☒ | ☐ | A description of any assumptions or corrections, such as tests of normality and adjustment for multiple comparisons |
| ☐ | ☒ | A full description of the statistical parameters including central tendency (e.g. means) or other basic estimates (e.g. regression coefficient) AND variation (e.g. standard deviation) or associated estimates of uncertainty (e.g. confidence intervals) |
| ☐ | ☒ | For null hypothesis testing, the test statistic (e.g. *F*, *t*, *r*) with confidence intervals, effect sizes, degrees of freedom and *P* value noted<br>*Give P values as exact values whenever suitable.* |
| ☒ | ☐ | For Bayesian analysis, information on the choice of priors and Markov chain Monte Carlo settings |
| ☒ | ☐ | For hierarchical and complex designs, identification of the appropriate level for tests and full reporting of outcomes |
| ☒ | ☐ | Estimates of effect sizes (e.g. Cohen's *d*, Pearson's *r*), indicating how they were calculated |

*Our web collection on statistics for biologists contains articles on many of the points above.*

## Software and code

Policy information about availability of computer code

| Data collection | Custom software implemented in Metamorph software package (Molecular Devices) by Cairn Research was used for single-molecule fluorescence microscopy data acquisition. |
|---|---|
| Data analysis | Single-molecule experimental data were analyzed using SPARTAN 3.7.0, tMAVEN and MATLAB R2021a and previous versions (MathWorks) using custom scripts which were previously made available publicly under an open source license at: https://github.com/puglisilab/Lapointe-2022-Nature. Figures were prepared in MATLAB, Adobe Illustrator, Excel 2016, ChimeraX 1.3 and Pymol (v. 2.4). Gel bands were integrated using ImageLab 6.1 software (BioRad).<br>Our Code availability statement reads: "The software used for single-molecule data processing and analysis is published[72,73] and freely available online. Additional scripts for downstream processing were made available publicly previously under an open-source license[74]." |

For manuscripts utilizing custom algorithms or software that are central to the research but not yet described in published literature, software must be made available to editors and reviewers. We strongly encourage code deposition in a community repository (e.g. GitHub). See the Nature Portfolio guidelines for submitting code & software for further information.

## Data

Policy information about availability of data

All manuscripts must include a data availability statement. This statement should provide the following information, where applicable:
- Accession codes, unique identifiers, or web links for publicly available datasets
- A description of any restrictions on data availability
- For clinical datasets or third party data, please ensure that the statement adheres to our policy

> Data availability. Structures were downloaded from the Protein Data Bank (https://www.rcsb.org/) using the accession codes shown in the main figures (6xdq, 6ztm and 6flq) or listed in the Source Data for Extended Data Fig. 3C,D. The published article includes all relevant data generated or analyzed during this study. The source data used in all the figures are presented in the Source Data file. All single-molecule traces, including transition assignments, are deposited in Zenodo with following link: https://zenodo.org/records/13271669. The raw data files are too large (several TBs of data) but are available upon request to O.D.

## Research involving human participants, their data, or biological material

Policy information about studies with human participants or human data. See also policy information about sex, gender (identity/presentation), and sexual orientation and race, ethnicity and racism.

| | |
|---|---|
| Reporting on sex and gender | N/A |
| Reporting on race, ethnicity, or other socially relevant groupings | N/A |
| Population characteristics | N/A |
| Recruitment | N/A |
| Ethics oversight | N/A |

Note that full information on the approval of the study protocol must also be provided in the manuscript.

# Field-specific reporting

Please select the one below that is the best fit for your research. If you are not sure, read the appropriate sections before making your selection.

☒ Life sciences          ☐ Behavioural & social sciences          ☐ Ecological, evolutionary & environmental sciences

For a reference copy of the document with all sections, see nature.com/documents/nr-reporting-summary-flat.pdf

# Life sciences study design

All studies must disclose on these points even when the disclosure is negative.

| | |
|---|---|
| Sample size | As customary in the field, we did not perform statistical analysis to predetermine sample size and no methods were used to predetermine sample size. As much single-molecule data as possible were collected to enable robust analyses. In principle, to obtain a rate constant from a single-molecule data set only a few molecules would be sufficient but the field usually reports 50-100 molecules per experiments. Therefore, we analyzed at least 100 events/molecules if possible and in most experiments several hundreds. |
| Data exclusions | Raw single-molecule time traces were excluded if they 1) consisted of more than one molecule, 2) lacked relevant fluorescence signals (for example, lack of Cy5, Cy3 or Cy3.5 signals corresponding to a fully assembled expressome), 3) were too noisy to unambiguously assign all the relevant events. |
| Replication | Quantitative measurements were reliably reproduced with 2-3 replicates (see extended Data Figures and Source Data) and were in agreement with data from other measurements, if possible. |
| Randomization | Samples were not randomized as this is not applicable to single-molecule biophysical and biochemical studies. Sample randomization is not applicable to biochemical experiments as for a specific experiment a single well-defined condition is measured. For data-evaluation the entire dataset was evaluated. For specific experiments, with much more molecules present that required for evaluation, a random selection of molecules was used for data-evaluation |
| Blinding | Investigators were not blinded as it does not involve clinical data, human or animal models or group allocation and is therefore not applicable to this study. The reasons that blinding is not applicable to biochemical experiments are that biochemical experiments rely on automated processes (such as a single-molecule microscopy measurement) and thus, are minimally affected by human bias. Furthermore, biochemical experiments are highly reproducible and can be verified by repetitions and measurements are very precise and thus have minimal observer bias. |

# Reporting for specific materials, systems and methods

We require information from authors about some types of materials, experimental systems and methods used in many studies. Here, indicate whether each material, system or method listed is relevant to your study. If you are not sure if a list item applies to your research, read the appropriate section before selecting a response.

## Materials & experimental systems

| n/a | Involved in the study |
|---|---|
| ☒ ☐ | Antibodies |
| ☒ ☐ | Eukaryotic cell lines |
| ☒ ☐ | Palaeontology and archaeology |
| ☒ ☐ | Animals and other organisms |
| ☒ ☐ | Clinical data |
| ☒ ☐ | Dual use research of concern |
| ☒ ☐ | Plants |

## Methods

| n/a | Involved in the study |
|---|---|
| ☒ ☐ | ChIP-seq |
| ☒ ☐ | Flow cytometry |
| ☒ ☐ | MRI-based neuroimaging |

## Plants

| | |
|---|---|
| Seed stocks | *Report on the source of all seed stocks or other plant material used. If applicable, state the seed stock centre and catalogue number. If plant specimens were collected from the field, describe the collection location, date and sampling procedures.* |
| Novel plant genotypes | *Describe the methods by which all novel plant genotypes were produced. This includes those generated by transgenic approaches, gene editing, chemical/radiation-based mutagenesis and hybridization. For transgenic lines, describe the transformation method, the number of independent lines analyzed and the generation upon which experiments were performed. For gene-edited lines, describe the editor used, the endogenous sequence targeted for editing, the targeting guide RNA sequence (if applicable) and how the editor was applied.* |
| Authentication | *Describe any authentication procedures for each seed stock used or novel genotype generated. Describe any experiments used to assess the effect of a mutation and, where applicable, how potential secondary effects (e.g. second site T-DNA insertions, mosiacism, off-target gene editing) were examined.* |

