## [Peer Review File · Nature]

Tracking transcription-translation coupling in real-time

Corresponding Author: Dr Olivier Duss

Version 1:

Reviewer comments:

Referee #1

(Remarks to the Author)

Thank you for sending the manuscript by Shahin et al. on tracking transcription translation coupling in real-time. In this manuscript the authors have setup a very powerful and elegant system to monitor transcription and translation and investigate the cooperation between RNA polymerase (RNAP) and a co-translating ribosome. I am convinced this toolbox will allow them to answer many fundamental questions about the cooperation between the two key players in bacterial gene expression and would like to congratulate them on this achievement!

The main claims can be summarized as follows:

- 1) The ribosome slows down as it approaches a stalled RNAP and before it collides;
- 2) The ribosome can couple over large mRNA distances separating the two machineries;
- 3) After a collision, the ribosome and RNAP couple less efficiently;
- 4) The ribosome increases transcription rates (without closely trailing RNAP and without translating).

I have some reservations as far as claim 1 is concerned (see below) and suggestions for additional experiments and would encourage the authors to consider them before a decision on the manuscript can be taken.

Major comments:

1.: How do transcription and translation rates compare to each other in the system of the authors. In a system that can faithfully recapitulate coupling we would expect that the rate of transcription and the rate of translation are comparable. Are translation rates comparable to transcription rates (i.e. is the rate of translation $\sim 1/3$ the rate of transcription)? I think it would be useful to know that explicitly so it is easy for a reader to judge.

2.: One of the claims that the authors make is that the collided state does not occur in vivo based on their observation that the ribosome slows down when approaching RNAP (see last paragraph of their introduction, lines 74-77). This is one of the major findings but I am not convinced the work provides enough evidence to support this claim. First of all, there is strong biochemical evidence that physical collisions occur (Proshkin et al., Science 2010; Stevenson-Jones PNAS 2020; Wee et al., Cell 2022; Woodgate et al., Nature 2024) and as a consequence, we need strong evidence if we want to reject it. Second, there is no structural argument against the formation of a collided state as defined by Wang et al and Webster et al – several independent studies in vitro and in vivo identified the collided state (Kohler et al., Science 2017, Wang et al., Science 2020, Webster et al., Science 2020, O'Reilly et al., Science 2020; Wee et al., Cell 2023) – in other words, RNAP and the ribosome can certainly adopt an orientation that has been called collided.

I am aware it has been argued to be non-physiological by Wang et al., Science 2020, but believe we still need more convincing evidence to be certain. The fact that it has only been observed in vivo upon exposure to a transcription inhibitor shows it is likely a rare and/or short-lived state but does not prove it does not occur.

3.: Along the same lines, I am not convinced the ribosome slows down before (BEFORE being the keyword here) it adopts the collided configuration. The issue here of course is whether or not we use the same metric for the distance between RNAP and the ribosome. I understand from the supplementary material that the authors use the distance from RNA 3'-end to and including ribosomal P-site. Fig. 2f suggests slowdown starts after addition of 4, maybe already 3 amino acids – however, addition of 4 amino acids reduces the distance to 34 nucleotides ($46 - 4 \times 3 = 34$) and at this separation distance the Cryo-EM structures identified already $\sim 100\%$ of the complex adopts the collided state (See Wang et al and Webster et al). Even at a distance of 37 nucleotides (3 amino acids added), the Cryo-EM structures suggested a significant fraction of RNAP adopts the collided configuration. It has been argued that the collided configuration is the result of minimizing the mRNA pathlength between A-site and RNAP active site and maximizing surface complementarity between RNAP and ribosome by Webster et al. and it seems the data presented here is in very good agreement with that. As the intervening

mRNA becomes shorter, the two machineries adopt a collided configuration - translation rates may slow down because the mRNA tension affects its alignment with the ribosomal A-site and/or ribosome translocation requires more force. The ribosome seems to halt after adding 6 amino acids – again, this appears to be in good agreement with biochemical work from Zenkin and co-workers (Stevenson-Jones, PNAS 2020). So my interpretation of the data would be that translation slows down as the mRNA tension between RNAP and the ribosome increases and this correlates with the appearance of a collided configuration (which was named collided because it is the end-result of letting the ribosome collide with a stalled RNAP as done by Kohler et al., Science 2017).

4.: The authors suggest that RNAP and the ribosome are able to couple even if long stretches of mRNA separate them. This is a very interesting observation and I do not want to appear pedantic here but the initial experiment was done without the coupling factor NusG. So what do the authors envision by coupling in this case – physical proximity (which FRET reports on) or physical interaction? I think it is important to clearly differentiate between kinetic coupling and physical coupling and spell it out. For physical coupling it is also important to acknowledge that the evidence to date suggests different interfaces (see cross linking data in Fan et al., NAR 2017, or cryo-EM of Demo et al., eLife 2017, or biochemical evidence for S1-RNAP interaction in Sukhodolets et al., RNA 2006) – do the FRET pairs exclude any of the alternative interfaces and which coupled state is likely adopted in a high-FRET state – presumably one that resembles a NusG-to-S10-coupled state (according to Fig. 3b) as observed in the Cryo-EM reconstructions albeit without NusG. Would any of the other positions (Ebright and co-workers called them TTC-B, TTC-C, and TTC-D, Weixlbaumer and co-workers referred to them as clusters 1-7) also give high FRET (after all, in absence of NusG, RNAP appears to have high degrees of rotational freedom around the mRNA axis).

5.: The authors suggest that in presence of NusG or NusG/NusA the fraction of complexes that remain coupled or remain in a high FRET state (depending on how we define coupled – see previous point) until the end of transcription is higher compared to their absence? Figure 4h, and 4j seem to suggest that this is due to NusG but NusA has little additional effect (e.g. compare w/o Nus and w/ NusA in 4h and 4j, less evident in 4i). I think this is in agreement with the poor conservation of the NusA-ribosome interface and if my interpretation is correct, could be stated more explicitly.

6.: Further to the previous point, if NusG is important to increase the fraction of complexes that remain physically close, an interesting experiment would be to only provide NusG-NTD – this still increases RNAP transcription rates but does no longer form a physical connection to the ribosome and the prediction would be that NusG-NTD should no longer influence the high-FRET state.

7.: The authors suggest the ribosome and RNAP couple less efficiently after a collision occurred and suggest this could be the result of the loss of the omega subunit. I think a simple experiment to verify this potentially important observation is to provide a large excess of omega in trans and see if this rescues coupling after a collision?

8.: The authors claim that the ribosome stimulates overall transcription rates (and in particular escape from a NusA-stabilized, paused RNAP). If I understand their experimental setup correctly, the ribosome does not translate in this setup and RNAP increases the distance to the ribosome by transcribing away from it. In other words, the ribosome cannot exert any physical force on RNAP but it may remain physically coupled (although in absence of NusG, not directly coupled to S10 but somehow still interacting with RNAP). The ribosome has very little (if any) effect in presence of NusG or NusG/NusA but it increases transcription rates (or pause escape rates) by a factor of 2 in presence of NusA alone. If this is true, what would prevent another ribosome (on a different transcript) in the vicinity to do the same? An interesting experiment that the authors may want to consider is to see if adding ribosomes in trans (assembled on an mRNA), can also trigger pause escape or if the ribosome needs to be on the shared mRNA to have the observed effect?

9.: Related to the previous point, as far as I know ref. 34 does not show a NusA-paused RNAP but a NusA-bound elongation complex (which may be more prone to pausing). A NusA paused RNAP complex was described in Guo et al., Mol Cell 2018. This is a minor point but what I would like to stress is that Guo et al. seem to show that NusA exhibits substantial movement relative to RNAP. I therefore wonder if the authors took that into account when proposing the model of NusA repositioning (Fig. 5d). Given NusA's mobility, I think it is unlikely that the author's assay captures NusA repositioning as claimed because NusA is likely fluctuating a lot on its own (and does not require the ribosome to reposition). NusA interacts with the RNAP flap-tip helix, which is flexible and allows NusA to oscillate while the NusA-NTD and NusA-CTD interact with the RNAP alpha-subunit CTDs, which are highly flexible and usually not observed in cryo-EM reconstructions.

Minor comments:

Line 57: The authors mention that head-swivelling is impaired in a collided expressome configuration. I'd like to point out that this is incorrect and there is no structural evidence that this is true. I think we need to avoid propagating it and it should be removed from the introduction.

Line 59 and line 77: The statement referring to the failure of identifying the collided state in vivo unless a transcription inhibitor was included is true but fails to acknowledge that a collided state in vivo is likely rare and transient. It is expected that a collided state can only be observed in cryo-ET studies when a sufficiently large number of ribosomes are captured in this state – and thus, only when an inhibitor was included that stalls all RNAP. However, it is no proof for the absence of this state – it just means it is potentially rare, or short-lived, or both.

Referee #2

(Remarks to the Author)

Quereshi et al generate a new single-molecule FRET microscopy system to interrogate the nature of the interaction between the bacterial RNA polymerase complex and the ribosome. The authors are able to convincingly show that they can track transcription and translation in real time, both in isolated systems and when they expect the systems to be functionally cooperating. Consistent with this expectation, the authors claim to observe translation slowing down as the ribosome approaches a stalled RNA polymerase. Furthermore, they purport to observe that ribosomes are able to functionally cooperate ('couple') across long stretches of mRNA through what they hypothesize is mRNA looping. They present data suggesting that this coupling is facilitated by the protein factors NusA and NusG. The authors conclude that rather than rescuing stalled RNA polymerase by colliding with it, ribosomes are able to reactivate a stalled RNA polymerase by stochastically sampling it over long distances. This conclusion is supported by the authors' claim that collided ribosomes are functionally damaged and fail to fully recapitulate normal coupling dynamics after collision.

This manuscript, the work it describes, and the authors' conclusions will be of broad interest beyond either the transcription or translation fields. It is a tour de force of single-molecule fluorescence imaging that extends the technique to questions of how biomolecular machines interact with each other at the intersection of biological processes. Nonetheless, in my view, the rigor of the single-molecule data analyses and the description of the experiments and analyses are lacking and, at times, flawed. In some instances, additional controls need to be performed and, in certain cases, conclusions need to be toned down. However, after appropriate revisions, I think this manuscript will be a suitable and exciting publication in Nature.

In my view, the manuscript would benefit from:

1. The methods section pertaining to the single-molecule experiments and statistical analyses are confusing and incomplete. All of the following needs to be addressed for this to be suitable for publication:

a. The authors need to explicitly state how many times each experiment was performed and how robust the experiments are to replication.

b. The number of trajectories analyzed in each experiment must be reported. Furthermore, the specific number of transitions fit for each individual dwell time analysis needs to be reported for every experiment.

c. FRET efficiency distributions (1D or 2D) need to be provided for each experiment.

d. The authors need to disclose how transitions were assigned. It is unclear if assignments were made manually, through thresholding, hidden Markov modeling, or some other analysis pipeline. The reader is left to assume the authors manually assigned transitions which, combined with the statement on lines 596-597 "selecting only for productive FRET states", creates a lot of concern regarding how the transitions were assigned. Manual assignment is unacceptable and the conclusions need to be validated using a different, unbiased transition assignment approach.

e. The authors indicate that the movies are taken at 5 frames per second. However, upon visual inspection of the example trajectories, they appear as if they contain much fewer frames. The authors should explicitly state whether the example trajectories are smoothed in any way and if the measurements are continuous or shuttered.

f. The authors make several arguments based on vague assessments of "goodness of fit". For example, the specific criteria the authors used to determine whether a fit was best described as monoexponential or biexponential needs to be explicitly stated.

g. The authors state that all of the data is contained within the paper and that the raw datasets are too large to deposit in a repository. However, much of the data (e.g., a pointer to a repository containing the extracted traces and the traces that were selected for analysis, a larger number of representative trajectories, how many transitions are included in each fit, how many times each experiment was performed, etc.) are missing from the manuscript. Moreover, the extracted traces and the traces selected for analysis can be uploaded to public repositories such as Zenodo, which is frequently used for deposition of single-molecule traces.

2. The authors refer to several structural studies as the basis of their motivation and conclusions in the introduction, throughout the article, and in the discussion. The most recent biochemical and structural work, however, have challenged some of the early structural work that the authors refer to (e.g., conclusions drawn from structural studies of the collided conformation and the physiological role and relevance of this collided conformation is highly debated; see recent work by Ebright and colleagues; Gottesman, Gonzalez, Frank, and colleagues; and Weixelbaum and colleagues, including the 2021 review in Transcription by Weixelbaum and colleagues). The authors should take care to incorporate a discussion of the more recent transcription-translation coupling work and corresponding references into their discussion.

3. The authors assert that the 30S h44 to 50S H101 translational FRET signal only reports on intersubunit rotation stimulated by the delivery of the tRNA and translocation, respectively. This is not correct, however. Multiple studies have shown that the ribosome stochastically fluctuates between the rotated and non-rotated forms and a pair of recent studies, one from Ermolenko and colleagues and one from Puglisi and colleagues have debunked the claim that the h44-H101 signal is insensitive to these stochastic fluctuations. Thus, while it is likely that the progress of translation is positively correlated with the number of fluctuations, these fluctuations cannot be used to precisely map the exact codon position of the ribosome on the mRNA. The authors must address this and Figures 1, 2, and Extended Data Figure 2 should be updated to

reflect this. Renaming the codon-by-codon analysis to “Transition” or “Fluctuation” “1,2,3...” would be sufficient.

4. It is unclear why the authors remark on the rotational state of the subunits when subunit joining occurs, especially considering the authors do not include IF1 or IF3 in their assays and are assembling ribosomal complexes through a non-native pathway. This point should either be clarified to explain its context relative to the work that is presented in the manuscript or just be removed entirely.

5. The experiments depicted in Figures 2e, f, and g, in which the authors drive a ribosome into a stalled RNA polymerase, contain a low overall number of molecules (81). Because most of the ribosomes display <4 sets of transitions, the number of dwell times analyzed to show the translational slow-down is quite low (and not explicitly stated). The authors run these experiments using a low amount of aa-tRNA and EF-G, as demonstrated by their work in Figure 1. These transitions should be sped up by including more aa-tRNA and EF-G so that a higher fraction of ribosomes reach the stalled complex before photobleaching. This would help the authors assess whether the slow transitions the ribosome experiences are due to steric interference preventing the delivery of aa-tRNA and/or EF-G or due to a physical contact with the RNA polymerase.

6. The authors convincingly show FRET between the RNA polymerase and the ribosome over what they claim are large distances of intervening mRNA. To validate this claim, the authors characterize how the rates of RNA polymerase and ribosome coupling and uncoupling depend on the distance of intervening mRNA. The errors reported for these rates are errors of fits, however. Again, the authors should report whether these experiments have been repeated and whether the errors across replicates are different than the errors of the fits reported here. In any case, the largest source of error should be used to assess these data. This is important because the authors' kinetic analyses show only a modest, if any, effect on the coupling rate as a function of distance of intervening mRNA and, perplexingly, show a distance dependence to the uncoupling rate. These unexpected and odd kinetic behaviors need to be addressed in the text. Additionally, their mRNA looping model could be more thoroughly tested by controlling the topological behavior of the RNA:

- a. Hybridizing an oligo to the intervening sequence on one of the shorter mRNA constructs (mRNA 46 or 85) to change the persistence length of the mRNA and measuring the coupling and uncoupling rates.
- b. Hybridizing an oligo which bridges the sequences nearby the ribosome and the RNA polymerase on either mRNA-185 or mRNA-457
- c. Cleaving the intervening RNA sequence with RNase H and characterizing the effect cleavage has on the coupling and uncoupling rates.

Without some form of additional characterization and validation, the conclusion that the mRNA loops during coupling needs to be toned down.

7. The new smFRET signal for detecting coupling is innovative and exciting. A discussion of this signal, including a characterization of how broad the distribution of FRET efficiencies observed in a 1D or 2D plot are, whether multiple FRET efficiency states are sampled, and what these FRET efficiency states might correspond to structurally, should be added.

8. The magnitude of the effects seen for NusG and NusA are surprisingly small. The authors should state how this compares to previously published observations of coupling with these factors. Moreover, when discussing NusA vs NusG vs Nus+NusG, the authors make their arguments solely on the differences they observe in the lifetimes of the interactions. The authors should expand their discussion to include whether the FRET efficiencies change or whether structural differences in the complexes are expected. Due to their muted effects, the authors would greatly benefit from the use of mutant NusA or NusG factors which abrogate binding to either the ribosome or RNA polymerase (e.g., NusG F165A)

9. The authors need to include an SDS-PAGE gel image of the protein factors used in this study.

Referee #3

(Remarks to the Author)

The manuscript by Qureshi and Duss investigates dynamic coupling between the transcribing RNA polymerase (RNAP) and translating ribosome. Transcription-translation coupling (TTC) is thought to guide regulatory decisions in specific operons and aid uninterrupted RNA synthesis globally, by insulating RNAP from premature termination by Rho and rescuing backtracked RNAPs. Coupling has been studied for decades and several recent cryoEM and cryoET structures captured the two machines poised at defined distances between them, revealing different complex architectures, with NusA and NusG proteins acting as bridges. While these structures revealed the molecular interface between the two machines and the bridges, they lack the information on the TTC dynamics in a “real-life” scenario when these complexes are actively moving. Here, the authors established a complete TTC system, which recapitulates the known properties of both machines, and used multi-color fluorescence to follow single molecules in real time. Using single-molecule FRET with strategically positioned fluorophores, the authors can monitor each translation step by analyzing the cyclical conformational changes in the ribosome; the RNA synthesis by average time it takes RNAP to get to the end of the template; and physical interactions (direct coupling) between 70S and RNAP.

The results show that coupling (i) can occur when RNAP and ribosome are separated by hundreds of nts, even though less efficiently as the nascent RNA grows and loops out; (ii) is promoted by NusA and NusG; (iii) can persist throughout transcription; and (iv) can assist RNAP elongation; the latter result was confirmed by bulk transcription assays. Interestingly, the ribosome slows down as it approaches the stalled RNAP to avoid a hard collision, and after the ribosome rear-ends

(artificially stalled) RNAP, the coupling is weakened upon escape, suggesting that a change occurs at the RNAP-ribosome interface during this collision. This change could be due to the loss of omega, seen in structures of collided machines, or a conformational change in RNAP; conformational heterogeneity has been observed in single-molecule analysis of RNA chain elongation. Even though the ribosome slows down, hard collisions may occur when RNAP is stalled for an extended period of time on the cell, and collision-induced changes could have regulatory significance.

The effect of Nus factors on the extent of coupling is inferred from the FRET signal observed between the DNA template 3' end and the 30S ribosome – this signal presumably reflects coupling at the end of the transcription cycle. Not being a FRET expert, I think it is a reasonable explanation. However, I think that in the panel a what is interpreted as DNA template dissociation is at least as likely to be RNA product release – since RNAP has a tendency to hold onto the DNA post termination. I am confused by Figs. 4h & i – in the presence of NusA, 40% of traces remain coupled throughout transcription, but more than 60% are coupled at the end; does NusA bring the other 20% together?

The authors show that the coupled ribosome makes RNA chain elongation less pause prone. In particular, NusA-promoted pausing is alleviated by the ribosome, and the authors argue that NusA repositioning represents an alternative way to activate RNAP, instead of the ribosome pushing RNAP from behind (which would require close coupling). These are different processes, however. NusA-stabilized pauses are typically hairpin-dependent, consistent with the data shown here, whereas backtracking occurs when RNAP can slide back unobstructed, by RNA structures or by a closely-coupled ribosome. NusA repositioning can simply affect its ability to interact with pause hairpins. A conformational change in RNAP induced by physical coupling could make RNAP pause resistant, not only at NusA-sensitive sites but overall.

These are very significant findings as the dynamic interactions of RNAP and 70S machines is monitored for the first time (as far as I know). Bridging over large RNA distances makes sense, and has been observed in other RNA transactions, and raises many questions. The most obvious, raised by the authors, is the control of Rho-dependent termination – the intervening RNA would serve as a loading platform for Rho, so inhibiting termination by mRNA masking seems not feasible. One possibility is NusG, which makes mutually exclusive contacts to Rho or the ribosome; if NusG CTD is bound to S10, it would not be available to aid Rho. But another possibility, supported by the authors data on 70S “activation” of RNAP is a conformational change in RNAP – as Rho requires RNAP to pause to trigger termination.

The manuscript is well written, easy to follow, and experimental design is depicted and described very clearly. The methods section is very thorough and should enable others to reproduce these experiments. The data analysis is convincing and supports the conclusions.

However, figures and legends are very busy and would benefit from simplifying. For example, in Fig. 4c,d,e etc, I would put -, A, G or AG symbols below graphs and the number of corresponding traces below these. Also, there is no need to repeat legends – if anything, not having an extra legend indicates that the two states (coupled and collided) are being compared directly. In Fig. 4d, the overlay does not help, in my opinion – the scales are very different, obviously – maybe just draw lines between left and right panels to stress how different they are.

In Fig. 3f, the scale is wrong – this should be fractions, not %

Fig. 3g should have bars as in f – no point in making this a graph with a real x-axis, and they should be the same regardless. References need to be checked carefully. Ref 8 is the same as Ref 29; there may be others

Version 2:

Reviewer comments:

Referee #1

(Remarks to the Author)

Thank you very much for sending the revised manuscript by Qureshi and Duss. The authors have addressed all my concerns and I suggest publication.

Referee #2

(Remarks to the Author)

This is an impressive, impactful study, and the authors have adequately addressed nearly all of our concerns. From our perspective, the manuscript is ready for publication once the authors have addressed two remaining minor points which can be addressed entirely through textual changes to the manuscript:

1. We would encourage the authors not to apply the ‘visual inspection’ correction to the hidden Markov models. By looking at the data before and after this correction, the correction does not seem to have any effect on the results of the analysis, the interpretation, or the conclusions. Thus, there doesn't seem to be a reasonable need for this correction. Alternatively, if the authors decide to keep this correction, then they should explicitly note the number of transitions that are altered by applying the correction for each experiment under each experimental condition. The ‘TL’ correction, on the other hand, which is applied with clear kinetic constraints, is appropriate and does seem to have an impact, albeit minor, on the analysis.

2. In the revised manuscript, the fact that the traces are smoothed is now described in the Methods. However, in addition to the description in the Methods, the authors should note that the traces were smoothed in all figure legends where smoothed traces are presented.

Referee #3

(Remarks to the Author)

The revised manuscript by Qureshi and Dus has been significantly improved by incorporation of new experiments, providing a more detailed methods section, improving old and adding new figures, and elaborating on a number of points that were somewhat unclear in the original manuscript.

In my opinion, this is an excellent study that would be of interest to a very broad audience, both because it addresses an important biological question at the interface of two steps in gene expression and because it describes powerful multi-color single-molecule fluorescence assays that can be adapted to studies of other cross-talking macromolecular complexes. I think that the revised manuscript is suitable for publication in Nature.

That said, there are a few rough spots in the manuscript that could benefit from editing. I list just a few below and suggests that the authors go through the text slowly to find others.

L139 Add "site" after "active"

L221 delete comma after mRNA

L420 maybe write "but failure to detect this state in vivo in absence of antibiotics was interpreted as evidence that the collisions are transient"

L423 delete "from pausing"

L427 delete "increasingly"

L430 maybe "to rescue arrested backtracked RNAPs"

Of note, NusA-stabilized hairpin pauses are rare – their prominence in the literature reflects the fact that they were the first type of physiologically-relevant pauses identified.

Response to reviewers

Referee #1 (Remarks to the Author):

Thank you for sending the manuscript by Shahin et al. on tracking transcription translation coupling in real-time. In this manuscript the authors have setup a very powerful and elegant system to monitor transcription and translation and investigate the cooperation between RNA polymerase (RNAP) and a co-translating ribosome. I am convinced this toolbox will allow them to answer many fundamental questions about the cooperation between the two key players in bacterial gene expression and would like to congratulate them on this achievement!

The main claims can be summarized as follows:

- 1) The ribosome slows down as it approaches a stalled RNAP and before it collides;
- 2) The ribosome can couple over large mRNA distances separating the two machineries;
- 3) After a collision, the ribosome and RNAP couple less efficiently;
- 4) The ribosome increases transcription rates (without closely trailing RNAP and without translating).

I have some reservations as far as claim 1 is concerned (see below) and suggestions for additional experiments and would encourage the authors to consider them before a decision on the manuscript can be taken.

Major comments:

1.: How do transcription and translation rates compare to each other in the system of the authors. In a system that can faithfully recapitulate coupling we would expect that the rate of transcription and the rate of translation are comparable. Are translation rates comparable to transcription rates (i.e. is the rate of translation $\sim 1/3$ the rate of transcription)? I think it would be useful to know that explicitly so it is easy for a reader to judge.

While in Figure 1 and part of Figure 2 we show that we can simultaneously track transcription and translation elongation, we used the advantage of in vitro reconstitutions to separate the functions of the individual processes. Specifically, in Fig. 2 we study ribosome collision into RNAP and thus, by design, need to stop the transcription machinery. In Fig. 3, we stopped both machineries to study coupling as a function of various defined intervening mRNA lengths, thus, by design stopped both machineries. In Fig. 5 and 6 (previously Fig. 4 and 5), we stopped the translation machinery in order to study 1) how the RNAP escapes a collision, 2) how long-range coupling occurs during active transcription elongation and 3) how the ribosome activates an RNAP even with long intervening mRNAs. To answer all these questions, we need defined and controllable intervening mRNA lengths and conditions and thus, experiments with comparable transcription and translation elongation speeds would complicate data interpretation. However, we understand the reviewer's point and now explicitly also comment on the in vivo translation rate in the result section of the manuscript.

respectively^{16,17,22}. We monitored translation of individual codons over several minutes and reliably quantified the lifetime in the rotated and non-rotated states at up to 500 nM aminoacyl-tRNA (aa-tRNA) and 160 nM EF-G concentrations (Extended Data Fig. 1g) with values one order of magnitude lower than in vivo^{23,24} but in agreement with previous single-molecule experiments^{19,25}.

2.: One of the claims that the authors make is that the collided state does not occur in vivo based on their observation that the ribosome slows down when approaching RNAP (see last paragraph of their introduction, lines 74-77). This is one of the major findings but I am not convinced the work provides enough evidence to support this claim. First of all, there is strong biochemical evidence that physical collisions occur (Proshkin et al., Science 2010; Stevenson-Jones PNAS 2020; Wee et al., Cell 2022; Woodgate et al., Nature 2024) and as a consequence, we need strong evidence if we want to reject it. Second, there is no structural argument against the formation of a collided state as defined by Wang et al and Webster et al – several independent studies in vitro and in vivo identified the collided state (Kohler et al., Science 2017, Wang et al., Science 2020, Webster et al., Science 2020, O'Reilly et al., Science 2020; Wee et al., Cell 2023) – in other words, RNAP and the ribosome can certainly adopt an orientation that has been called collided.

I am aware it has been argued to be non-physiological by Wang et al., Science 2020, but believe we still need more convincing evidence to be certain. The fact that it has only been observed in vivo upon exposure to a transcription inhibitor shows it is likely a rare and/or short-lived state but does not proof it does not occur.

We agree with the reviewer that the collided state may, and probably does, occur in vivo as rare and/or short-lived state even though it was not detected in vivo in absence of a transcription inhibitor. In fact, we have already mentioned this in the discussion of the original version:

ribosome to rescue a back-tracked RNAP from pausing^{10,11}. However, no collided ribosome/RNAP complexes were detected in vivo in absence of antibiotics⁸, which can be explained by our findings that the ribosome slows down before colliding into the RNAP, making ribosome/RNAP collisions less likely to occur frequently. Instead, more efficient

In order to further stress this point, we have now also included following sentence in the introduction section mentioned by the reviewer:

machineries, we find that the ribosome slows down while colliding with the RNAP, providing a biophysical explanation why the collided state has so far not been detected in vivo and may only occur as a less frequent state, for example, when RNAP is stalled on a DNA lesion¹⁴. We

Furthermore, also in response to reviewer 2 point 2, we have extended the discussion, again concluding that the collided state is likely rare but physiological relevant.

The functional relevance of the collided state is debated^{7,36,37}. It was structurally characterized by several independent groups⁶⁻¹⁰ and its inability to be detected in vivo in absence of antibiotics⁸ was interpreted as evidence that the collisions are transient³⁷. However, it was also argued that these collisions are non-physiological⁷. Ribosome-RNAP collisions are a possible mechanism in vitro for the ribosome to rescue a back-tracked RNAP from pausing^{10-12,14} but functionally only at a separation of 28-29 nucleotides of intervening mRNA between RNAP active site and ribosome P-site^{10,14}. Structural studies demonstrate that a substantial fraction of the expressomes adopt a collided conformation at a distance of 37 nucleotides or smaller^{6,7} and our data show that the ribosome starts to increasingly slow down at an intervening mRNA distance in which the expressome preferentially adopts a collided conformation. We therefore postulate that functional collisions, occurring at 28-29 nucleotides intervening mRNA, are likely less frequent events, but relevant, for example, to rescue long-lived, by back-tracking paused RNAPs or to selectively destroy non-functional transcription complexes stalled on DNA lesions¹⁴. Instead, more efficient ribosome-induced RNAP activation from NusA-

3.: Along the same lines, I am not convinced the ribosome slows down before (BEFORE being the keyword here) it adopts the collided configuration. The issue here of course is whether or not we use the same metric for the distance between RNAP and the ribosome. I understand from the supplementary material that the authors use the distance from RNA 3'-end to and including ribosomal P-site. Fig. 2f suggests slowdown starts after addition of 4, maybe already 3 amino acids – however, addition of 4 amino acids reduces the distance to 34 nucleotides ($46 - 4 \times 3 = 34$) and at this separation distance the Cryo-EM structures identified already ~100% of the complex adopts the collided state (See Wang et al and Webster et al). Even at a distance of 37 nucleotides (3 amino acids added), the Cryo-EM structures suggested a significant fraction of RNAP adopts the collided configuration. It has been argued that the collided configuration is the result of minimizing the mRNA pathlength between A-site and RNAP active site and maximizing surface complementarity between RNAP and ribosome by Webster et al. and it seems the data presented here is in very good agreement with that. As the intervening mRNA becomes shorter, the two machineries adopt a collided configuration - translation rates may slow down because the mRNA tension affects its alignment with the ribosomal A-site and/or ribosome translocation requires more force. The ribosome seems to halt after adding 6 amino acids – again, this appears to be in good agreement with biochemical work from Zenkin and co-workers (Stevenson-Jones, PNAS 2020). So my interpretation of the data would be that translation slows down as the mRNA tension

between RNAP and the ribosome increases and this correlates with the appearance of a collided configuration (which was named collided because it is the end-result of letting the ribosome collide with a stalled RNAP as done by Kohler et al., Science 2017).

-This is a very interesting and important point mentioned by the reviewer. In order to address this additional mechanistic aspect raised by the reviewer, we have repeated our translation experiments under colliding conditions at 10-fold higher aa-tRNA and EF-G concentrations. Interestingly, the slow-down of the ribosome upon approaching the RNAP is not much affected by the increased factor concentration, suggesting that the slow-down we observe is indeed because the mRNA tension affects its alignment with the ribosomal A-site and/or ribosome translocation requires more force in the collided conformation as suggested by the reviewer.

-We have changed “before colliding” to “while colliding” in the introduction and results section.

-We now also mention this new interesting aspect in the text, supported by an additional extended Data figure:

By increasing the aa-tRNA and EF-G concentrations 10-fold during colliding conditions, ribosome slow-down is not significantly affected (Extended Data Fig. 2d), suggesting that the physical basis for slow-down is not due to steric interference affecting the delivery of aa-tRNA and/or EF-G but due to increased mechanical force required for performing a translation cycle¹⁰ while structurally being in or transiently visiting the collided state^{6,7}. Lastly, we observe that

-Finally, we have also added text to the discussion (see also previous point 2):

RNAP active site and ribosome P-site^{10,14}. Structural studies demonstrate that a substantial fraction of the expressomes adopt a collided conformation at a distance of 37 nucleotides or smaller^{6,7} and our data show that the ribosome starts to increasingly slow down at an intervening mRNA distance in which the expressome preferentially adopts a collided conformation. We

4.: The authors suggest that RNAP and the ribosome are able to couple even if long stretches of mRNA separate them. This is a very interesting observation and I do not want to appear pedantic here but the initial experiment was done without the coupling factor NusG. So what do the authors envision by coupling in this case – physical proximity (which FRET reports on) or physical interaction? I think it is important to

clearly differentiate between kinetic coupling and physical coupling and spell it out. For physical coupling it is also important to acknowledge that the evidence to date suggests different interfaces (see cross linking data in Fan et al., NAR 2017, or cryo-EM of Demo et al., eLife 2017, or biochemical evidence for S1-RNAP interaction in Sukhodolets et al., RNA 2006) – do the FRET pairs exclude any of the alternative interfaces and which coupled state is likely adopted in a high-FRET state – presumably one that resembles a NusG-to-S10-coupled state (according to Fig. 3b) as observed in the Cryo-EM reconstructions albeit without NusG. Would any of the other positions (Ebright and co-workers called them TTC-B, TTC-C, and TTC-D, Weixlbaumer and co-workers referred to them as clusters 1-7) also give high FRET (after all, in absence of NusG, RNAP appears to have high degrees of rotational freedom around the mRNA axis).

We have added a new extended Data Fig. 3c,d (and corresponding source Data file) reporting all expected distances between both FRET-labels on RNAP and ribosome for all published papers with deposited and released structural data and thereby confirm that we can detect FRET only for the coupled states and that we can detect all published coupled states with our constructs. Weixlbaumer et al. have reported uncoupled states that would show FRET but these were constructs with an intervening mRNA sequence of 38 nt, where proximity without physical coupling could indeed lead to FRET. However, our shortest intervening mRNA construct for the coupled state was 46 nt. Ebright et al., reported coupled states (TTC-C and TTC-D) with distances outside the FRET range but, again, these were reported only for constructs smaller than our 46 nt construct. Finally, deposited structures of complexes between 30S and RNAP from Demo & Korostelev, Elife, 2017 are not relevant in the context of active translation elongation but rather translation initiation and likely will not form between an actively transcribing RNAP and translating ribosome (see also point 8 reviewer 1). Overall, FRET is specific for the coupled state and does not exclude any so far reported coupled states related to TC-TL coupling.

5.: The authors suggest that in presence of NusG or NusG/NusA the fraction of complexes that remain coupled or remain in a high FRET state (depending on how we define coupled – see previous point) until the end of transcription is higher compared to their absence? Figure 4h, and 4j seem to suggest that this is due to NusG but NusA has little additional effect (e.g. compare w/o Nus and w/ NusA in 4h and 4j, less evident in 4i). I think this is in agreement with the poor conservation of the NusA-ribosome interface and if my interpretation is correct, could be stated more explicitly.

This is a good point and we have added a sentence mentioning this aspect:

10b,c). That the coupling is mainly maintained by NusG is in agreement with a highly conserved NusG-ribosome interaction surface in contrast to the poor NusA-ribosome interface conservation^{6-8,39}. Overall, our data show that RNAP and ribosome can remain physically

6.: Further to the previous point, if NusG is important to increase the fraction of complexes that remain physically close, an interesting experiment would be to only provide NusG-NTD – this still increases RNAP transcription rates but does no longer form a physical connection to the ribosome and the prediction would be that NusG-NTD should no longer influence the high-FRET state.

-This is a very good additional control. We have repeated our experiments with NusG-NTD and in addition, with another NusG mutant (F165A), which abolishes binding to the ribosome. Our data confirms that both domains are needed for functional coupling but that transcription is accelerated also in the mutants as in the WT.

-We have added additional text and corresponding Figure and Extended Figures:

around 80 % in presence of both factors (Fig. 5f). In contrast, NusG-NTD³⁸ or NusG-F165A³⁴, two mutants which can interact only with the RNAP, show no increase in functional coupling (Figure 5f) but retain their ability to increase the transcription rate (Extended Data Figure 10b,c). That the coupling is mainly maintained by NusG is in agreement with a highly

7.: The authors suggest the ribosome and RNAP couple less efficiently after a collision occurred and suggest this could be the result of the loss of the omega subunit. I think a simple experiment to verify this potentially important observation is to provide a large excess of omega in trans and see if this rescues coupling after a collision?

This is a good suggestion. In order to address whether addition of omega in trans could rescue coupling after collision, we have recombinantly expressed the omega subunit and have added 1uM omega subunit to our experiments. The total cellular concentration of omega (rpoZ protein product) is around 10uM (Schmidt & Heinemann, Nature Biotechnology, 2016) but most is likely RNAP-bound. We therefore chose a concentration of 1uM for the freely available omega subunit available to bind in trans. However, we do not see any difference in coupling efficiency in presence of omega when transcribing out of the collided state. While this could mean that the omega subunit is not getting ejected upon collision, we think that it is as likely to be a kinetic problem: omega can only bind once the RNAP has escaped from the collided state and now needs to bind RNAP in a situation when the RNAP is actively transcribing away from the ribosome thereby increasing the intervening mRNA sequence length, making coupling increasingly less efficient (especially as omega is not bound yet and maybe binding is more difficult on a potentially changed RNAP interface). Testing all these models is out of the scope for this study but would be interesting to follow-up on future studies dedicated only on this aspect.

While we could remove our hypothesis on how the ribosome/RNAP interface could be changed upon collision (removing blue part below), we prefer to keep it in, also because it is supported by the missing cryoEM density of the omega subunit in two papers (Webster et al. and Wang et al.) and because it provides a good hypothesis to be tested in future studies.

collisions reduce subsequent coupling efficiency. We hypothesize that the more transient nature of coupling following a ribosome-RNAP collision may be attributed to a structural or compositional change of the ribosome/RNAP interface upon collision, for example, the loss of the omega subunit from RNAP, as suggested by the missing electron density in the cryoEM maps of the collided state^{6,7}.

8.: The authors claim that the ribosome stimulates overall transcription rates (and in particular escape from a NusA-stabilized, paused RNAP). If I understand their experimental setup correctly, the ribosome does not translate in this setup and RNAP increases the distance to the ribosome by transcribing away from it. In other words, the ribosome cannot exert any physical force on RNAP but it may remain physically coupled (although in absence of NusG, not directly coupled to S10 but somehow still interacting with RNAP). The ribosome has very little (if any) effect in presence of NusG or NusG/NusA but it increases transcription rates (or pause escape rates) by a factor of 2 in presence of NusA alone.

If this is true, what would prevent another ribosome (on a different transcript) in the vicinity to do the same? An interesting experiment that the authors may want to

consider is to see if adding ribosomes in trans (assembled on an mRNA), can also trigger pause escape or if the ribosome needs to be on the shared mRNA to have the observed effect?

-This is a very important point raised by the reviewer. In order to test whether a ribosome on a different transcript can also activate RNAP or whether RNAP activation can only occur if the ribosome is on the shared mRNA, we added 1uM ribosome (which was assembled on an mRNA) to our pause-escape bulk transcription assays. We did not detect any RNAP activation suggesting that RNAP activation can only occur by a ribosome on the same mRNA. We note that we cannot exclude the possibility that in the cell a ribosome could activate RNAP in trans, because cellular ribosome concentrations are higher. However, we also note that interactions between the transcription elongation complex (in contrast to core RNAP) and mRNA-loaded ribosome (in contrast to vacant 70S) are very weak or even absent in agreement with our data. See, for example, Webster et al, Science, 2020: “Copurification of RNAP with ribosomes was substantially reduced when the mRNA did not support concurrent ribosome binding, but RNAP that lacked DNA or mRNA entirely (RNAP-core) bound ribosomes more stably” and Fan et al., NAR, 2017 show that the TEC is hardly able to interact with the ribosome.

-We think that this point mentioned by the reviewer is important to state in the manuscript and therefore, have now added following sentence and extended Data figure:

decreases the pause escape lifetime 2.2-fold from 182 ± 9 s to 84 ± 17 s). Repeating our experiments by adding an mRNA-assembled ribosome to our actively transcribing RNAPs in trans, we find no ribosome-induced activation (Extended Data Figure 9f) supporting the requirement for a shared mRNA between the two machineries for functional activation.

9.: Related to the previous point, as far as I know ref. 34 does not show a NusA-paused RNAP but a NusA-bound elongation complex (which may be more prone to pausing). A NusA paused RNAP complex was described in Guo et al., Mol Cell 2018. This is a minor point but what I would like to stress is that Guo et al. seem to show that NusA exhibits substantial movement relative to RNAP. I therefore wonder if the authors took that into account when proposing the model of NusA repositioning (Fig. 5d). Given NusA's mobility, I think it is unlikely that the author's assay captures NusA repositioning as claimed because NusA is likely fluctuating a lot on its own (and does not require the ribosome to reposition). NusA interacts with the RNAP flap-tip helix, which is flexible and allows NusA to oscillate while the NusA-NTD and NusA-CTD interact with the RNAP alpha-subunit CTDs, which are highly flexible and usually not observed in cryo-EM reconstructions.

We understand this minor point raised by the reviewer and agree that instead of having the ribosome actively “repositioning” NusA, instead the ribosome rather “prevents” NusA from assuming a pause-promoting conformation by sterical hindrance. We have therefore rephrased this part, have modified the figure accordingly and have removed

our NusA-repositioning assay from the figure panel. Furthermore, we have replaced the NusA-bound elongation complex by the structure of the NusA-paused RNAP complex described in Guo et al (in both structures, NusA assumes a very similar conformation).

Structurally, both NusA-mediated RNAP pausing⁴⁴ as well as NusA-associated coupling in the expressome were described⁷. The superposition of both structures shows that the NusA conformation significantly differs between expressome and NusA-bound RNAP structures and that the NusA pause-promoting conformation cannot occur in the coupled expressome as NusA would clash with the 30S ribosomal subunit (Fig. 6d). Overall, our data are consistent with a model in which long-range coupling between the ribosome and a NusA-mediated paused RNAP prevents NusA to assume a pause-promoting conformation, thereby activating transcription.

Minor comments:

Line 57: The authors mention that head-swivelling is impaired in a collided expressome configuration. I'd like to point out that this is incorrect and there is no structural evidence that this is true. I think we need to avoid propagating it and it should be removed from the introduction.

We have removed this part from the introduction.

Line 59 and line 77: The statement referring to the failure of identifying the collided state in vivo unless a transcription inhibitor was included is true but fails to acknowledge that a collided state in vivo is likely rare and transient. It is expected that a collided state can

only be observed in cryo-ET studies when a sufficiently large number of ribosomes are captured in this state – and thus, only when an inhibitor was included that stalls all RNAP. However, it is no proof for the absence of this state – it just means it is potentially rare, or short-lived, or both.

We have rephrased this part:

machineries, we find that the ribosome slows down while colliding with the RNAP, providing a biophysical explanation why the collided state has so far not been detected in vivo and may only occur as a less frequent state, for example, when RNAP is stalled on a DNA lesion¹⁴. We

We have also extended the discussion on this aspect as pointed out in point 2 of reviewer 1 above.

Referee #2 (Remarks to the Author):

Quereshi et al generate a new single-molecule FRET microscopy system to interrogate the nature of the interaction between the bacterial RNA polymerase complex and the ribosome. The authors are able to convincingly show that they can track transcription and translation in real time, both in isolated systems and when they expect the systems to be functionally cooperating. Consistent with this expectation, the authors claim to observe translation slowing down as the ribosome approaches a stalled RNA polymerase. Furthermore, they purport to observe that ribosomes are able to functionally cooperate ('couple') across long stretches of mRNA through what they hypothesize is mRNA looping. They present data suggesting that this coupling is facilitated by the protein factors NusA and NusG. The authors conclude that rather than rescuing stalled RNA polymerase by colliding with it, ribosomes are able to reactivate a stalled RNA polymerase by stochastically sampling it over long distances. This conclusion is supported by the authors' claim that collided ribosomes are functionally damaged and fail to fully recapitulate normal coupling dynamics after collision.

This manuscript, the work it describes, and the authors' conclusions will be of broad interest beyond either the transcription or translation fields. It is a tour de force of single-molecule fluorescence imaging that extends the technique to questions of how biomolecular machines interact with each other at the intersection of biological processes. Nonetheless, in my view, the rigor of the single-molecule data analyses and the description of the experiments and analyses are lacking and, at times, flawed. In some instances, additional controls need to be performed and, in certain cases, conclusions need to be toned down. However, after appropriate revisions, I think this manuscript will be a suitable and exciting publication in Nature.

In my view, the manuscript would benefit from:

1. The methods section pertaining to the single-molecule experiments and statistical

analyses are confusing and incomplete. All of the following needs to be addressed for this to be suitable for publication:

a. The authors need to explicitly state how many times each experiment was performed and how robust the experiments are to replication.

We have generated a Source Data file containing all data presented in the figures and state how many replicates were performed for each dataset. We also mention this now in the figure captions. Furthermore, to demonstrate how robust the key experiments are to replication, we compare replicates of all key experiments in several Extended Data figures. Specifically, we compare:

- replicates for Fig. 2 in Extended Data Fig. 1f and in the Source Data file related to Fig. 2F to support slow-down of ribosome while colliding
- replicates for updated Fig. 3 & new Fig. 4 in Extended Data Figure 4 to support decreasing coupling efficiency with increasing mRNA length
- replicates for Fig. 6c (previously Fig. 5c) in Extended Fig. 10a and replicates for Fig. 6e (previously Fig. 5f) in Extended Fig. 9e,g to support functional activation of NusA-paused RNAP by the ribosome
- additional replicates in other Extended Data Figures.

For all other experiments, we indicated error bars from different biological replicates with the raw data points summarized in the Source Data file.

Below is one example for a replicate shown as Extended Data Figure 4a:

b. The number of trajectories analyzed in each experiment must be reported. Furthermore, the specific number of transitions fit for each individual dwell time analysis needs to be reported for every experiment.

In addition to the number of evaluated molecules/trajectories, we have now also indicated the number of dwells times evaluated for each experiment in the figure captions or the figures panels themselves. Please see also more information in the new Source Data file.

c. FRET efficiency distributions (1D or 2D) need to be provided for each experiment.

We note that for our 3-color experiments with the spectrally overlapping Cy3, Cy3.5 and Cy5 dyes, FRET efficiency distributions will be highly biased by the spectral bleedthrough from the processes tracked with the Cy3.5 channel and therefore, are not meaningful to be reported. However, we understand the reviewers point and have therefore remeasured all the data which can be meaningfully recorded with only two colors (Cy3 and Cy5) and have plotted all FRET efficiency distribution histograms (as replicates; see previous point). They are reported in new main Figs. 3 and 4 and Extended Data Figs. 2 and 4.

d. The authors need to disclose how transitions were assigned. It is unclear if assignments were made manually, through thresholding, hidden Markov modeling, or some other analysis pipeline. The reader is left to assume the authors manually assigned transitions which, combined with the statement on lines 596-597 “selecting only for productive FRET states”, creates a lot of concern regarding how the transitions were assigned. Manual assignment is unacceptable and the conclusions need to be validated using a different, unbiased transition assignment approach.

For the 3-color data, we have assigned the transitions using thresholding as described in detail in the methods section of Duss et al., Nature communications, 2018 including its corresponding Supplementary Figure 11 and with the same data evaluation pipeline used in the Puglisi lab for all multi-color single-molecule studies to date (e.g. Lapointe & Puglisi, Nature, 2022). In order to more specifically illustrate how thresholding was performed with the 3-color data presented in this manuscript, we have added several representative single-molecule traces of different experiment types into Extended Data Fig. 7 (see also Extended Data Fig. 1) and have significantly extended the methods section. We note that with the spectrally overlapping Cy3, Cy3.5 and Cy5 dyes used in our single-molecule experiments, assignment by hidden Markov modeling would be highly biased or impossible due the spectral bleedthrough from the processes tracked with the Cy3.5 channel (as already mentioned in point 1c above).

In order to validate and further support the assignment of our 3-color data using thresholding, especially the translation elongation data in Fig. 2, we have repeated all 3-color experiments (which can be meaningfully recorded with only two colors Cy3 and Cy5) and automatically assigned the transitions with hidden markov modeling using the newest published tMAVEN pipeline (see also point 3 of reviewer 2 for more details on the assignment of the translation elongation data). We have substantially extended our methods sections to illustrate our transition assignment including the addition of Extended Data Fig. 1 and Extended Data Fig. 7.

e. The authors indicate that the movies are taken at 5 frames per second. However, upon visual inspection of the example trajectories, they appear as if they contain much fewer frames. The authors should explicitly state whether the example trajectories are smoothed in any way and if the measurements are continuous or shuttered.

The data was smoothed for representation by zero-phase digital filtering using the matlab `filtfilt` function, smoothing 3 frames. We have described this now in the methods section. Of note, the HMM assignment in tMAVEN was performed on unfiltered data.

For representation, the single-molecule traces were smoothed by zero-phase digital filtering by 3 points using the `filtfilt` function in matlab, but unsmoothed data was used for data evaluation.

f. The authors make several arguments based on vague assessments of “goodness of fit”. For example, the specific criteria the authors used to determine whether a fit was best described as monoexponential or biexponential needs to be explicitly stated.

We have added a sentence to the methods section to explicitly state how we decided to fit to single versus bi-exponential fit.

non-linear least squares methods). If initial double-exponential fitting of the data yielded a population that was represented less than 10 %, the data was classified as following single-exponential kinetic behavior⁷⁴.

We have also extended the Statistics section in the methods part.

g. The authors state that all of the data is contained within the paper and that the raw datasets are too large to deposit in a repository. However, much of the data (e.g., a pointer to a repository containing the extracted traces and the traces that were selected for analysis, a larger number of representative trajectories, how many transitions are included in each fit, how many times each experiment was performed, etc.) are missing from the manuscript. Moreover, the extracted traces and the traces selected for analysis can be uploaded to public repositories such as Zenodo, which is frequently used for deposition of single-molecule traces.

We have now deposited all single-molecule traces used in the entire study in Zenodo with following link: <https://zenodo.org/records/13271669>.

We now also show more representative traces in Extended Data Figs. 1 and 7.

As already pointed out in points 1a and 1b of reviewer 2, we have generated a Source Data file containing all data presented in the figures and how many replicates were performed for each experiment and how many transitions are included in the fits.

2. The authors refer to several structural studies as the basis of their motivation and conclusions in the introduction, throughout the article, and in the discussion. The most recent biochemical and structural work, however, have challenged some of the early structural work that the authors refer to (e.g., conclusions drawn from structural studies

of the collided conformation and the physiological role and relevance of this collided conformation is highly debated; see recent work by Ebricht and colleagues; Gottesman, Gonzalez, Frank, and colleagues; and Weixelbaum and colleagues, including the 2021 review in Transcription by Weixelbaum and colleagues). The authors should take care to incorporate a discussion of the more recent transcription-translation coupling work and corresponding references into their discussion.

We have now substantially extended the discussion and cited all relevant and also more recent papers that we are aware of and hope we did not miss one.

The functional relevance of the collided state is debated^{7,36,37}. It was structurally characterized by several independent groups⁶⁻¹⁰ and its inability to be detected in vivo in absence of antibiotics⁸ was interpreted as evidence that the collisions are transient³⁷. However, it was also argued that these collisions are non-physiological⁷. Ribosome-RNAP collisions are a possible mechanism in vitro for the ribosome to rescue a back-tracked RNAP from pausing^{10-12,14} but functionally only at a separation of 28-29 nucleotides of intervening mRNA between RNAP active site and ribosome P-site^{10,14}. Structural studies demonstrate that a substantial fraction of the expressomes adopt a collided conformation at a distance of 37 nucleotides or smaller^{6,7} and our data show that the ribosome starts to increasingly slow down at an intervening mRNA distance in which the expressome preferentially adopts a collided conformation. We therefore postulate that functional collisions, occurring at 28-29 nucleotides intervening mRNA, are likely less frequent events, but relevant, for example, to rescue long-lived, by back-tracking paused RNAPs or to selectively destroy non-functional transcription complexes stalled on DNA lesions¹⁴. Instead, more efficient ribosome-induced RNAP activation from NusA-

3. The authors assert that the 30S h44 to 50S H101 translational FRET signal only reports on intersubunit rotation stimulated by the delivery of the tRNA and translocation, respectively. This is not correct, however. Multiple studies have shown that the ribosome stochastically fluctuates between the rotated and non-rotated forms and a pair of recent studies, one from Ermolenko and colleagues and one from Puglisi and colleagues have debunked the claim that the h44-H101 signal is insensitive to these stochastic fluctuations. Thus, while it is likely that the progress of translation is positively correlated with the number of fluctuations, these fluctuations cannot be used to precisely map the exact codon position of the ribosome on the mRNA. The authors must address this and Figures 1, 2, and Extended Data Figure 2 should be updated to reflect this. Renaming the codon-by-codon analysis to “Transition” or “Fluctuation” “1,2,3...” would be sufficient.

We are aware that depending on the experimental conditions, spontaneous intersubunit rotations can complicate the analysis and assignment of real translation transitions. However, we chose experimental conditions in which we can minimize the interference from spontaneous intersubunit rotations. We are using polymix buffer, which was shown by the most recent study from the Ermolenko lab (Das & Ermolenko, JMB, 2023) to reduce the fraction of ribosomes showing spontaneous fluctuations by 20-fold with only 1% of the h44-cy3/H101-cy5 ribosomes showing spontaneous fluctuations (Das & Ermolenko, JMB, 2023). Furthermore, we chose slower translation conditions in which the timescale of the spontaneous intersubunit rotations is faster ($0.3\text{-}2\text{ s}^{-1}$; see Das & Ermolenko, JMB, 2023) and not even all spontaneous fluctuations may be detected at our framerate of 200 ms. The difference in timescales between real translations and intersubunit fluctuations is especially pronounced during ribosome slow-down for which the timescale of translation and spontaneous intersubunit rotations can differ by 2 orders of magnitude.

However, we understand that this is an important point and should be addressed and explained in the manuscript.

While in our original 3-color datasets, we did not observe significant evidence for spontaneous fluctuations (Figure 2), we repeated these experiments under more sensitive 2-color conditions both with Cy3 and Cy3B. Remarkably, we can observe for a subset of molecules, trains of spontaneous intersubunit rotations when the ribosome remains trapped in the rotated state after collision. This is in agreement with observations that spontaneous intersubunit fluctuations are significantly diminished when shifted to the non-rotated state (Das & Ermolenko, JMB, 2023). In order to automatically separate the short spontaneous fluctuations from the longer real translation transitions, we first automatically assigned the 2-color data using hidden Markov modeling using the newly published tMAVEN pipeline and then truncated each trace after the occurrence of two consecutive short-lived ($<5\%$ tile time of normal translations; see methods and Extended Data Fig. 1e) non-rotated state transitions (probability of occurrence under normal translations is 0.25%; see methods) in order to remove all spontaneous fluctuations that appear once the ribosome is entirely collided with the polymerase and remaining trapped in the rotated state. Using this unbiased approach, we find that the majority of the traces show a maximum of 6 translation cycles till collision supporting our ability to separate real translations from spontaneous fluctuations.

We have significantly extended the methods section, including new Extended Data Fig. 1c-e, and now also specified this in the main text and refer also to a new supplementary methods section.

[Translation elongation at the single amino-acid resolution can be monitored by the 30S-h44/50S-h101 FRET pair under our experimental conditions (see methods and Supplementary Methods), which reports on the two main steps of translation elongation: peptidyl transfer and

4. It is unclear why the authors remark on the rotational state of the subunits when subunit joining occurs, especially considering the authors do not include IF1 or IF3 in their assays and are assembling ribosomal complexes through a non-native pathway. This point should either be clarified to explain its context relative to the work that is presented in the manuscript or just be removed entirely.

We agree with the reviewer that this information is not relevant and have therefore removed it from the manuscript.

5. The experiments depicted in Figures 2e, f, and g, in which the authors drive a ribosome into a stalled RNA polymerase, contain a low overall number of molecules (81). Because most of the ribosomes display <4 sets of transitions, the number of dwell times analyzed to show the translational slow-down is quite low (and not explicitly stated). The authors run these experiments using a low amount of aa-tRNA and EF-G, as demonstrated by their work in Figure 1. These transitions should be sped up by including more aa-tRNA and EF-G so that a higher fraction of ribosomes reach the stalled complex before photobleaching. This would help the authors assess whether the slow transitions the ribosome experiences are due to steric interference preventing the delivery of aa-tRNA and/or EF-G or due to a physical contact with the RNA polymerase.

-In order to address the first point, we recorded an additional two full dataset for each colliding and elongation conditions. In addition to the original 3-color data presented in the manuscript, we repeated the experiments using simpler two-color experiments both with Cy3 and Cy3B (see also point 3 reviewer 2 above). We merged the data from all three datasets for each elongating and colliding conditions, obtaining 200-300 dwells for most transitions of the first 6 amino acids (see Fig. 2e-g, methods and Source Data file), further confirming our findings of ribosome slow-down upon collision.

-To address the second, very interesting, point of the reviewer to increase aa-tRNA and EF-G concentrations, we recorded data at 10-fold higher EF-G (1.5 μ M) and aa-RNA (500 nM) concentrations. As already detailed in point 3 of reviewer 2 above, spontaneous fluctuations can only be separated from normal translations at appropriately slow translation elongation rates. In order to address the question on whether increased factor concentrations would lead to an increased translation rate also during colliding conditions, we were not able anymore to track translation elongation at the single-amino acid resolution. Instead, we plotted the longest, second longest or third longest rotated dwell time per trace, an approach which is less sensitive to short spontaneous fluctuations, and compared these for 1) elongating, 2) colliding at low and 3) colliding at high factor concentrations. Remarkably, these data indicate that translation rates during collisions are not significantly affected by the factor concentrations suggesting that the slow-down we observe is not due to steric hindrance preventing factor binding but rather due to increased forces due to physical ribosome/RNAP contacts.

-We now also mention this new interesting aspect in the text, supported by an additional extended Data figure:

By increasing the aa-tRNA and EF-G concentrations 10-fold during colliding conditions, ribosome slow-down is not significantly affected (Extended Data Fig. 2d), suggesting that the physical basis for slow-down is not due to steric interference affecting the delivery of aa-tRNA and/or EF-G but due to increased mechanical force required for performing a translation cycle¹⁰ while structurally being in or transiently visiting the collided state^{6,7}. Lastly, we observe that

6. The authors convincingly show FRET between the RNA polymerase and the ribosome over what they claim are large distances of intervening mRNA. To validate this claim, the authors characterize how the rates of RNA polymerase and ribosome coupling and uncoupling depend on the distance of intervening mRNA. The errors reported for these rates are errors of fits, however. Again, the authors should report whether these experiments have been repeated and whether the errors across replicates are different than the errors of the fits reported here. In any case, the largest source of error should be used to assess these data. This is important because the authors' kinetic analyses show only a modest, if any, effect on the coupling rate as a function of distance of intervening mRNA and, perplexingly, show a distance dependence to the uncoupling rate. These unexpected and odd kinetic behaviors need to be addressed in the text.

We agree with the reviewer that we see only a modest effect of the coupling rate as a function of the intervening mRNA (now shown as Extended Data Fig. 5d with a replicate and with errors as standard deviation from replicates reported in the main text).

(Extended Data Fig. 5a) with recoupling becoming slower with increasing intervening mRNA lengths, taking in average 10.1 ± 1.4 s and 24.3 ± 5 s to recouple for the mRNA-85 and mRNA-193 constructs, respectively (Extended Data Fig. 5d, e-h).

Our interpretation of the increased uncoupling rate with increasing intervening distance is that increasing "RNA mass" between RNAP and ribosome pushes the two machines apart. As we had already mentioned in the main text and extended Data Fig 4 of the original manuscript, the majority of the traces remain coupled till photobleaching and only a small fraction of the coupled traces shows dynamic uncoupling/recoupling, and this small fraction provides the information for the kinetic interpretation of the coupling/uncoupling rates. We have therefore repeated our measurements to increase the observation time till photobleaching (optimizing oxygen removal). When selecting only traces for which photobleaching of the Cy5-RNAP occurs after 100 seconds, we find again that the majority of the coupled lifetimes are photobleaching limited with a median dwell time of 150 sec for the three shortest construct lengths (see new extended Data Fig. 5c). This indicates that coupling for the majority of molecules is long enough to allow coupling during transcription of an entire gene, except after a collision where coupling becomes more transient as detailed in original Figure 4 (now Fig. 5). We therefore rewrote (shortened) this part in order to discuss the new and more relevant data on how the FRET efficiency distributions change with increasing intervening mRNA as explained in detail in the next point 7.

Additionally, their mRNA looping model could be more thoroughly tested by controlling the topological behavior of the RNA:

a. Hybridizing an oligo to the intervening sequence on one of the shorter mRNA constructs (mRNA 46 or 85) to change the persistence length of the mRNA and measuring the coupling and uncoupling rates.

These are great experiments suggested by the reviewer. We have hybridized DNA oligos to both intervening mRNAs 46 and 85 and have plotted the FRET efficiency distributions (see also next point 7 requested by the reviewer) in presence and in absence of the oligos (see Extended Data Figure 4a-d). While hybridization of a 15 nt long DNA oligo only shows a slight effect on the FRET efficiency distribution of the mRNA-46, upon 51 nt long oligo hybridization to the mRNA-85, the coupled states are completely lost. We also note that coupling/uncoupling dynamics cannot be determined here as they are completely abolished because both oligos are long enough such that the DNA-oligo /RNA duplex is expected to be significantly longer-lived than the experimental time (see e.g. Xu & Zhou, NatComm, 2017: [10.1038/ncomms14902](https://doi.org/10.1038/ncomms14902)).

We have also added an additional sentence into the main text.

state TTC-B ($E_{\text{FRET}} \sim 0.3$)^{6,7}. Targeted hybridization of DNA oligonucleotides to the intervening mRNA, results in an increase in the inter ribosome-RNA distance distribution for mRNA-46 and a complete loss of coupling for mRNA-85 and the longer mRNAs (Extended Data Fig. 4b-d) further validating our assignment. Using this classification, we plotted the

b. Hybridizing an oligo which bridges the sequences nearby the ribosome and the RNA polymerase on either mRNA-185 or mRNA-457

We have designed two different DNA oligos for both longer RNA lengths to bridge ribosome and RNAP as suggested by the reviewer but we were not successful in increasing the coupling efficiency (see plot above). In fact, we even lost coupling completely also for the 193 nt construct. However, this is not unexpected. First, it is possible that due to the strong propensity for RNA structure formation (see Duss et al., Cell, 2019 or Rodgers et al., Cell, 2019) hybridization of the long bridging DNA oligos is very inefficient or incomplete with the experimental inability to heat-anneal the oligos to the large expressome complex. Second, even if the oligos were properly hybridized, the RNA mass (including the additional bridging DNA oligo), may actually rather push away the two machines from each other rather than bringing them closer.

c. Cleaving the intervening RNA sequence with RNase H and characterizing the effect cleavage has on the coupling and uncoupling rates.

We are not sure whether we understood the suggested experiment properly but note that upon cleaving the intervening mRNA by DNA oligo hybridization and RNase H, the two machines would be completely separated and recoupling could only occur if both ribosome and RNAP can interact with each other without sharing the same mRNA. Due to the low immobilization efficiency on a cover slip (and washing after immobilization), after RNase H cleavage, the concentration of the resulting mRNA-loaded ribosome in solution would be extremely low with a very low chance of rebinding (“recoupling”). Nevertheless, also in response to reviewer 1 point 8, we have tested whether a mRNA-associated ribosome at a concentration of 1 μM could have functional coupling and we do not detect any effect, in agreement with previous finding that mRNA-associated ribosomes and TECs do hardly interact with each other (see more details in reviewer 1 point 8).

Without some form of additional characterization and validation, the conclusion that the mRNA loops during coupling needs to be toned down.

Overall, our additional data presented above (for example, complete uncoupling upon DNA oligo hybridization to 85nts construct) and our new data presented in the next point 7 (FRET efficiency distributions change with increasing intervening mRNA length) as well as our new data on Nus-factor dependent compaction in mRNA-85 construct (see point 8) further support our model for mRNA looping. Finally, we would also like to stress that the real-time transcription data with stalled ribosome in Fig. 5a,b (original

Fig. 4a,b), and new Extended Data Fig. 7, with a very characteristically increasing FRET signal between 30S-cy3 and DNA-cy3.5, or DNA-cy3.5 and RNAP-cy5, when the RNAP slowly approaches the 3'-end of the DNA template represents additional very strong evidence for mRNA looping during long-range coupling.

7. The new smFRET signal for detecting coupling is innovative and exciting. A discussion of this signal, including a characterization of how broad the distribution of FRET efficiencies observed in a 1D or 2D plot are, whether multiple FRET efficiency states are sampled, and what these FRET efficiency states might correspond to structurally, should be added.

We thank the reviewer for this important suggestion, which have provided very exciting new results. We have plotted FRET efficiency histograms as new additions to Figures 3 and with repetitions shown in Extended Data Figure 4 and have added an additional results section.

order to distinguish uncoupling events from photobleaching (methods). By plotting the FRET efficiency distributions, we could describe the inter ribosome-RNAP distance distribution for the ensemble of the expressome molecules. We observe a gradual shift of the FRET efficiency distribution towards smaller values in agreement with increasing RNAP-ribosome separation when increasing intervening mRNA length (Fig. 3f and Extended Data Fig. 4a). For mRNA-46, we find two well-defined peaks, which we assign to uncoupled and coupled expressomes, in agreement with the reported cryoEM structures^{6,7} (Extended Data Figure 3). In contrast, for mRNA-85, we observe a broadening of the distribution, which shifts towards a single peak with $E_{\text{FRET}} \sim 0$ for the longest mRNA-457. While hidden Markov modeling (HMM) allows the assignment of various states (3 states predicted; see methods), dozens of different cryoEM structures were determined describing a broad structural distribution of coupled expressomes^{6,7} (Extended Data Figure 3c, d and Source Data), including a new preprint describing additional loosely coupled states (TTC-LC) detected with intervening mRNA sequences longer than 50 nucleotides³³. Therefore, unambiguous detection and assignment of the various structural states to specific E_{FRET} values is not possible. However, taking the available structural knowledge into account, we can classify our data into three broad classes: uncoupled complexes ($E_{\text{FRET}} = 0$), loosely coupled expressome complexes TTC-LC ($E_{\text{FRET}} \sim 0.1$)³³ and coupled expressomes in state TTC-B ($E_{\text{FRET}} \sim 0.3$)^{6,7}. Targeted hybridization of DNA oligonucleotides to the intervening mRNA, results in an increase in the inter ribosome-RNA distance distribution for mRNA-46 and a complete loss of coupling for mRNA-85 and the longer mRNAs (Extended Data Fig. 4b-d) further validating our assignment. Using this classification, we plotted the fraction of coupled states (both TTC-B and TTC-LC) as a function of the intervening mRNA length and find that the fraction of coupled expressomes decreases from $65 \pm 4\%$ for mRNA-46 to $8 \pm 1\%$ for mRNA-457 (Fig. 3c and Extended Data Fig. 5b).

Fig. 3: Real-time tracking of ribosome/RNAP coupling in dependence of intervening mRNA length. **a**, DNA template design. **b**, Location of labeling sites within coupled **expressome** (pdb: 6xdq).

8. The magnitude of the effects seen for NusG and NusA are surprisingly small. The authors should state how this compares to previously published observations of coupling with these factors. Moreover, when discussing NusA vs NusG vs Nus+NusG, the authors make their arguments solely on the differences they observe in the lifetimes of the interactions. The authors should expand their discussion to include whether the FRET efficiencies change or whether structural differences in the complexes are expected. Due to their muted effects, the authors would greatly benefit from the use of mutant NusA or NusG factors which abrogate binding to either the ribosome or RNA polymerase (e.g., NusG F165A)

We assume that by “effects seen for NusG and NusA are surprisingly small” the reviewer means the coupled lifetimes originally plotted in Fig. 4c (now Fig. 5c), because the functional effect of NusG is very significant (factor presence increases the fraction of molecules that are coupled at end of transcription in our assay from 40% to almost 80%, in original Fig. 4h,i, now Fig. 5f).

The plotted “time coupled” in original Fig. 4c (now Fig. 5c), to what we think the reviewer is referring to, are mostly limited by photobleaching or the end of transcription as pointed out in the main text of the original manuscript (“37 % of the traces showed a 30S-Cy3 to RNAP-Cy5 FRET signal till the end of transcription without any uncoupling or recoupling events....However, for the remaining 63 % of the molecules, the RNAP-Cy5 signal photobleached before the end of transcription”) and therefore comparing

lifetimes is not meaningful. However, we agree that the way we originally plotted Fig. 4c (now Fig. 5c) can be confusing by placing the various factor compositions side-by-side and possibly implying that we detect differences in coupling lifetime for the various factor compositions. Therefore, we rearranged this panel to directly compare coupled versus collided, which is the intended effect to be represented (after collision, coupling becomes more transient).

Furthermore, we now also refer in main text to previous literature to support our findings of the strong effect of NusG on functional coupling (Fig. 5e,f):

conservation^{6-8,39}. Overall, our data show that RNAP and ribosome can remain physically coupled during active transcription elongation, with hundreds of nucleotides of intervening mRNA looping-out in between, and that the coupling efficiency is increased by transcription factor NusG, in agreement with previous findings that NusG is important for transcription-translation coupling^{5,34-37}.

In order to address the request to include mutant NusG into our experiments, we have repeated our experiments with NusG-F165A and in addition, with the N-terminal part of NusG, both which can only bind to RNAP but not to the ribosome. Our data confirm that both domains are needed for functional coupling but that transcription is accelerated also in the mutants as in the WT.

-We have added additional text and corresponding Figure and Extended Figures:

around 80 % in presence of both factors (Fig. 5f). In contrast, NusG-NTD³⁸ or NusG-F165A³⁴, two mutants which can interact only with the RNAP, show no increase in functional coupling (Figure 5f) but retain their ability to increase the transcription rate (Extended Data Figure 10b,c). That the coupling is mainly maintained by NusG is in agreement with a highly conserved NusG-ribosome interaction surface in contrast to the poor NusA-ribosome interface conservation^{6-8,39}. Overall, our data show that RNAP and ribosome can remain physically

Finally, we thank the reviewer for suggesting to evaluate the FRET efficiencies in presence of various Nus factor compositions because this lead to very exciting new results.

We wrote a new section and added a new additional small main Fig. 4 (including extended Data Figure):

Nus factors modulate expressome compaction

Transcription factor NusG is responsible for bridging the ribosome with the RNAP^{5,34-37} and restrains RNAP motions⁶. In contrast, just a single NusA-only bound expressome structure was obtained at low resolution at a short intervening mRNA sequence of 39 nt and suggesting loosening of the expressome structure⁷. We repeated our coupling experiments for mRNA-85 in presence of either one or both Nus factors. Our data show that in presence of NusG, irrespective of NusA, the E_{FRET} value of the coupled state increases in agreement with compaction of the expressome (Fig. 4 and Extended Data Fig. 4e). In contrast, addition of NusA alone, resulted in a loosening of the expressome structure. Overall, these findings illustrate how the two transcription factors NusA and NusG modulate expressome compaction and are in agreement with NusG being the main driver of expressome stability.

Fig. 4: Nus factors modulate expressome compaction. a, FRET distribution histograms are displayed in dependence of Nus factors for mRNA-85. Number of analyzed molecules (n) is indicated. Data was combined from 2 replicates. Uncoupled, loosely coupled and coupled states are indicated. b, Schematics for factor-dependent loosening (NusA) and compaction (NusG) of expressome structure. FRET distribution histogram shown for mRNA-85 without factors is the same as shown in Fig. 3f.

9. The authors need to include an SDS-PAGE gel image of the protein factors used in this study.

We now show an SDS-PAGE gel of all the factors in Extended Data Fig. 1j.

Referee #3 (Remarks to the Author):

The manuscript by Qureshi and Duss investigates dynamic coupling between the transcribing RNA polymerase (RNAP) and translating ribosome. Transcription-translation coupling (TTC) is thought to guide regulatory decisions in specific operons and aid uninterrupted RNA synthesis globally, by insulating RNAP from premature termination by Rho and rescuing backtracked RNAPs. Coupling has been studied for decades and several recent cryoEM and cryoET structures captured the two machines poised at defined distances between them, revealing different complex architectures, with NusA and NusG proteins acting as bridges. While these structures revealed the molecular interface between the two machines and the bridges, they lack the information on the TTC dynamics in a “real-life” scenario when these complexes are actively moving. Here, the authors established a complete TTC system, which recapitulates the known properties of both machines, and used multi-color fluorescence to follow single molecules in real time. Using single-molecule FRET with strategically positioned fluorophores, the authors can monitor each translation step by analyzing the cyclical conformational changes in the ribosome; the RNA synthesis by average time it takes RNAP to get to the end of the template; and physical interactions (direct coupling) between 70S and RNAP.

The results show that coupling (i) can occur when RNAP and ribosome are separated by hundreds of nts, even though less efficiently as the nascent RNA grows and loops out; (ii) is promoted by NusA and NusG; (iii) can persist throughout transcription; and (iv) can assist RNAP elongation; the latter result was confirmed by bulk transcription assays. Interestingly, the ribosome slows down as it approaches the stalled RNAP to avoid a hard collision, and after the ribosome rear-ends (artificially stalled) RNAP, the coupling is weakened upon escape, suggesting that a change occurs at the RNAP-ribosome interface during this collision. This change could be due to the loss of omega, seen in structures of collided machines, or a conformational change in RNAP; conformational heterogeneity has been observed in single-molecule analysis of RNA chain elongation. Even though the ribosome slows down, hard collisions may occur when RNAP is stalled for an extended period of time on the cell, and collision-induced changes could have regulatory significance.

The effect of Nus factors on the extent of coupling is inferred from the FRET signal observed between the DNA template 3' end and the 30S ribosome – this signal presumably reflects coupling at the end of the transcription cycle. Not being a FRET expert, I think it is a reasonable explanation. However, I think that in the panel a what is interpreted as DNA template dissociation is at least as likely to be RNA product release – since RNAP has a tendency to hold onto the DNA post termination.

While we agree with the reviewer that RNAP may stay on DNA after termination, we know that DNA is dissociated because we have immobilized the entire TEC via the 5' end of the nascent RNA and have labeled the DNA with two Cy3.5 dyes. Thus, loss of Cy3.5 signal demonstrates dissociation of DNA from the immobilized mRNA (see also Duss & Williamson, Nature Communications, 2018 for more details on this setup).

I am confused by Figs. 4h & i – in the presence of NusA, 40% of traces remain coupled throughout transcription, but more than 60% are coupled at the end; does NusA bring the other 20% together?

We thank the reviewer for bringing this to our attention because the value of NusA in panel i) was actually an error (the value of NusA in panel h) is correct). Apart from checking all the source data, we have also verified this by remeasuring all the data in this figure. We also agree that showing both panels h and i is confusing. In fact, they represent the same data in a slightly different way: they both show fraction of molecules that are coupled at transcription end from a total of either a) only those molecules that were coupled at the beginning of the experiment (original panel h) or from all the molecules (original panel i). We now only show original panel i) in main figure as it is the better comparison to original panel j) for the collided ribosome because for the collided ribosome no molecules are coupled at the start of the experiments (because they start as collided complexes). These are now the new panels as Fig. 5e,f (previously Fig. 4i,j):

The authors show that the coupled ribosome makes RNA chain elongation less pause prone. In particular, NusA-promoted pausing is alleviated by the ribosome, and the authors argue that NusA repositioning represents an alternative way to activate RNAP, instead of the ribosome pushing RNAP from behind (which would require close coupling). These are different processes, however. NusA-stabilized pauses are typically hairpin-dependent, consistent with the data shown here, whereas backtracking occurs when RNAP can slide back unobstructed, by RNA structures or by a closely-coupled ribosome. NusA repositioning can simply affect its ability to interact with pause hairpins. A conformational change in RNAP induced by physical coupling could make RNAP pause resistant, not only at NusA-sensitive sites but overall.

We have specified this interesting aspect by extending the discussion:

We therefore postulate that functional collisions, occurring at 28-29 nucleotides intervening mRNA, are likely less frequent events, but relevant, for example, to rescue long-lived, by back-tracking paused RNAPs or to selectively destroy non-functional transcription complexes stalled on DNA lesions¹⁴. Instead, more efficient ribosome-induced RNAP activation from NusA-dependent hairpin-stabilized pausing via long-range coupling can occur during the entire transcription process, over several hundred nucleotides separation, therefore relevant for an average *E. coli* mRNA length of 1000 nucleotides⁴⁹. Long-range physical coupling may also

These are very significant findings as the dynamic interactions of RNAP and 70S machines is monitored for the first time (as far as I know). Bridging over large RNA distances makes sense, and has been observed in other RNA transactions, and raises many questions. The most obvious, raised by the authors, is the control of Rho-dependent termination – the intervening RNA would serve as a loading platform for Rho, so inhibiting termination by mRNA masking seems not feasible. One possibility is NusG, which makes mutually exclusive contacts to Rho or the ribosome; if NusG CTD is bound to S10, it would not be available to aid Rho.

We have added this important point to the discussion:

57/85 nucleotides⁵²) while the RNAP and the ribosome are physically coupled. Whether Rho could push away the ribosome in order to reach the RNAP for terminating transcription and how this is coordinated with NusG, which can bind either Rho or the ribosome with its C-terminal domain⁵⁰, remains to be investigated.

But another possibility, supported by the authors data on 70S “activation” of RNAP is a conformational change in RNAP – as Rho requires RNAP to pause to trigger termination.

The manuscript is well written, easy to follow, and experimental design is depicted and described very clearly. The methods section is very thorough and should enable others to reproduce these experiments. The data analysis is convincing and supports the conclusions.

We thank for the reviewer for acknowledging our detailed methods section and convincing data analysis which supports the conclusions.

However, figures and legends are very busy and would benefit from simplifying. For example, in Fig. 4c,d,e etc, I would put -, A, G or AG symbols below graphs and the number of corresponding traces below these. Also, there is no need to repeat legends – if anything, not having an extra legend indicates that the two states (coupled and collided) are being compared directly. In Fig. 4d, the overlay does not help, in my opinion – the scales are very different, obviously – maybe just draw lines between left and right panels to stress how different they are.

We agree that some of the figures are busy and have tried include all the suggestions provided by the reviewer including shortening of the figure captions. Below is Fig. 5c-f (originally Fig. 4c,d,e etc).

In Fig. 3f, the scale is wrong – this should be fractions, not %
 Fig. 3g should have bars as in f – no point in making this a graph with a real x-axis, and they should be the same regardless.

Based on the other reviewer's comments, we have added new data to Fig. 3 and therefore have substantially changed it.

Original Fig. 3g is now Extended Data Fig. 5b with bars as suggested by the reviewer:

References need to be checked carefully. Ref 8 is the same as Ref 29; there may be others

They have been corrected.

We thank the three reviewers for their excellent comments raised during the entire revision process, which significantly increased the quality of the manuscript and we are excited that they are all satisfied with our revisions and suggest publication.

Referees' comments:

Referee #1 (Remarks to the Author):

Thank you very much for sending the revised manuscript by Qureshi and Duss. The authors have addressed all my concerns and I suggest publication.

Referee #2 (Remarks to the Author):

This is an impressive, impactful study, and the authors have adequately addressed nearly all of our concerns. From our perspective, the manuscript is ready for publication once the authors have addressed two remaining minor points which can be addressed entirely through textual changes to the manuscript:

1. We would encourage the authors not to apply the 'visual inspection' correction to the hidden Markov models. By looking at the data before and after this correction, the correction does not seem to have any effect on the results of the analysis, the interpretation, or the conclusions. Thus, there doesn't seem to be a reasonable need for this correction. Alternatively, if the authors decide to keep this correction, then they should explicitly note the number of transitions that are altered by applying the correction for each experiment under each experimental condition. The 'TL' correction, on the other hand, which is applied with clear kinetic constraints, is appropriate and does seem to have an impact, albeit minor, on the analysis.

We thank the reviewer for pointing this out. We decide to keep the visual inspection correction, as at times automated assignment was suboptimal, e.g. FRET intensity changes were not anti-correlated and thus, the assignment wrong. As the reviewer suggested, we determined the slight changes in the number of transitions for the relevant experiments and have added a phrase into the Methods section:

for Cy3B this manual correction reduced the final average transitions per trace from 14 to 12 for elongating conditions and from 5 to 4 for colliding conditions. Second, traces were corrected

2. In the revised manuscript, the fact that the traces are smoothed is now described in the Methods. However, in addition to the description in the Methods, the authors should note that the traces were smoothed in all figure legends where smoothed traces are presented.

We have now mentioned that traces were smoothed in every relevant figure caption.

Referee #3 (Remarks to the Author):

The revised manuscript by Qureshi and Dus has been significantly improved by incorporation of new experiments, providing a more detailed methods section, improving old and adding new figures, and elaborating on a number of points that were somewhat unclear in the original manuscript.

In my opinion, this is an excellent study that would be of interest to a very broad audience, both because it addresses an important biological question at the interface of two steps in gene expression and because it describes powerful multi-color single-molecule fluorescence assays that can be adapted to studies of other cross-talking macromolecular complexes. I think that the revised manuscript is suitable for publication in Nature.

That said, there are a few rough spots in the manuscript that could benefit from editing. I list just a few below and suggests that the authors go through the text slowly to find others.

L139 Add "site" after "active"

L221 delete comma after mRNA

L420 maybe write "but failure to detect this state in vivo in absence of antibiotics was interpreted as evidence that the collisions are transient"

L423 delete "from pausing"

L427 delete "increasingly"

L430 maybe "to rescue arrested backtracked RNAPs"

Of note, NusA-stabilized hairpin pauses are rare – their prominence in the literature reflects the fact that they were the first type of physiologically-relevant pauses identified.

As suggested by the reviewer, we have done the following corrections to the main text and have read the entire text again correcting for rough spots:

P-site to RNAP active site, Fig. 2g)^{6,7,11}. Delivery of all translation elongation factors together

|201 intervening mRNA₅ results in an increase in the inter ribosome-RNA distance distribution for

355 The functional relevance of the collided state is debated^{7,36,37}. It was structurally characterized
356 by several independent groups⁶⁻¹⁰ ~~and its inability to be~~ but failure to detect this state in vivo
357 in absence of antibiotics⁸ was interpreted as evidence that the collisions are transient³⁷.
358 However, it was also argued that these collisions are non-physiological⁷. Ribosome-RNAP
359 collisions are a possible mechanism in vitro for the ribosome to rescue a back-tracked RNAP
360 ~~from pausing~~^{10-12,14} but functionally only at a separation of 28-29 nucleotides of intervening
361 mRNA between RNAP active site and ribosome P-site^{10,14}. Structural studies demonstrate that
362 a substantial fraction of the expressomes adopt a collided conformation at a distance of 37
363 nucleotides or smaller^{6,7} and our data show that the ribosome starts to ~~increasingly~~ slow down
364 at an intervening mRNA distance in which the expressome preferentially adopts a collided
365 conformation. We therefore postulate that functional collisions, occurring at 28-29 nucleotides
366 intervening mRNA, are likely less frequent events, but relevant, for example, to rescue ~~long-~~
367 ~~lived, by back-tracking paused RNAPs~~ arrested backtracked RNAPs or to selectively destroy
368 non-functional transcription complexes stalled on DNA lesions¹⁴. Instead, more efficient
369 ribosome-induced RNAP activation from NusA-dependent hairpin-stabilized pausing via long-
370 range coupling can occur during the entire transcription process, over several hundred
371 nucleotides separation, therefore relevant for an average *E. coli* mRNA length of 1000
372 nucleotides⁴⁹. Long-range physical coupling may also occur between the RNAP and a non-